# Programmable antisense oligomers for phage functional genomics

Milan Gerovac[1,2,4], Leandro Buhlmann[2], Yan Zhu[1], Svetlana Đurica-Mitić[2], Valentin Rech[2], Samuel Carien[2], Tom Gräfenhan[3], Linda Popella[2] & Jörg Vogel[1,2 ✉]

Bacteriophages are the most abundant entities on earth and exhibit vast genetic and phenotypic diversity. Exploitation of this largely unexplored molecular space requires identification and functional characterization of genes that act at the phage–host interface. So far, this has been restricted to few model phage–host systems that are amenable to genetic manipulation. Here, to overcome this limitation, we introduce a non-genetic mRNA targeting approach using exogenous delivery of programmable antisense oligomers to silence genes of DNA and RNA phages. A systematic knockdown screen of core and accessory genes of the nucleus-forming jumbo phage ΦKZ, coupled to RNA-sequencing and microscopy analyses, reveals previously unrecognized proteins that are essential for phage propagation and that, upon silencing, elicit distinct phenotypes at the level of the phage and host response. One of these factors is the RNase H-like protein ΦKZ155 (also known as Nlp2), which acts at a major decision point during infection, linking the formation of the protective phage nucleus to phage genome amplification. This non-genetic antisense oligomer-based gene silencing method promises to be a versatile tool for molecular discovery in phage biology, will help to elucidate defence and anti-defence mechanisms in non-model phage–host pairs, and offers potential for optimizing phage therapy and biotechnological procedures.

The growing interest in bacteriophages is driven by the richness of unexplored genes that emerge in the phage–host conflict[1,2]. Targeted mapping and characterization of these genes is key to understanding a phage's infection cycle and its ability to counter host defences, but the diversity and genetic intractability of many phages and their hosts pose major challenges. Phage functional genomics approaches have used targeted inhibition of viral gene expression through, for example, small RNAs[3,4] or CRISPR–Cas technology[5–7]. However, the limited genetic tractability of many phage–host pairs due to defence systems that target foreign DNA[1,2] or phage-mediated neutralization of Cas enzymes[8,9] remains a barrier, even for intensely studied phages such as the model jumbo phage ΦKZ of the major human pathogen *Pseudomonas aeruginosa*.

ΦKZ-like phages are of interest not only because of their potential for phage therapy of recalcitrant infections[10], but also owing to their complex infection cycle, which features the sequential formation of membrane- and protein-bound compartments. ΦKZ injects its 280-kb dsDNA genome together with a virion RNA polymerase (vRNAP) in an early phage infection (EPI) vesicle for immediate phage gene transcription[11–13]. Subsequently, a 'phage nucleus' forms, in which the non-virion RNA polymerase (nvRNAP) continues transcription[14,15]. The phage genome is then replicated and loaded into phage capsids (Extended Data Fig. 1a). Several conserved phage factors enable the formation and organization of the phage nucleus—for example, the nuclear shell protein chimallin (ChmA)[14,16,17], PicA (also known as Imp1), which mediates cargo trafficking into the phage nucleus[18,19], and the tubulin-like protein PhuZ, which centres the phage nucleus in the middle of the cell and helps to traffic newly assembled capsids[14,20,21]. Nonetheless, it remains unclear how many of the approximately 400 annotated protein-coding genes[22] of ΦKZ have an essential role in the phage infection cycle. Notably, the phage nucleus shields the phage genome from CRISPR–Cas and restriction enzymes[23,24], which makes targeted inhibition of ΦKZ genes challenging.

Here we present a straightforward and broadly applicable non-genetic route to assess gene essentiality and function in phage–host interactions, utilizing exogenous delivery of synthetic antisense oligomers (ASOs) via a cell-penetrating peptide (CPP) into the bacterial cytosol. Such ASOs are typically 9–12 nucleobases in length and can be programmed to selectively inhibit protein synthesis by binding to the ribosome binding site (RBS) of a target mRNA of interest[25,26]. Seeking to identify key factors in the intricate infection cycle of ΦKZ in *P. aeruginosa*, we performed systematic inhibition of phage mRNA translation coupled with phenotypic, transcriptome and proteome analyses. Our results suggest that more ΦKZ proteins than previously appreciated act to steer the phage replication cycle, including the conserved ΦKZ155 protein, which acts at a key decision point prior to ΦKZ genome amplification.

## ASOs can silence phage transcripts

Like other bacterial species, *P. aeruginosa* is amenable to ASO-induced gene silencing, as demonstrated by antimicrobial activity of ASOs

[1]Helmholtz Institute for RNA-based Infection Research (HIRI), Helmholtz Centre for Infection Research (HZI), Würzburg, Germany. [2]Institute for Molecular Infection Biology (IMIB), Faculty of Medicine, University of Würzburg, Würzburg, Germany. [3]Core Unit Systems Medicine, University of Würzburg, Würzburg, Germany. [4]Present address: Helmholtz Centre for Infection Research (HZI), Braunschweig, Germany. ✉e-mail: joerg.vogel@uni-wuerzburg.de

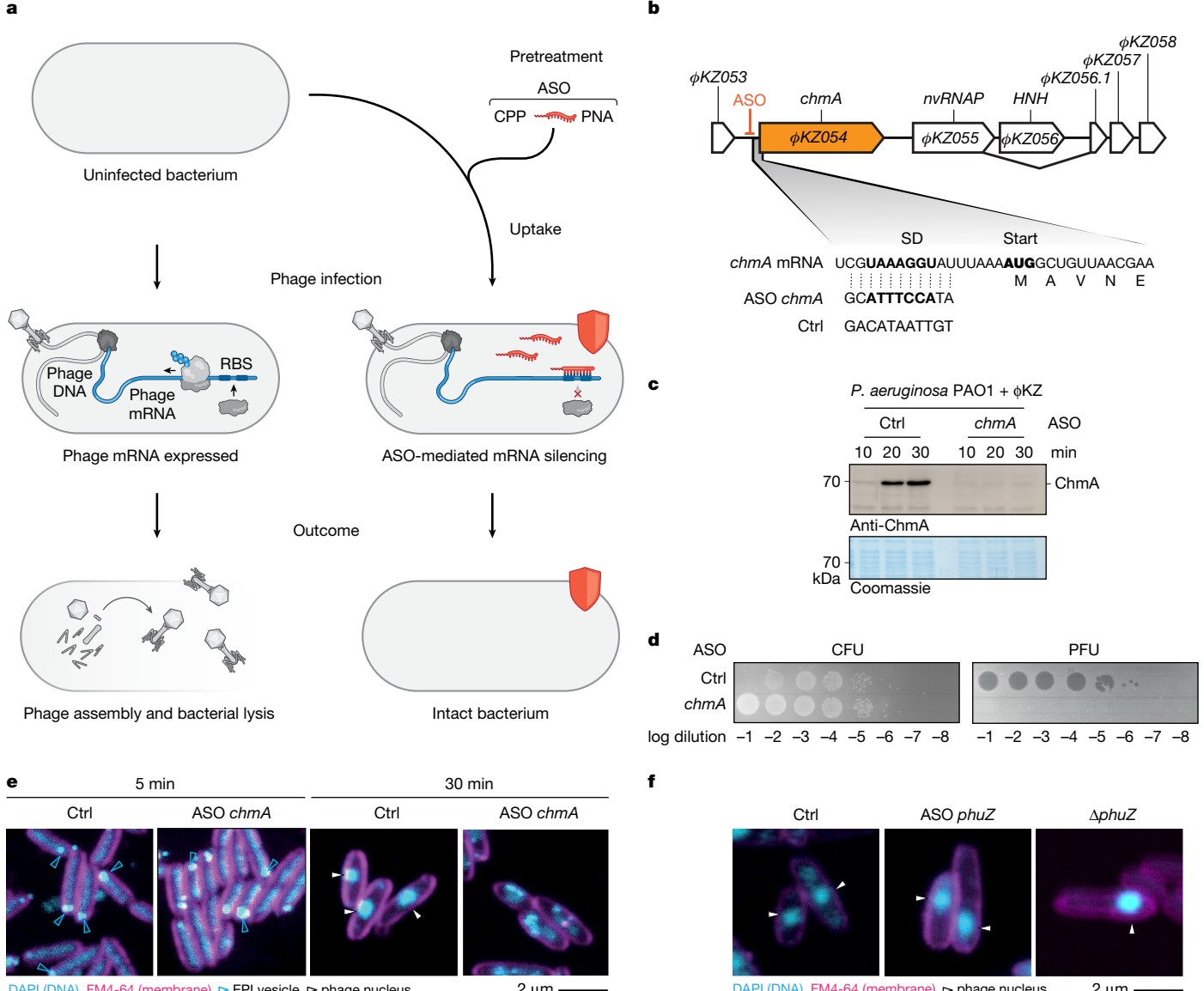

**Fig. 1 | ASOs silence ΦKZ transcripts in *Pseudomonas*. a**, ASOs are delivered into bacterial cells by a CPP. Upon phage infection, ASOs bind the target phage mRNA at the RBS, preventing translation. If the encoded protein is essential, phage propagation is halted, protecting the bacterium (red shield). **b**, The *chmA* locus and translation initiation region. ASOs targeting the RBS and the non-targeting control ASO (Ctrl) are depicted. **c**, PAO1 cells were pretreated with 6 µM ASOs for 30 min before ΦKZ infection at an MOI of 5. Cells were collected at the indicated time points and ChmA was quantified by immunoblotting. A Coomassie-stained gel served as loading control. One representative example of four independent experiments is shown. **d**, PAO1 cells were pretreated for 30 min with 6 µM ASOs. After ΦKZ infection at an MOI of 0.0001, cells were incubated for 180 min. The phage–cell suspension was spotted on LB plates to assess the number of CFUs and on LB plates with a PAO1 lawn to assess PFUs. A high phage titre causes lysis of nearby bacterial cells after spotting, leading to a lack of CFUs at low dilutions in the control. A representative result of two independent experiments is shown. **e**, PAO1 cells were pretreated with 8 µM ASOs for 30 min and infected with ΦKZ at an MOI of 10, followed by chemical crosslinking and staining of membranes (FM4-64) and DNA (DAPI) at either 5 or 30 min p.i. **f**, PAO1 cells were pretreated with 8 µM ASO targeting the phage spindle gene *phuZ* or a control ASO for 30 min, infected with an MOI of 10 and imaged as in **e**. In parallel, PAO1 cells were infected with a *phuZ* deletion phage at an MOI of 0.001. Quantification in Extended Data Fig. 2a. Uncropped images of blots are presented in Supplementary Fig. 1.

directed at mRNAs of essential proteins[27,28]. This previous work includes ASOs based on peptide nucleic acid (PNA)[27], whose pseudopeptide backbone protects it from nucleolytic and proteolytic degradation. As ΦKZ mRNAs are translated by host ribosomes, we reasoned that they should also be amenable to ASO-mediated silencing (Fig. 1a). Yet, it was unclear whether the rapid overloading of the transcription–translation machinery in infected cells or the phage nucleus would interfere with ASO activity. To establish proof of concept for phage transcript inhibition with antisense PNA, we designed an 11mer ASO to target the Shine–Dalgarno (SD) sequence of *chmA* mRNA (gene *ΦKZ054*; Fig. 1b),

which encodes the nuclear shell protein[14]. An ASO that does not target any specific gene served as non-targeting control. Western blot analysis of *P. aeruginosa* cells preincubated with these ASOs showed that 20 min post infection (p.i.) with ΦKZ—that is, within the first round of replication—the *chmA*-targeting ASO prevented ChmA protein synthesis, even at a high multiplicity of infection (MOI) of 5 (Fig. 1c).

ChmA is an essential phage protein, so its gene cannot be deleted in ΦKZ[29]. ASO targeting of its mRNA should phenocopy the essentiality of this gene. We therefore assessed cell survival by quantifying colony-forming units (CFUs) and estimated phage plaque efficiency

by quantifying plaque-forming units (PFUs) after multiple rounds of replication. This experimental set-up enables us to amplify the effects on phage progeny, to sample delayed replication times and to score effects that become apparent only in the next infection round. Notably, the anti-*chmA* ASO strongly reduced phage progeny, eliminating plaque formation (Fig. 1d). This allowed us to systematically optimize ASO variables (carrier peptide and length) and experimental conditions (time pre- or post-treatment, ASO toxicity, concentration and MOI) to increase the dynamic range of the phage plaque efficiency score (Extended Data Fig. 1b–i). On the basis of these optimization experiments, we recommend the following conditions for ΦKZ and *P. aeruginosa*: an 11mer ASO with an (RXR)$_4$XB carrier peptide used at 6 μM concentration in Mueller–Hinton medium with 30 min ASO pre-treatment. We observed the largest dynamic range for phage plaques for an infection at an MOI of 0.0001 after three replication rounds. This corresponds to spotting after 3 h p.i. and resulted in nearly complete host lysis. The extent of CFU clearing is more evident at higher phage titres and longer replication times (Extended Data Fig. 1i). Using these conditions at an MOI of 10 also enables screening of knockdown effects within less than one replication round.

Next, we monitored the effects of ASO on progression of the phage replication cycle by imaging the formation of phage-derived cellular compartments. At 5 min p.i., the EPI vesicle was visible in both control and anti-*chmA* ASO-treated cells (Fig. 1e), indicating successful infection. At 30 min p.i., the phage nucleus had formed in the control sample, evident as a central DNA-containing structure, but not in the anti-*chmA* ASO sample. Instead, we observed smaller DNA-containing structures, presumably multiple EPI vesicles resulting from the high MOI of 10 needed for a synchronized infection. Thus, in the absence of ChmA, no phage nucleus forms and this arrests the phage replication cycle. Combined with the complete loss of plaque formation (Fig. 1d), this implies that the EPI vesicle cannot support phage replication. Similar observations have recently been made with CRISPR–Cas-based inhibition of *chmA* mRNA in the *Escherichia coli* phage Goslar[7,13].

To compare the effect of ASO-mediated knockdown of a phage mRNA with that of deleting the corresponding gene, we silenced the non-essential *phuZ* mRNA to prevent mid-cell positioning of the phage nucleus by the PhuZ spindle protein[20]. Bacteria pretreated with an anti-*phuZ* ASO exhibited the loss of phage nucleus centring observed with a Δ*phuZ* phage[29], effectively phenocopying the phage gene deletion (Fig. 1f; quantification in Extended Data Fig. 2a).

## ASO design and specificity

Our previous global analyses of ASO-mediated knockdown in different bacteria[30–33] culminated in MASON[34], an ASO design algorithm that predicts effective and selective ASO inhibitors of protein synthesis based on various criteria such as melting temperature ($T_m$), low self-complementarity, low purine percentage and mRNA target site localization. MASON considers central mismatches as detrimental to ASO efficacy[34], and as expected, two central mismatches abrogated the activity of the *chmA* ASO (Extended Data Fig. 2b). To establish effective ASO sites within a phage 5′ mRNA region, we tiled the *chmA* mRNA in the −37 to +44-nt window relative to its AUG start codon. Of note, all 19 ASOs that bind at or close to the SD sequence or AUG with a $T_m$ between 35 and 58 °C strongly inhibited ChmA synthesis and phage plaque efficiency (Extended Data Fig. 2c). Thus, multiple effective ASOs can be designed for a given target to mitigate the risk of off-targeting and false-positive readout of phenotypes.

To address ASO specificity and potential off-target effects at the protein level, we performed proteomics on ΦKZ-infected *P. aeruginosa* pretreated with ASOs against three different mRNAs: *chmA*, the nvRNAP transcript *ΦKZ055* and *picA* (gene *ΦKZ069*) (Supplementary Data 1). Sampling a 10 min time course of ΦKZ infection, we did not observe substantial ASO-dependent changes in the amounts of the 1,130 *P. aeruginosa* proteins detected with high confidence, which argues against widespread off-target effects in the host (Extended Data Fig. 3a). Similarly, we analysed the phage proteome at 7.5 and 10 min p.i.—that is, when phage proteins became abundant. We observed a specific reduction in the levels of ChmA, nvRNAP or PicA (Extended Data Fig. 3b) without strong effects on other phage proteins. Notably, all three targets are co-transcribed with additional genes, but the respective ASOs inhibited only the targeted cistron. For example, the *chmA* ASO suppressed synthesis of ChmA but not that of the nvRNAP encoded by the downstream cistron (Extended Data Fig. 3c–e). Thus, ASOs can be used for selective suppression of protein synthesis within polycistronic transcripts.

## Versatile ASO applications in phage biology

ASOs are particularly attractive for phage–host interaction studies in genetically intractable bacteria, targeting either phage or host genes. Many bacterial strains collected from patients are refractory to genome modification techniques that require stable transformation with DNA[35], such as the genomically diverse *P. aeruginosa* isolates PaLo8, PaLo9, PaLo39 and PaLo44 (PRJNA731114). Nonetheless, ASOs targeting *chmA* effectively inhibited phage plaque and nucleus formation in these clinical isolates (Fig. 2a and Extended Data Fig. 4a–c). Thus, ASOs can bypass the need for genetic tractability in functional studies of essential phage factors.

Host-encoded defence systems are a major barrier to phage replication, but their genomic disruption is challenging. To test ASOs as alternative inhibitors, we selected the JukAB defence system in *P. aeruginosa* strain PA14, which prevents early phage transcription, DNA replication and nucleus assembly[36]. Owing to constitutive expression of the *jukAB* locus, anti-*jukA* ASOs were added 2.5 h before infection to allow more time for JukA protein depletion before ΦKZ exposure. ASO-mediated silencing of *jukA* mRNA rendered *P. aeruginosa* PA14 bacteria susceptible to ΦKZ (Fig. 2b), indicating successful inactivation of the defence system. In the arms race between phages and bacteria, phages constantly evolve counter-defence mechanisms to overcome host barriers. ΦKZ encodes the ribosome-associated protein ΦKZ014, which appears to circumvent a defence system that is specific to the clinical isolate PaLo44[37]. Although the *ΦKZ014* gene can be disrupted when ΦKZ is propagated in the *P. aeruginosa* PAO1 host, the resultant ΔΦKZ014 phage does not produce progeny in the PaLo44 isolate[37]. In agreement with this strain tropism, ASO-mediated knockdown of the *ΦKZ014* mRNA impaired ΦKZ replication in PaLo44 but not in PAO1 (Fig. 2c), effectively phenocopying the corresponding phage gene mutant. Thus, ASOs can be used as tools to dissect defence and anti-defence loci in relevant native settings and ultimately to achieve phage propagation in non-permissive strains.

To further explore the versatility of ASOs in phage biology, we tested distinct phage–host pairs. Targeting *chmA* (also known as *gp296*) in the RAY phage[38], which infects the Gram-negative plant pathogen *Pantoea agglomerans* (family *Erwiniaceae*), effectively abolished plaque formation (Extended Data Fig. 4d). Similarly, we established proof of principle in a Gram-positive species, which has a very different envelope structure. Targeting expression of DNA polymerase (gene *31*) of the *Bacillus subtilis* phage SPO1 caused a 100-fold reduction in plaque efficiency (Extended Data Fig. 4e). Finally, to prove applicability beyond phages with a DNA genome, we targeted different genes of the RNA phage PP7, which infects *P. aeruginosa* PAO1 (ref. 39). Using an adjusted regime, ASOs targeting the PP7 genes *rep* (RNA-dependent RNA polymerase) or *lys* (lysis) completely abrogated plaque formation (Extended Data Fig. 4f). Thus, mRNA-targeting ASOs can be applied to diverse hosts and phages and are agnostic to the type of phage genome.

## ASO screen for essential phage proteins

The programmable nature of ASOs allows larger screens for genes that are crucial for phage propagation. Of the annotated 377 ΦKZ genes,

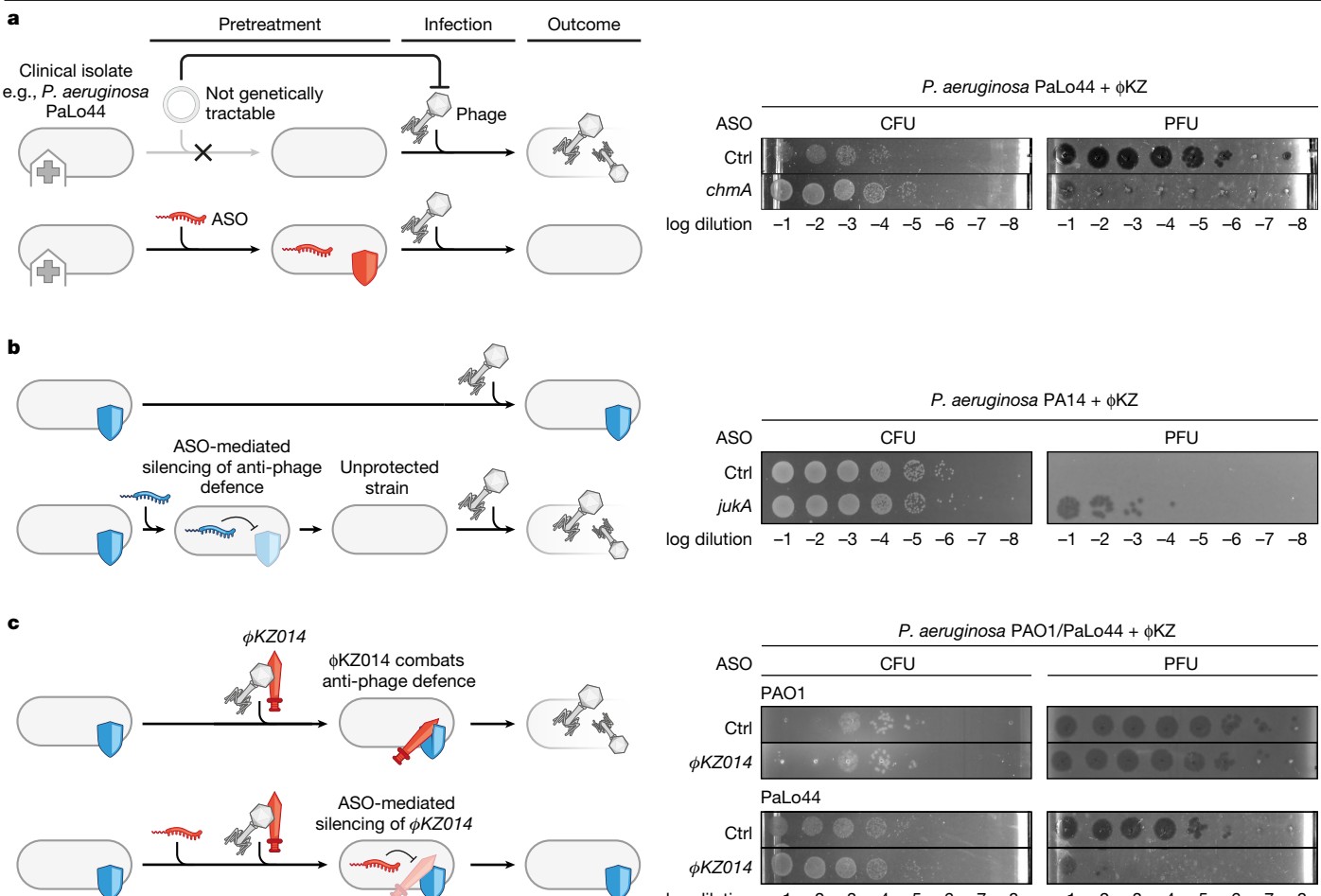

**Fig. 2 | ASO applications in phage biology. a–c**, Left, schematic representation of potential ASO applications, such as targeting clinical isolates (**a**), sensitizing bacteria to phage infection by silencing bacterial anti-phage defence systems (**b**), or protecting bacteria from phage infection by silencing phage genes that affect anti-phage defence systems (**c**). Anti-phage defence systems are depicted as shields, and phage proteins that overcome bacterial defences are depicted as swords. **a**, Right, PaLo44 cells were pretreated with 6 µM ASO targeting *chmA* for 30 min. Cells were infected with ΦKZ at an MOI of 0.0001 and incubated for 180 min followed by CFU and PFU quantification. **b**, Right, PA14 cells were inoculated, grown for 30 min and pretreated with 6 µM ASO targeting *jukA* for 150 min to ensure JukA depletion before infection. Cells were infected with ΦKZ at an MOI of 0.0001 and incubated for 180 min followed by CFU and PFU quantification. **c**, Right, PAO1 and PaLo44 cells were pretreated with 6 µM ASO targeting *ΦKZO14* for 30 min. Cells were infected with ΦKZ at an MOI of 0.0001 and incubated for 180 min followed by CFU and PFU quantification. Representative results of two independent experiments (**a–c**).

85% have no predicted function[22,40]. This paucity of functional knowledge extends to the smaller core genome of *Chimalliviridae*, which consists of seven blocks of genes with probably interlinked functions, plus five independent genes[38]. To discover additional essential genes for ΦKZ propagation, we screened 75 core and annotated genes (omitting most of the structural genes) using ASOs and CFU and PFU readout in *P. aeruginosa* PAO1.

Overall, ASO-mediated knockdown of one-third of these genes (24) led to a strong effect (++ or +++) on phage plaque efficiency with multiple logs of PFU reduction and CFU recovery (Table 1, Extended Data Fig. 5a and Supplementary Data 2). As expected, several mRNAs of known essential phage factors, such as *chmA* (*ΦKZO54*), nvRNAP subunits (*ΦKZO55* and *-068*) and the major head protein (*ΦKZ120*), produced strong effects (Extended Data Fig. 5b). Similarly, ASOs against the mRNA of the phage nucleus import protein PicA (*ΦKZO69*)[18,19] abrogated plaque formation. Moreover, several uncharacterized genes produced strong plague formation phenotypes upon ASO targeting— for example, the *ΦKZO49* gene, which encodes a SH3 domain protein interactor of the phage DNA polymerase[41] (Extended Data Fig. 5b). Of note, transcript abundance and protein levels of these hits varied substantially, indicating that targeting of highly expressed genes with ASOs is possible (Supplementary Table 1).

Knockdown of other genes produced milder effects—such as reducing plaque formation by only two to three orders of magnitude—but nevertheless protected the host from lysis, as judged by CFU counts. An example is *ΦKZ144*, which encodes an endolysine for final phage release from the cell[42] (Extended Data Fig. 5b). Systematic western blot analysis revealed 11 additional hits for which ASO-mediated knockdown negatively affected ChmA protein accumulation without substantial changes in the CFU and PFU counts (Table 1). These include genes encoding a putative RAD2/SF2 helicase (*ΦKZO75*), a predicted DEAD/DEAH box helicase (*ΦKZ203*), the macro domain-containing protein ΦKZ104, a virion RNA polymerase subunit (*ΦKZ176*)[43], as well as other uncharacterized core and non-core genes. We predict that these genes benefit phage fitness in more competitive growth conditions or in *P. aeruginosa* hosts with different defence systems. The vRNAP subunit ΦKZ176 might be a special case, as this protein is injected into the host cell upon attack—thus, the ASO-mediated mRNA inhibition will reduce ΦKZ176 protein levels only in subsequent infection rounds. This, in turn, will then reduce ChmA levels, as early transcription of *chmA* from the EPI vesicle is vRNAP-dependent[13,15,44]. In summary, our screen yielded 56 phage genes whose knockdown caused varying effects on phage propagation, offering promising leads for in-depth studies using ASO treatment in combination with phenotypic or multi-omics readouts.

## Table 1 | Top hits from the ASO screen

| Annotated gene function | Locus tag ΦKZ | PFU/CFU | ChmA |
|---|---|---|---|
| **Core genome** | | | |
| ChmB nuclear shell protein | 002 | + | + |
| Terminase | 025 | | + |
| PhuZ spindle protein | 039 | ++ | ++ |
| Uncharacterized | 042 | +++ | +++ |
| Uncharacterized | 049 | ++ | +++ |
| DNA polymerase | 050 | ++ | ++ |
| Uncharacterized | 052 | + | ++ |
| ChmA major nuclear shell protein | 054 | +++ | +++ |
| RNA polymerase β subunit | 055 | +++ | +++ |
| Uncharacterized | 059 | ++ | +++ |
| Nuclease | 065 | ++ | ++ |
| Uncharacterized | 066 | + | |
| ChmC RNA-binding protein | 067 | ++ | +++ |
| Sigma factor | 068 | ++ | +++ |
| PicA nuclear shell import protein | 069 | +++ | +++ |
| RAD2/SF2 helicase | 075 | | ++ |
| Uncharacterized | 079 | | + |
| DNA polymerase | 082 | + | +++ |
| Uncharacterized | 089 | ++ | ++ |
| Major capsid protein | 120 | +++ | +++ |
| Uncharacterized | 122 | + | + |
| Virion protein | 129 | | + |
| Uncharacterized | 147 | | ++ |
| Uncharacterized | 153 | | + |
| RNase HI domain protein | 155 | +++ | +++ |
| Uncharacterized | 161 | | ++ |
| SbcC-like nuclease | 165 | + | + |
| Uncharacterized | 171 | ++ | ++ |
| Uncharacterized | 174 | +++ | +++ |
| Virion RNA polymerase subunit | 176 | + | +++ |
| Uncharacterized | 177 | +++ | +++ |
| **Non-core genome** | | | |
| Ribosome binding protein[a] | 014 | +++ | +++ |
| Dip RNA degradasome inhibitor | 037 | + | ++ |
| HNH nuclease | 056 | + | ++ |
| Uncharacterized | 064 | + | +++ |
| HNH nuclease | 072 | +++ | ++ |
| Head protein | 094 | + | + |
| Macro domain protein | 104 | | + |
| Virion protein | 119 | ++ | ++ |
| Uncharacterized | 124 | + | ++ |
| Virion protein | 126 | | + |
| Endolysine | 144 | ++ | +++ |
| Uncharacterized | 151 | – | – |
| Uncharacterized | 154 | ++ | ++ |
| Uncharacterized | 169 | + | ++ |
| Uncharacterized | 186 | +++ | +++ |
| RNase H | 199 | + | + |
| DEAD/DEAH box helicase | 203 | | + |
| AAA family ATPase | 208 | | + |
| Continued | | | |

| Annotated gene function | Locus tag ΦKZ | PFU/CFU | ChmA |
|---|---|---|---|
| Thymidylate synthase | 235 | + | + |
| Nuclease | 272 | | + |
| Uncharacterized | 283 | | + |
| Uncharacterized | 286 | | ++ |
| HNH nuclease | 296 | | + |
| Ribonucleotide-PP reductase | 305 | ++ | +++ |
| Ribonucleotide reductase | 306 | – | – |

Effects on PFU/CFU and ChmA levels for each respective knockdown are depicted as no effect (), weak (+), effective (++) or very effective (+++) plaque or ChmA level reduction and increased plaque or ChmA levels (–); for details see Supplementary Table 1 and Supplementary Data 2. A minimum of two ASOs were tested per gene, and the stronger effect is shown. [a]*ΦKZ014* knockdown caused a reduction in plaques only in the PaLo44 strain.

Indirect effects of ASOs were also observed: 22 out of 176 tested ASOs were generally toxic to *P. aeruginosa* (Extended Data Fig. 5c). Other ASOs increased PFUs by one order of magnitude, indicating that the targeted proteins, such as ΦKZ151, act to counter-regulate the infection or alter host vulnerability to the phage (Extended Data Fig. 5c). Furthermore, four ASOs induced plaques without phage treatment, perhaps by activating a Pf1-like prophage[45] (Extended Data Fig. 5c).

## Molecular phenotypes of phage genes

Phage infection is a fine-tuned process, and not all involved factors will yield a macroscopic phenotype such as altered CFU and PFU counts upon ASO knockdown. We reasoned that a global method such as RNA sequencing (RNA-seq) would detect additional 'molecular phenotypes', defined as specific transcriptional dysregulation in infected cells[46,47]. To establish this approach for ΦKZ, we performed RNA-seq after *chmA* knockdown, which arrests the infection cycle at the level of the EPI vesicle (Fig. 1e and ref. 13). Total bacterial RNA was extracted at 10, 15, 20, 25 and 35 min after ΦKZ infection and analysed by RNA-seq (Supplementary Data 3). A principal components analysis showed different trajectories for the non-targeting and targeting condition (Fig. 3a). In agreement with previous work[37,44,48], phage transcripts accounted for around 45% of all coding sequence reads at 10 min p.i., increasing to around 70% at 35 min p.i. (Extended Data Fig. 6a,b). This was similar for both conditions, but ASO-mediated knockdown of *chmA* resulted in almost no RNA-seq reads from middle/late phage genes (seen from 20 min p.i. onward), indicating that phage nucleus formation is required for the transcription of these genes by the nvRNAP (Fig. 3b and Extended Data Fig. 6b–d, classes E and F). By contrast, early phage transcripts showed prolonged expression (Extended Data Fig. 6b,d, classes A and B), possibly because vRNAP-driven transcription continues. *chmA* knockdown also resulted in a loss of phage-encoded tRNAs (Extended Data Fig. 6c), consistent with their dependence on the nvRNAP[44]. Collectively, these data support a model in which following phage genome transfer from the EPI vesicle to the phage nucleus, the nvRNAP takes over phage transcription with concomitant cessation of vRNAP activity[11,13,15,44,49].

The *chmA* mRNA itself was unaffected by the ASO treatment, suggesting that its translational inhibition does not entail mRNA depletion (Fig. 3b). This would also explain why ASOs that target polycistronic phage genes are cistron-specific (Extended Data Fig. 3c–e). However, accelerated mRNA decay was observed with several ASO targets in *E. coli*[31], highlighting the importance of monitoring levels of individual mRNAs when targeting co-transcribed genes.

Our ability to selectively prevent ChmA protein synthesis also allowed us to determine the extent to which phage nucleus formation prevents a late host response. Analysing host transcript changes with the control ASO, we did not observe a substantial transcriptional response at later stages of infection compared with 10 min p.i., when host takeover is

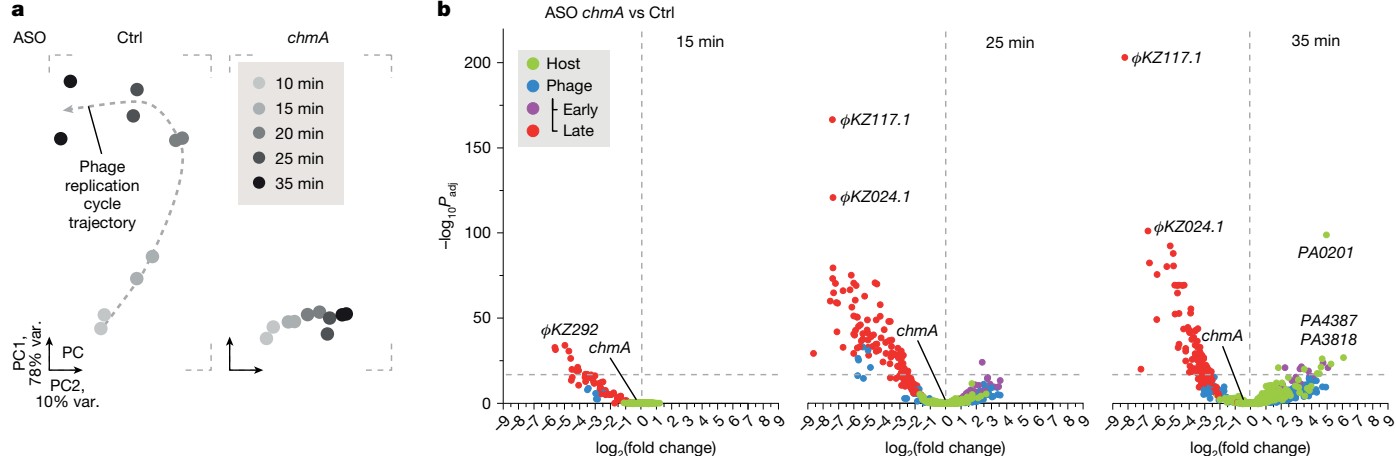

**Fig. 3 | Transcriptional response after knockdown of ChmA. a**, PAO1 cells were pretreated with 6 µM ASO targeting *chmA* for 30 min. Cells were infected with ΦKZ at an MOI of 5 and incubated for the indicated times post infection followed by RNA-seq. To reduce the dimensionality of the dataset, we projected the data based on transcript abundance of each annotated gene on two dimensions via principal components analysis. Each dot represents an independent experiment. Var., variance. **b**, Volcano plot of the log₂-transformed fold change (log₂(fold change)) of transcripts of samples treated with the ASO targeting *chmA* compared with the control ASO at the indicated time points. Two independent replicates were merged by geometrical averaging and *P* values were calculated by the Wald test (two-sided) using DESeq2.

thought to be complete (Extended Data Fig. 6a). By contrast, knockdown of *chmA* resulted in higher expression of 13 out of approximately 3,637 detected *P. aeruginosa* transcripts at 35 min p.i. (Fig. 3b). Among these genes were *PA0201*, which encodes a hypothetical hydrolase; the gene encoding the membrane protein FxsA, which is linked to phage exclusion and may prevent superinfection[50,51]; and *PA3818*, which encodes the type III secretion system regulator SuhB, which is important for *Pseudomonas* virulence[52]. Thus, even if no phage nucleus forms, the host response remains limited, demonstrating the protective nature of the EPI vesicle.

Next, we performed RNA-seq analysis of phage and host transcripts after ASO-mediated silencing of 56 positive hits from our screen (Supplementary Data 4). At 15 min p.i., we observed diverse effects on phage transcript levels of genomic islands that did not cluster clearly (Extended Data Fig. 7a,b). At 30 min p.i., ASOs that depleted ChmA, the nvRNAP subunit ΦKZ055, the nvRNAP sigma factor ΦKZ068, PicA (gene *ΦKZ069*), ΦKZ122, ΦKZ124 and ΦKZ151 produced strong effects on phage transcription (Supplementary Data 4). ChmA, nvRNAP and PicA stood out because they strongly clustered based on ΦKZ transcript level alterations (Fig. 4a and Extended Data Fig. 7c). Other *t*-distributed stochastic neighbour embedding (*t*-SNE) clusters exhibited more subtle transcriptional effects in the complementary hierarchical clustering (Extended Data Fig. 7c). Of note, reducing the levels of PicA resulted in a similar RNA-seq profile to loss of ChmA and nvRNAP, supporting the notion that PicA is the main nuclear import factor for newly synthesized nvRNAP subunits[18,19].

On the host side, we observed diverse responses upon knockdown of different phage genes (Supplementary Data 4, Extended Data Fig. 7d–g). For example, knockdown of *ΦKZ174* not only abrogated plaque formation, but also was accompanied by lower expression of almost 5% of all detected *P. aeruginosa* genes. Many of these genes serve metabolic functions, as if the ΦKZ174 protein acts to prevent infected bacteria from metabolic shutdown. Knockdown of *ΦKZ082* did not change the PAO1 transcriptome but induced transcription of the Pf4 prophage locus (Extended Data Fig. 7h). As awakening of the Pf4 filamentous bacteriophage can result in partial lysis of the bacterial population[53], ΦKZ might use the ΦKZ082 protein to suppress the Pf4 prophage to ensure its own propagation. In summary, coupling ASO-mediated knockdown to RNA-seq revealed distinct transcriptional profiles of perturbed ΦKZ genes and predicted clusters of ΦKZ factors with different perturbation modes.

## ΦKZ155 is essential for phage replication

Searching for new ΦKZ factors with key roles in the phage replication cycle, we observed a set of genes—*ΦKZ120*, *ΦKZ155* and *ΦKZ177*—whose silencing strongly reduced plaque formation but hardly altered phage gene expression up to 35 min p.i. (Supplementary Data 4). This can be rationalized for *ΦKZ120* (major head protein) and *ΦKZ177* (function unknown), as these are late genes (Supplementary Data 4). By contrast, *ΦKZ155* was highly expressed at 20 min p.i. and its expression was dependent on ChmA (Extended Data Fig. 8a). This suggested that the ΦKZ155 protein has an important role after the initial formation of the phage nucleus.

Although *ΦKZ155* is part of the core genome of nucleus-forming bacteriophages[38], its role in the infection cycle of ΦKZ or related phages was unknown. Its predicted structure shows an N-terminal RNase HI domain (Pfam domain PF00075) and a well-structured C-terminal domain[19] (local distance difference test >0.9, AlphaFold3 server, Fig. 4b). To understand the cellular consequences of ΦKZ155 depletion, we performed fluorescence microscopy imaging and found that ASO-mediated knockdown of ΦKZ155 prevented phage nucleus formation (Fig. 4c). Targeting the putative homologue Gp176 in *P. aeruginosa* phage ΦPA3 with a different ASO (Extended Data Fig. 8b) also caused loss of a visible phage nucleus (Fig. 4c). Notably, ΦKZ155 and Gp176 show only 65% sequence identity, but are structurally highly similar (Fig. 4b), and are thus likely to serve the same function.

To test whether ASO-mediated knockdown of an essential gene such as *ΦKZ155* can be reversed through complementation with an insensitive allele, we expressed the open reading frame (ORF) of *ΦKZ155* from a plasmid with a heterologous promoter and a 5′ untranslated region with no binding site for the anti-*ΦKZ155* ASO. This plasmid fully restored nuclear maturation and phage plaque efficiency upon ASO-mediated knockdown of the phage-borne *ΦKZ155* gene (Fig. 4d and Extended Data Fig. 8c). Complementation was also successful with a catalytic centre mutant (ΦKZ155(D102N)) with no in vitro RNase H activity (Extended Data Fig. 8c–e). Thus, ΦKZ155 (whose gene is co-transcribed with *ΦKZ154*, *ΦKZ156* and *ΦKZ156.1*)[49] serves an essential yet nuclease-independent function. Notably, western blot analysis showed that the absence of a visible phage nucleus was not caused by a shortage of ChmA (Fig. 4e). Moreover, our RNA-seq data indicated that phage nucleus-dependent transcription is unabated in the absence of ΦKZ155, which implies successful import of the nvRNAP

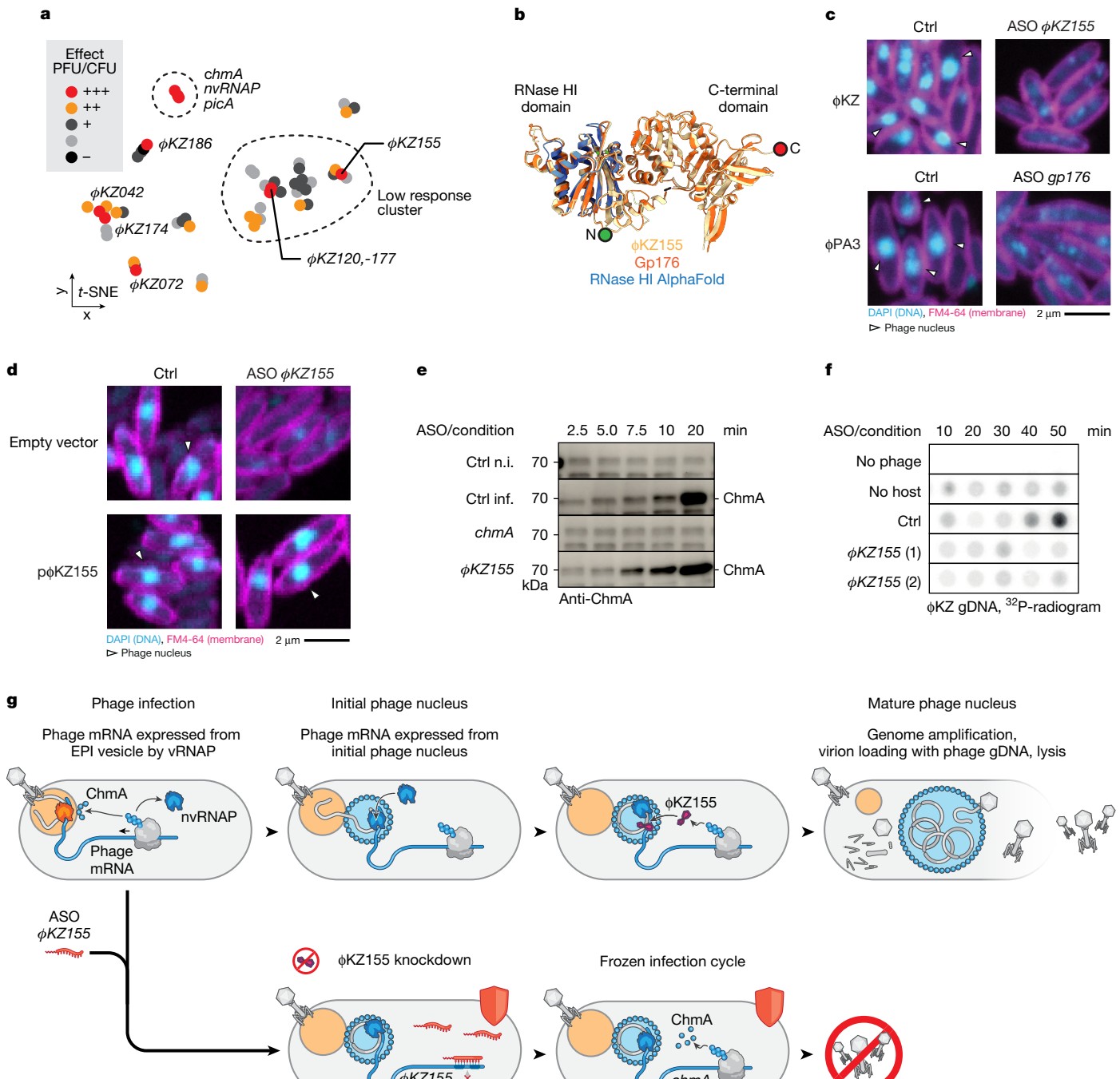

**Fig. 4 | ΦKZ155 is important for phage nucleus maturation and genome amplification. a**, $\log_2$-transformed fold change values for each ΦKZ gene from Supplementary Data 4 at 30 min p.i. were clustered by *t*-SNE. Discussed clusters are highlighted with a dashed line. **b**, AlphaFold3 structure prediction of ΦKZ155 (wheat), the ΦPA3 homologue Gp176 (orange) and an RNase HI domain (blue). N terminus, green; C terminus, red. **c**, PAO1 cells were pretreated with 8 μM ASO targeting *ΦKZ155* (ΦKZ) or *gp176* (ΦPA3) for 30 min. Cells were infected with ΦKZ or ΦPA3 at an MOI of 5 and incubated for 35 or 50 min, respectively, followed by chemical crosslinking and staining of DNA (DAPI) or membranes (FM4-64). **d**, PAO1 cells were transformed with a plasmid encoding *ΦKZ155* or the empty vector. Cells were pretreated with 8 μM ASO targeting *ΦKZ155* for 30 min, infected with ΦKZ at an MOI of 10, plasmid-encoded ΦKZ155 was induced with 0.2% arabinose, and the cells were incubated for 30 min, followed by chemical crosslinking and staining as in **c**. Results in **c**,**d** are representative of two independent experiments. **e**, PAO1 cells were pretreated

with 6 μM ASO targeting *chmA* or *KZ155* for 30 min. Cells were infected with ΦKZ at an MOI of 5 and incubated for the indicated times followed by immunoblotting. inf., infected; n.i., non-infected. A representative result of three independent experiments is shown. **f**, PAO1 cells were pretreated with 6 μM ASO targeting *ΦKZ155* for 30 min. Cells were infected with ΦKZ at an MOI of 5 and incubated for the indicated times followed by DNA extraction and dot blotting. Phage genomic DNA (gDNA) was detected with a radiolabelled probe (JVO-23213). A representative result of two independent experiments is shown. **g**, Model of ΦKZ155 function. After phage infection and initial nucleus formation, the ΦKZ155 protein is imported into the nucleus and has a role in phage nucleus maturation and genome amplification. Upon knockdown of *ΦKZ155*, the phage nucleus remains immature, although ChmA is expressed. The phage genome is not amplified, and phage infection is halted. Uncropped images of blots are available in Supplementary Fig. 1.

into the nascent phage nucleus (Supplementary Data 4). Thus, we propose that the knockdown of *ΦKZ155* arrests the ΦKZ cycle at a checkpoint immediately following initial phage nucleus formation. This is consistent with the ΦKZ155 protein localizing inside the phage nucleus[19] (Extended Data Fig. 8f), where it may drive phage nucleus maturation.

To better understand why loss of ΦKZ155 arrests the infection cycle, we wanted to explore whether phage nucleus size might be linked to genome copy number. Therefore, we quantified phage DNA by dot blot analysis. In this assay, phage DNA became detectable at 40 min p.i. in *P. aeruginosa* PAO1 pretreated with the control ASO. By contrast, ASO-mediated silencing of *ΦKZ155* inhibited phage genome amplification (Fig. 4f), as did silencing of the homologous *gp176* gene in phage ΦPA3 (Extended Data Fig. 8g). Of note, we generally observed no degradation of the host genome (Extended Data Fig. 8h), in agreement with a previous report[48]. Thus, in the absence of ΦKZ155, the phage infection cycle freezes at a crucial stage when the nascent phage nucleus matures by amplification of the phage genome (Fig. 4g). Such defined arrested states achieved by ASO-based depletion of phage factors will help to dissect the molecular underpinning of key decision points in the progressing phage infection cycle.

## Discussion

ASO-mediated gene silencing should be of broad use in phage biology. Although our primary goal was to identify key genes of the *P. aeruginosa* phage ΦKZ, we also show that this functional genomics tool is readily adaptable to other DNA and RNA phages and hosts, including bacteria that are genetically intractable. ASOs can be used to study many aspects of phage–host interactions beyond the scoring of gene essentiality, which includes the suppression of a host-encoded phage defence system or the identification of molecular phenotypes by coupling ASO treatment to sensitive readouts such as RNA-seq. Our screen of one-fifth of the ORFs of the model jumbo phage ΦKZ clearly demonstrates that ASO-mediated mRNA silencing of many of these ORFs produce detectable macroscopic or molecular phenotypes. Of note, this screen sampled a single host under one growth condition, and is therefore likely to underestimate the proportion of functional genes.

The ASOs used here were designed to translationally silence mRNAs. Other attractive targets would be small noncoding RNAs, a class of molecules whose roles in phage–host interactions are becoming more apparent[4,54]. Similarly, ASOs could help to illuminate gene functions for the expanding class of minimal RNA replicators such as viroids and viroid-like covalently closed circular RNAs, many of which replicate in environmental bacteria[55,56]. We also foresee applications in phage therapy, for example, in the optimization of production strains or phage cocktails, or in industrial settings, for example, protecting starter cultures from detrimental phage infections. Such applications will benefit from a main distinguishing feature of ASO-based mRNA knockdown, which is that it does not produce a genetically modified organism.

In relation to jumbo phage biology, ΦKZ155 emerges as a conserved phage-encoded RNase H-like protein with an essential function in the temporal coordination of phage nucleus maturation and genome amplification. This is consistent with the localization of ΦKZ155 within the phage nucleus, where phage genome amplification takes place. ΦKZ155 has also been used to study PicA-mediated import into the phage nucleus[19]. Using an ectopically expressed GFP-tagged variant, we observed that ΦKZ155 forms a single punctum at the periphery of the nucleus (Extended Data Fig. 8f), similar to the recently reported localization of PicA (also known as Imp1) and supporting the proposed stable interaction between both proteins[18,19]. Future studies should focus on how ΦKZ155 licenses phage genome amplification at this stage of the infection cycle. RNase HI domains, such as in ΦKZ155, typically recognize RNA–DNA heteroduplexes, cleaving the RNA strand[57]. Although the molecular mechanisms underlying phage genome replication are largely unknown, it has been suggested that phage-related RNA–DNA

duplexes are resolved by either phage- or host-encoded RNase H[58]. Although ΦKZ155 is a bona fide RNase H (Extended Data Fig. 8d), our complementation assay shows that this nucleolytic activity does not explain why this protein is essential (Extended Data Fig. 8c,e). Nonetheless, the RNase HI fold is conserved among ΦKZ155 homologues, which strongly suggests a nucleic acid-related function. The N-terminal RNase HI domain of ΦKZ155 could potentially serve a sensory rather than catalytic function in phage genome replication.

Engineering phage mutants is not trivial, and is impossible for essential genes, but our ASO method combined with plasmid complementation offers a new experimental route to dissect the role of key residues and the overall domain structure of ΦKZ155. Ultimately, this will help to illuminate the transient state shortly after the phage genome is translocated from the EPI vesicle into the phage nucleus. In addition, it is notable that ΦKZ encodes at least 10 nucleases (2.5% of all annotated genes). Their targets and catalytic mechanisms remain largely unknown, except for those of ΦKZ179, whose homologue in phage ΦPA3 acts in inter-phage competition[59]. Our ASO screen indicates that the predicted HNH nuclease ΦKZ072 has a strong phenotype similar to those of ChmA or ΦKZ155 (Table 1). ΦKZ072 is expressed early and in a phage nucleus-independent manner (Supplementary Data 3), suggesting that it has a role in early host takeover. Given our ability to inhibit their expression, it will be interesting to unravel how this nuclease and others with milder phenotypes (ΦKZ056, ΦKZ165 and ΦKZ199; Table 1) act at different stages in the finely orchestrated phage replication cycle.

Despite its anticipated broad applicability, ASO-based gene silencing also has limitations. First, it relies on efficient delivery of the ASO into the bacterial cytosol, with CPPs such as (KFF)₃K and (RXR)₄XB being commonly used carriers. However, CPP efficacy is influenced by bacterial envelope composition[60] and may be limited by toxicity at higher doses. Alternative carriers, including siderophores and nanoparticles, are currently being developed[25]. Second, ASOs are designed to block translation but they may also trigger mRNA degradation[30,31]. Although this appeared to be rare among the ΦKZ mRNAs targeted here, induced RNA decay must be kept in mind when targeting polycistronic mRNAs in other species. Third, effective ASO design should take into account challenges from incomplete genome annotations, such as misannotated ORFs and the presence of introns in phage genes. Finally, since ASOs typically cause knockdown rather than complete knockout and cannot target virion-packaged proteins, such as vRNAP, during the initial infection, the phenotypes may be mild. Nonetheless, with careful consideration of these factors when selecting the phage–host system, ASOs will be a powerful tool.

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

## Methods

### ASO design

ASOs were designed to bind at the RBS using the MASON algorithm (https://mason.helmholtz-hiri.de)[34] and the NCBI sequence and annotation files (ΦKZ: NC_004629.1, ΦPA3: NC_028999.1, PP7: NC_001628.1, SPO1: NC_011421.1, RAY: NC_041973.1, PA14: NC_008463.1, PAO1: NC_002516.2). ASO length was set to 11 nt and the allowed mismatches for off-targets were set to 4 nt. ASOs were selected based on the following scoring values: melting temperature (45–55 °C), low purine percentage (25–35%) and few predicted off-targets in distinct translation initiation regions of the phage (<3). The scoring values for the used ASOs are given in Supplementary Data 5. At least two ASOs were designed for each targeted gene. The control ASO sequence was GACATA ATTGT (ctrl., JVPNA-79). ASOs were commercially ordered at Peps 4LS (Heidelberg) with a peptide backbone (PNA) and an N-terminal RXR (RXRRXRRXRRXRXB), KFF (KFFKFFKFFK) or TAT (GRKKRRQRRRYK) CPP. The initial concentration was adjusted to 1 mM in water based on the specific extinction coefficient using absorption. ASOs were stored at −20 °C. Prior to use, ASOs were thawed at room temperature, heated for 5 min at 55 °C and then cooled down to room temperature. All ASO sequences are listed in Supplementary Data 5, for example, JVPNA-72 for *chmA* knockdown, JVPNA-125 for *nvRNAP* knockdown, JVPNA-964 for *PicA* knockdown, and JVPNA-960 for *PhuZ* knockdown.

### CFU and PFU assay

*P. aeruginosa* strains PAO1 (JVS-11761, DSMZ: DSM22644), PaLo8, PaLo9, PaLo39, PaLo44 (R. Lavigne laboratory, KU Leuven, Belgium), and PA14 (R. Lavigne laboratory, KU Leuven, Belgium) were grown in LB medium overnight at 37 °C and 220 rpm. Cells were inoculated 1:100 and grown in Mueller–Hinton medium at 37 °C and 220 rpm to an absorbance (optical density) of 0.3 (absorption given at 600 nm). ASOs were added at 6 μM final concentration to 50 μl cultures and incubated for 30 min. Cells were infected with ΦKZ at an MOI = 0.0001 and the cells were incubated for 180 min, which corresponds to three full replication rounds. Five microlitres of cell culture were diluted in series and spotted on LB plates and on 0.5% LB soft agar plates with the susceptible strain PAO1 (one volume of 0.5% LB soft agar at 42 °C, was mixed with 0.01 volume cells at an optical density of 0.5, and poured into a plate). Plates were imaged with the Typhoon 7000 phosphor imager (GE Healthcare) in fluorescence mode.

For *jukA* silencing in PA14, the cells were inoculated 1:100 in Mueller–Hinton medium, after 30 min cells were pretreated with 6 μM ASOs. The cells were infected after 150 min pre-incubation time with ΦKZ at an MOI = 0.0001, and incubated further for 180 min followed by CFU and PFU quantification.

For PP7 infection, PAO1 cells were inoculated 1:100 in Mueller–Hinton medium and treated at optical density 0.3 with 0.05 μM final concentration ASOs, additionally every 30 min 0.05 μM final concentration ASOs were added. Cells were infected 30 min after starting treatment with PP7 at an MOI = 0.00001, and cells were collected for CFU and PFU analysis after 120 min.

For RAY phage, *P. agglomerans* (DSMZ: DSM3493) cells were grown overnight in LB and were then inoculated 1:100 in Mueller–Hinton medium and grown at 37 °C, at optical density 0.3, cells were pretreated with 6 μM RXR–ASOs for 30 min. Cells were infected with phage at an MOI = 0.0001, followed by incubation for 300 min and spotting.

For SPO1 phage, *B. subtilis* 168 (BGSC: 1A1) was grown overnight in LB medium and then inoculated 1:100 in Mueller–Hinton medium, and grown at 37 °C, at optical density 0.3, cells were pretreated with 6 μM KFF–ASOs for 30 min. Cells were infected with phage at an MOI = 0.0001, followed by incubation for 180 min and spotting.

### Immunoblotting

Infected cell cultures were mixed with final 1× SDS–PAGE loading dye (60 mM Tris/HCl pH 6.8, 0.2 g ml⁻¹ SDS, 0.1 mg ml⁻¹ bromophenol blue, 77 mg ml⁻¹ DTT, 10% (v/v) glycerol) and were boiled for 10 min at 95 °C for denaturation. Protein samples were analysed by SDS–PAGE and blotted onto methanol-preactivated polyvinylidene (PVDF) membranes. As a loading control, we used Coomassie staining of a second gel where we loaded the same sample volume. ChmA was produced as previously described[16] in *E. coli* BL21-CodonPlus (DE3)-RIL cells (Agilent Technologies, JVS-12280, chloramphenicol resistance) that were transformed with pET-M14(+) plasmid carrying the *chmA* gene with a N-terminal His-V5-TEV-tag (pMiG118). ChmA was purified as described before[16]. The tag could not be removed in the purification procedure. Commercial antibody sera were generated by Eurogentec. Rabbits were immunized with the purified ChmA protein. The rabbit serum (no. 2481) was used at a 1:10,000 dilution together with anti-rabbit HRP-conjugated antibody (Thermo Scientific, 31460) in a 1:10,000 dilution in 5% BSA/TBST for ChmA detection in immunoblotting. Antibody specificity was validated in immunoblotting by comparison between ΦKZ-infected and non-infected cells that yielded a defined band at 70 kDa corresponding to ChmA only in infected cells (Fig. 1c).

### Microscopy

Agarose (0.85% (w/v)) was dissolved in fivefold water-diluted LB medium and boiled to melt. The liquid agarose was poured on microscope slides with one slide pair at each side as a spacer and one slide on top to form a closed gel slice as described[61]. After solidification, approximately 1 cm × 1 cm pads were cut. Cells were grown in Mueller–Hinton medium at 220 rpm and 37 °C to an $A_{600 nm} \approx 0.3$, then preincubated with 8 μM ASOs for 30 min, and infected with ΦKZ or ΦPA3 at an MOI = 10. Of note, ΦKZ Δ*phuZ* had a low titre owing to deletion of *phuZ* and could only be infected with an MOI of ~0.001. At indicated time points the phage replication cycle was quenched by cooling the cells on ice for 10 min and the cells were pelleted at 8,000*g* for 5 min. The supernatant was removed, and the cells were resuspended in 500 μl 4% paraformaldehyde and incubated for 15 min on ice. Afterwards cells were washed with PBS and were resuspended in 50 μl PBS for storage at 4 °C. For the imaging of the ΦKZ155 knockdown, we crosslinked bacteria in the medium with 2% glutaraldehyde for 30 min on ice, followed by the addition of 5% formaldehyde for 30 min on ice, as previously described[62]. Cells were stained with 16 μM FM4-64, and 360 nM DAPI and 5 μl were layered onto 1.2% agarose pads. The pad was placed with the side of application downwards into a μ-Slide 8 Well high Grid-500 (BD Biosciences). Transmission and fluorescence were detected with a confocal laser scanning microscope Leica SP5. Images were processed with ImageJ (1.53).

For the imaging of ΦKZ155–GFP, PAO1 cells were transformed with a plasmid (pLBu005) encoding *ΦKZ155-gfp* under the control of an arabinose-inducible pBAD promoter and selected on gentamycin plates. The cells (JVS-13713) were grown to an optical density of 0.25 and induced with arabinose at indicated concentrations followed by phage infection with an MOI = 10 at optical density 0.3. The collection, crosslinking and imaging of cells was conducted as described above for wild-type cells.

### Proteomics

PAO1 cells were grown overnight in LB medium and were then inoculated 1:100 in Mueller–Hinton medium and grown at 37 °C and 220 rpm for 150 min to optical density 0.3. Cells were pretreated for 30 min with 6 μM ASOs against *chmA*, nvRNAP transcript (*ΦKZ055*), and *picA*. Subsequently, cells were infected with ΦKZ at an MOI = 5. At 2.5, 5.0, 7.5 and 10.0 min p.i. cells were collected with addition of 1/3 volume 4× Bolt SDS sample buffer (Invitrogen), and were immediately boiled at 95 °C for 5 min. Mass spectrometry sample preparation and measurement was conducted at the Proteomics Core Facility EMBL Heidelberg as previously described[37]. Proteins were quantified using label-free quantification (LFQ). Enrichment as $\log_{10}$-transformed *P* values was calculated with a two-sided Student's *t*-test in MaxQuant Perseus (2.1.3)[63].

## RNA preparation and RNA-seq

PAO1 was grown overnight in LB at 37 °C 220 rpm. Cells were inoculated 1:100 in Mueller–Hinton medium at 37 °C and grown to an optical density of 0.3. 6 µM ASO (JVPNA-79 for control or JVPNA-72 for *chmA* inhibition) were added to 1.6 ml of cell culture and incubated for 30 min at 37 °C 220 rpm. Cells were infected with ΦKZ at an MOI = 5. At indicated time points, 250 µl were removed and incubated on ice. Infection efficiency was independently validated by confocal microscopy and CFU spotting with 50 µl of cells. RNA was isolated from 200 µl of cells using the RNAsnap procedure[64]. Two volumes of RNAprotect (Qiagen) were added and cells were incubated for 5 min. Cells were pelleted at full speed for 20 min at 4 °C and the supernatant was removed. The pellet was resuspended in 100 µl SNAP buffer (0.025% SDS, 18 mM EDTA, 1% β-mercaptoethanol, 95% formamide (RNA-grade)). Samples were incubated for 7 min at 95 °C, cell debris was pelleted at full speed for 5 min at room temperature, and the supernatant was transferred to a new tube. 1.5 volumes of ethanol were added to the supernatant and the sample was mixed by pipetting. The sample was loaded onto a miRNeasy mini column (Qiagen) two times and spun at full speed for 20 s at room temperature. Columns were washed two times with 500 µl RPE buffer (Qiagen) and spun at 8,000*g* for 1 min at room temperature. One final spin was used to dry the column in an empty tube. Thirty microlitres of RNase-free water was added to the column and the RNA eluted at 8,000*g* for 1 min at room temperature. The elution was repeated with the flow-through to recover more RNA. The RNA concentration was determined by absorption at 260 nm. RNA was stored at −80 °C.

RNA-seq was performed at the CoreUnit SysMed at the University of Würzburg. DNA was digested with DNaseI and the rRNA was depleted with the Lexogen RiboCOP META depletion kit. RNA library was prepared with the CORALL Total RNA-Seq Library Prep Kit V1 (Lexogen). The library was sequenced on a NextSeq2000 (Illumina) machine with a P1-seq kit (single-end 1× 100 bp, Illumina). RNA-seq analysis for the ChmA knockdown and the screen was conducted with READemption 0.4.3 and 2.0.4, respectively[65]. Reads were aligned for PAO1 and ΦKZ to NC_002516 and NC_004629, respectively. Enrichment of transcripts was calculated with the DeSeq2 module in READemption (2.0.4)[65]. Read coverage was illustrated with the Integrated Genomics Viewer (IGV)[66].

We defined early and middle/late phage transcripts based on their significant enrichment ($log_2$fold > 2 or $log_2$fold < −2, $−log_{10}$-adjusted *P* value > 10) between the control 35- and 10-min samples (Extended Data Fig. 6a, as in ref. 44).

## Structure prediction

Structures were predicted from protein sequences using Google AlphaFold3 server (https://alphafoldserver.com/)[67]. This information is subject to AlphaFold Server Output Terms of Use found at https://alphafoldserver.com/output-terms (Google LLC). Ongoing use is subject to AlphaFold Server Output Terms of Use and of any modifications made. For the RNase HI domain structure in Fig. 4b, the predicted reference structure A0A2A2IBB4 was used from the AlphaFold Protein Structure Database (https://alphafold.ebi.ac.uk/).

## Complementation assay

PAO1 cells were transformed with pLBu019_ΦKZ155-TEV-3xFlag and pLBu021_ΦKZ155CDN-TEV-3xFlag and selected with gentamycin. Cells were grown overnight in LB medium with gentamycin. For the assay, no gentamycin was used in the medium. Cells were inoculated 1:100 in Mueller–Hinton medium. Cells were treated with 6 µM ASOs against control or *ΦKZ155* for 30 min. For spotting, cells were infected with ΦKZ at an MOI = 0.0001, the complementation gene was induced with 0.2% arabinose, and cells were spotted at 180 min p.i. For imaging, cells were infected with ΦKZ at an MOI = 10, the complementation gene was induced with 0.2% arabinose, and cells were chemically crosslinked

at 35 min p.i. and stained with FM4-64 and DAPI as described in the microscopy section and imaged.

## In vitro translation

Template DNA was produced via PCR and Taq-polymerase followed by gel purification. Primers JVO-23244 and JVO-23245 were used to amplify wt *ΦKZ155* with a T7 promoter from ΦKZ lysate, and for *ΦKZ155^{D102N}*, pLBu021 was used as template. Template DNA (250 ng) was supplemented in 10 µl PURExpress in vitro protein synthesis kit mix (NEB, E6800). The reaction mix was incubated for 120 min at 37 °C and was subsequently used for assays.

## Cleavage assays

RNA template (JVRNA-001, AUAUAAGGGAACAUAGAUAAACCCCUCC CUAAUAAAAUG) was labelled with 5′-$^{32}$P. For the RNA–DNA duplex, the radiolabelled RNA was mixed with twice the amount of the reverse complement DNA (JVO-23273), boiled and slowly cooled down to room temperature in a water bath to anneal the duplexes. One picomole of RNA or RNA–DNA duplex was added to 2 µl PURExpress IVT mix that was translating for 2 h a control, ΦKZ155 or ΦKZ155(D102N). The mix was incubated for 1 h, mixed with 1 volume GLII buffer, boiled for 5 min, rapidly cooled down on ice, and loaded onto a 6% 6 M Urea–PAGE (19:1) gel that was run at 300 V for 2 h. The gel was transferred onto Whatman filter paper, vacuum dried, and a phosphor image screen was used to read out the autoradiogram on a Typhoon FLA7000 imager (GE Healthcare). As a positive control, we used a commercially available RNase H (NEB, M0297S).

## Southern dot blotting

PAO1 cells were grown to an optical density of 0.3 in Mueller–Hinton medium and were pretreated with 6 µM ASOs against *ΦKZ155* (ΦKZ) or *gp176* (ΦPA3) for 30 min. Cells were infected at an MOI = 5 with ΦKZ. At indicated time points a fraction of the culture was removed and 1% SDS was added followed by boiling at 95 °C for 5 min. Subsequently, the DNA was extracted from the sample with phenol choloroform:isoamyl alcohol, and subsequently with one volume of chloroform. The aqueous phase was supplemented with 1.5 volume 1 M NaOH and 15 mM EDTA (pH 9). The sample was heated for 3 min at 95 °C and put on ice for 5 min. The solution was filtered with a dot blot apparatus through an equilibrated (0.3 M NaOH) and positively charged nylon membrane. Subsequently, the membrane was dried and the DNA crosslinked via exposition to UV light for 5 min. The membrane was equilibrated with hybridization solution two times, and a radiolabelled oligo was added for hybridization overnight starting at 60 °C for 1 h and then 48 °C overnight. The membrane was washed once for 15 min with 2× SSC and with 0.5× SSC, and a screen was used for phosphor imaging on a Typhoon FLA7000 imager (GE Healthcare).

## Reporting summary

Further information on research design is available in the Nature Portfolio Reporting Summary linked to this article.

## Data availability

The genome annotation files used for ASO design are available at the National Center for Biotechnology Information (NCBI) (https://www.ncbi.nlm.nih.gov/) under accession numbers NC_004629.1 (ΦKZ), NC_028999.1 (ΦPA3), NC_001628.1 (PP7), NC_011421.1 (SPO1), NC_002516.2 (PAO1), NC_008463.1 (PA14) and NC_041973.1 (RAY). Raw sequencing data and coverage files have been deposited at Gene Expression Omnibus[68] for ChmA knockdown and the screen with accession numbers GSE269401 and GSE269911, respectively. Proteomics data have been deposited at PRIDE[69] with the identifier PXD062538. Processed data are provided in Supplementary Data 1–4. Materials are listed in Supplementary Data 5. Source data for Fig. 3, Extended Data

Figs. 2a, 3a,b and 6a are provided with this paper. Uncropped images of blots are available in Supplementary Fig. 1. Other source data are deposited at Zenodo (https://doi.org/10.5281/zenodo.16357062 (ref. 70)).

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

**Acknowledgements** The authors thank L. Vogel and B. Plaschke for technical assistance; E. Hauschild for microscopy; A. Sparmann for discussions and for editing the manuscript; the EMBL Proteomics Core Facility (Heidelberg) and J. Schwarz for performing the proteomics experiment; J. Pogliano for providing the RAY phage; J. Bondy-Denomy for providing the ΦPA3 and ΦKZ Δ*phuZ* phages; Y.-H. Cho for providing the PP7 phage; and R. Lavigne for providing the *P. aeruginosa* strains PA14 and PaLo strains. The work was funded by Deutsche Forschungsgemeinschaft (DFG) project 465133664 in the Priority Programme 'New Concepts in Prokaryotic Virus-host Interactions—From Single Cells to Microbial Communities' (SPP 2330) and by a Gottfried Wilhelm Leibniz award (DFG Vo875/18), both awarded to J.V. M.G. acknowledges funding from the Federal Ministry of Research, Technology and Space under the Microbial Stargazing project (01KX2324). J.V. is a member of CNATM (Cluster for Nucleic Acid Therapeutics Munich; Cluster4Future programme project ID 03ZU1201CA) and the DFG Cluster of Excellence EXC 3113/1: Cluster for Nucleic Acid Sciences and Technologies (NUCLEATE).

**Author contributions** M.G. and J.V. conceived the project and designed experiments. M.G. and L.B. executed the ASO experiments and screens, and optimized the procedures for targeting of other species. S.Đ.-M. established initial experiments and designed ASOs. L.B. and Y.Z. investigated the role of ΦKZ155. L.B. and S.C. performed microscopy experiments. V.R. performed PP7 experiments. L.P. performed experiments during the revision process. M.G., L.B. and T.G. performed RNA-seq experiments. J.V. and M.G. wrote the manuscript and all authors contributed to editing. J.V. procured funding.

**Funding** Open access funding provided by Helmholtz-Zentrum für Infektionsforschung GmbH (HZI).

**Competing interests** The HZI filed a patent application (EP24191874.7) on which J.V., M.G., and S.Đ.-M. are inventors, covering the method for targeting and mapping of essential phage genes. The other authors declare no competing interests.

**Additional information**
**Correspondence and requests for materials** should be addressed to Jörg Vogel.

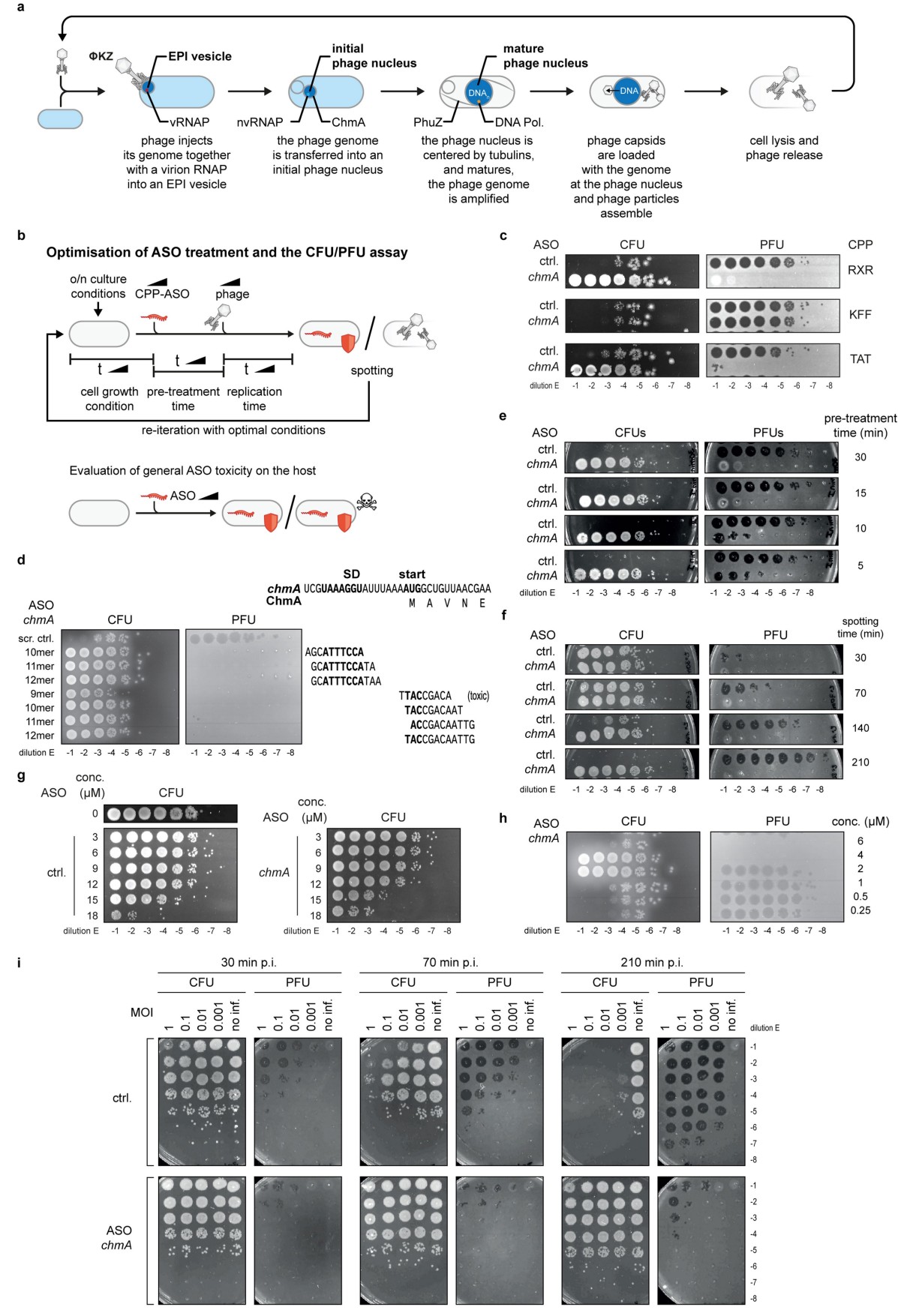

**Extended Data Fig. 1** | See next page for caption.

**Extended Data Fig. 1 | Optimization of ASO treatment. a**. Replication cycle of ΦKZ. **b**. Scheme depicting the optimization for the CFU/PFU assay (top) and assessment of ASO toxicity (bottom). **c**. PAO1 cells were pretreated with 6 μM ASO against *chmA* linked to the carrier peptides (RXR)$_4$XB, (KFF)$_3$K or TAT for 30 min. Cells were infected with ΦKZ at an MOI = 0.0001 and incubated for 180 min followed by CFU/PFU determination. Representative result of two independent experiments. **d**. PAO1 cells were pretreated with 6 μM ASO with indicated lengths and target sequences against RBS and AUG of *chmA* for 30 min. Cells were infected with ΦKZ at an MOI = 0.0001 and incubated for 180 min followed by CFU/PFU determination. SD, Shine-Dalgarno sequence. Representative result of three independent experiments. **e**. PAO1 cells were pretreated with 6 μM ASO against *chmA* for 30, 15, 10 or 5 min. Cells were infected with ΦKZ at an MOI = 0.0001 and incubated for 180 min followed by CFU/PFU determination. **f**. PAO1 cells were pretreated with 6 μM ASO against *chmA* for 30 min. Cells were infected with ΦKZ at an MOI = 0.0001 and incubated for 30, 70, 140, and 210 min followed by CFU/PFU determination. **g**. PAO1 cells were pretreated with 3 to 18 μM ASO against *chmA* for 30 min. Cells were incubated for 180 min followed by CFU/PFU determination. **h**. PAO1 cells were pretreated with 0.25 to 6 μM ASO against *chmA* for 30 min. Cells were infected with ΦKZ at an MOI = 5 and incubated for 180 min followed by CFU/PFU determination. **i**. PAO1 cells were pretreated with 6 μM ASO against *chmA* for 30 min. Cells were infected with ΦKZ at an MOI between 0.001 and 1 and incubated for 30, 70, 210 min followed by CFU/PFU determination.

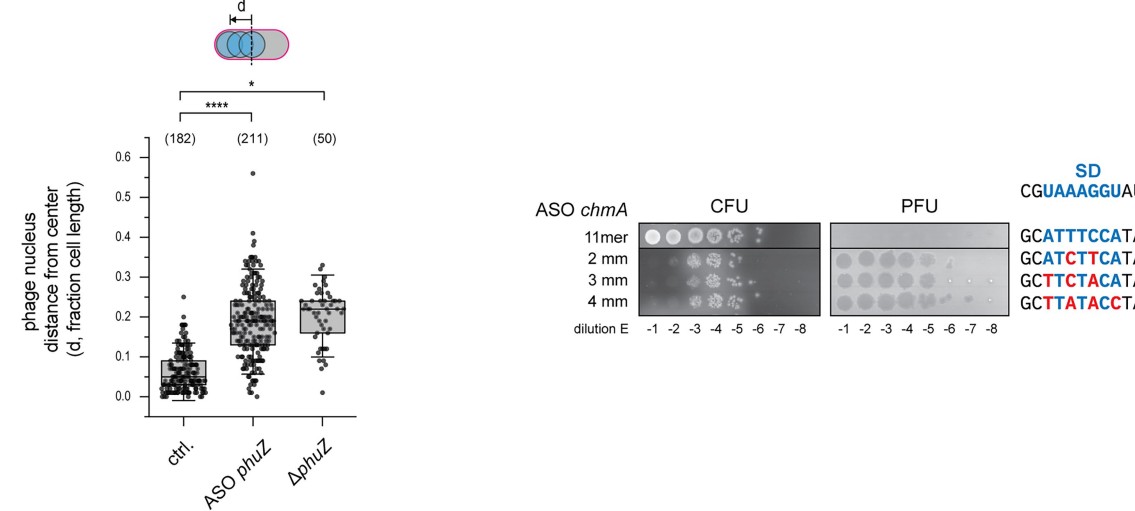

**a**

d

phage nucleus distance from center (d, fraction cell length)

(182)   (211)   (50)

****
*

0.6
0.5
0.4
0.3
0.2
0.1
0.0

ctrl.   ASO *phuZ*   Δ*phuZ*

**b**

ASO *chmA*        CFU                    PFU

**SD**
CG**UAAAGGU**AU

11mer                                    GC**ATTTCCA**TA
2 mm                                     GC**AT**C**TT**CA TA
3 mm                                     GC**TT**C**TA**CATA
4 mm                                     GC**TTATACC**TA

dilution E   -1 -2 -3 -4 -5 -6 -7 -8   -1 -2 -3 -4 -5 -6 -7 -8

**c**

CFU        PFU        IB ChmA

*chmA*                    **SD**        **start**
AGUACUUAGGUUCUUCUUGAUCAAGAAACCAUAAUCG**UAAAGGU**AUUUAAA**AUG**GCUGUUAACGAAAACGAAAUCGGCACCGUUCAAACUCAAGCUAC

|  | %_SC_bases | Tm (°C) | pur_perc | OT_TIR_0mm | OT_TIR_1mm |
|---|---|---|---|---|---|
| control |  |  |  |  |  |
| TCATGAATCCA |  |  |  |  |  |
| AATCCAAGAAG |  |  |  |  |  |
| AAGAAGAACTA |  |  |  |  |  |
| GAACTAGTTCT |  |  |  |  |  |
| AGTTCTTTGGT |  |  |  |  |  |
| TCTTTGGTATT |  |  |  |  |  |
| TTGGTATTAGC |  |  |  |  |  |
| GTATTAGC**ATT** |  |  |  |  |  |
| TTAGC**ATTTCC** |  |  |  |  |  |
| TAGC**ATTTCCA** |  |  |  |  |  |
| AGC**ATTTCCA**T |  |  |  |  |  |
| > GC**ATTTCCA**TA  < primary ASO in study |  |  |  |  |  |
| C**ATTTCCA**TAA |  |  |  |  |  |
| **ATTTCCA**TAAA |  |  |  |  |  |
| **TTTCCA**TAAAT |  |  |  |  |  |
| **TTCCA**TAAATT |  |  |  |  |  |
| **TCCA**TAAATTT |  |  |  |  |  |
| **CCA**TAAATTT**T** |  |  |  |  |  |
| **CA**TAAATTT**TA** |  |  |  |  |  |
| **A**TAAATTT**TAC** |  |  |  |  |  |
| TAAATTT**TACC** |  |  |  |  |  |
| AAATTT**TACC**G |  |  |  |  |  |
| AATTT**TACC**GA |  |  |  |  |  |
| ATTT**TACC**GAC |  |  |  |  |  |
| TTT**TACC**GACA |  |  |  |  |  |
| TT**TACC**GACAA |  |  |  |  |  |
| T**TACC**GACAAT |  |  |  |  |  |
| **TACC**GACAATT |  |  |  |  |  |
| **ACC**GACAATTG |  |  |  |  |  |
| **CC**GACAATTGC |  |  |  |  |  |
| CGACAATTGCT |  |  |  |  |  |
| GACAATTGCTT |  |  |  |  |  |
| AATTGCTTTTG |  |  |  |  |  |
| TGCTTTTGCTT |  |  |  |  |  |
| TTTTGCTTTAG |  |  |  |  |  |
| TGCTTTAGCCG |  |  |  |  |  |
| TAGCCGTGGCA |  |  |  |  |  |
| GTGGCAAGTTT |  |  |  |  |  |
| AAGTTTGAGTT |  |  |  |  |  |
| TGAGTTCGATG |  |  |  |  |  |

-1 -2 -3 -4 -5 -6 -7 -8   -1 -2 -3 -4 -5 -6 -7 -8
dilution E

-70 kDa ChmA

min.   18   23   18   0   0
max.   91   67   82   1   3

**Extended Data Fig. 2** | See next page for caption.

**Extended Data Fig. 2 | Quantification of *phuZ* ASO treatment and gene deletion and ASO specificity assessment by mismatches and tiling. a**. PAO1 cells were pretreated with 8 μM ASO (JVPNA-960) against the phage spindle apparatus gene *phuZ* or a control ASO (JVPNA-79) for 30 min, infected with ΦKZ at an MOI = 10, followed by chemical crosslinking and staining of membranes (FM4-64) and DNA (DAPI) at either 5 min or 30 min p.i. In parallel, PAO1 cells were infected with a *phuZ* deletion phage at MOI = 0.001. The distance of the phage nucleus from the cell centre (d) was measured. Quantitation is based on one representative example of four independent experiments for the ASO-knockdown and on one representative example of two independent experiments for the Δ*phuZ* phage[29]. The number of counted cells is indicated in brackets. Box plot indicate median, first and third quartiles (box), whiskers indicate ±1.5 SD, * p < 0.05, **** p < 0.0001, two-tailed Mann-Whitney test. **b**. Mismatches (mm, red) were introduced to the ASO targeting *chmA*. PAO1 cells were pretreated with 6 μM of the indicated ASOs for 30 min. Cells were infected with ΦKZ at an MOI = 0.0001 and incubated for 180 min followed by CFU/PFU determination. Representative result of two independent experiments. **c**. ASOs were tiled along the RBS and AUG window of the *chmA* transcript. PAO1 cells were pretreated with 6 μM indicated ASOs against *chmA* for 30 min. Cells were infected with ΦKZ at an MOI = 0.0001 and incubated for 180 min followed by CFU/PFU determination. For the quantification of ChmA, PAO1 cells were pretreated with 6 μM indicated ASOs against *chmA* for 30 min. Cells were infected with ΦKZ at an MOI = 5 and incubated for 30 min followed by quantification of ChmA via immunoblotting. Representative result of two independent experiments. ASO self-complementarity (SC), melting temperature (Tm), purine percentage (pur_perc), and off-targets (OT) with 0 and 1 miss-match (mm) in the phage and host are indicated as heatmaps. ASO parameters were determined with the ASO checker tool of MASON (mason.helmholtz-hiri.de/ASO_checker). ASOs directed against the A/U-rich region between the SD and AUG fail to repress ChmA protein synthesis, likely due to their low melting temperature. Source data for panel **(a)** are available online.

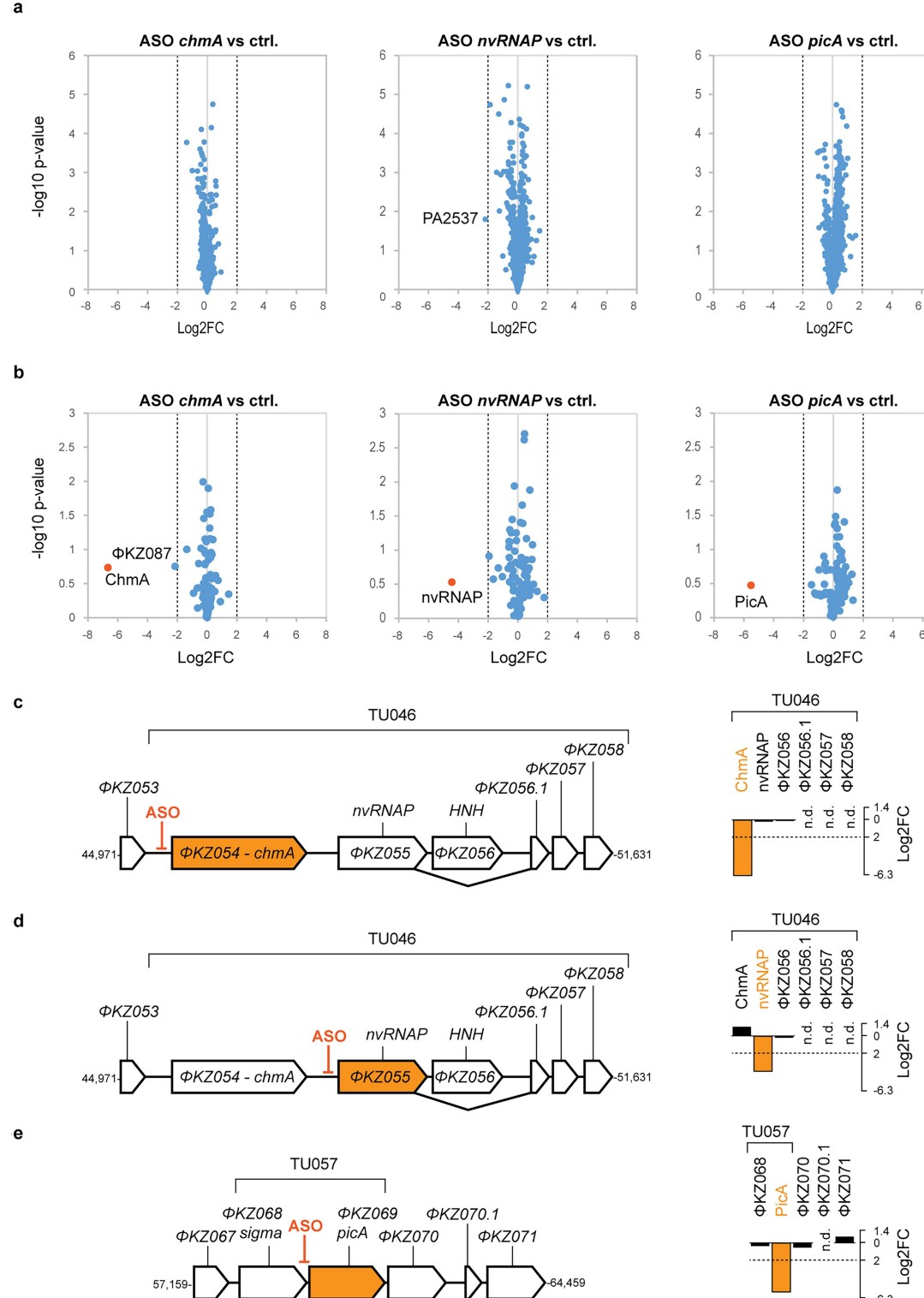

**Extended Data Fig. 3** | See next page for caption.

**Extended Data Fig. 3 | ASO knockdown specifically downregulates targets.** PAO1 cells were pretreated with 6 μM ASO against *chmA*, *ΦKZ055* (nvRNAP), and *picA* for 30 min. Cells were infected with ΦKZ at an MOI = 5 and incubated for 2.5, 5.0, 7.5, and 10 min followed by harvesting in non-reducing SDS-PAGE loading dye. Subsequently, proteomics was performed on the samples and proteins were quantified by label-free quantification (LFQ). **a.** Counts for host proteins at all timepoints were pooled to calculate enrichment and log10 p-value (calculated with MaxQuant Perseus, a two-sided Student's t-test was used with no adjustments for multiple comparisons). Filter criteria were applied (only one protein in protein group, >=4 unique peptides, >40 peptide posterior error probabilities score, sum of average LFQ intensity was filtered at >1E8 counts), n = 1,130 host proteins were considered. Host protein levels were not altered in the range log2 fold change (log2FC) <-2 or >2 and a −log10 p-value > 2. **b.** Counts for phage proteins at 7.5 and 10 min p.i. timepoints were pooled to calculate enrichment and log10 p-value (calculated with MaxQuant Perseus, a two-sided Student's t-test was used with no adjustments for multiple comparisons). Filter criteria were applied (sum of log2LFQ counts for all three ASO treatments >25 counts); n = 95 phage proteins were considered. Log2FC was calculated based on average counts. When a protein was not detected, we set a pseudo-count=1. ΦKZ016 and ΦKZ165 were lower in the non-targeting ASO control sample and therefore omitted. The structural protein ΦKZ094 was only detected in the sample treated with an ASO targeting *chmA* at 7.5 min p.i. but not at 10 min or in the control sample and was also omitted. **c-e.** Left, Schematic overview of the transcriptional units (TU) of the targeted transcripts, based on Putzeys et al. 2024 (ref. 49). The position of ASO is indicated. Right, Log2FC of protein levels at 10 min p.i. based on LFQ counts. Source data for panel **(a)** and **(b)** are available online.

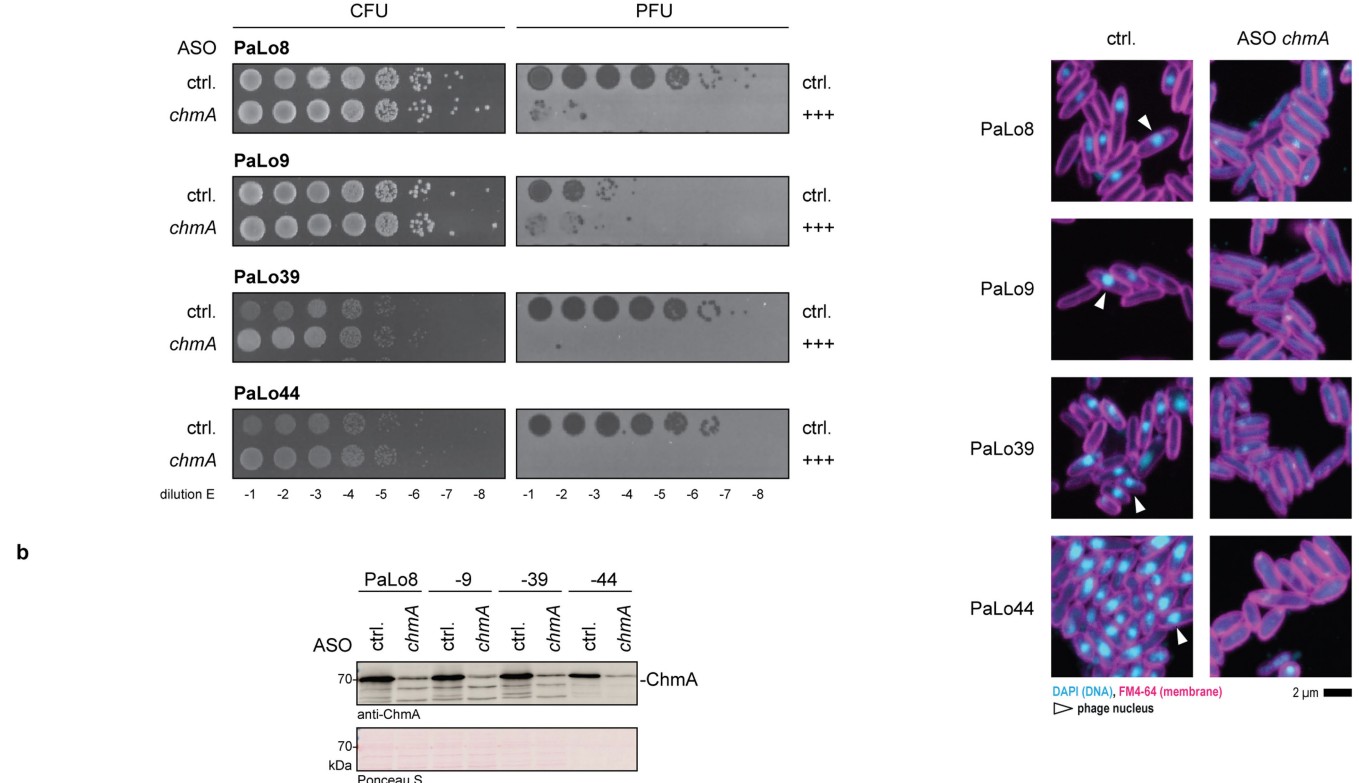

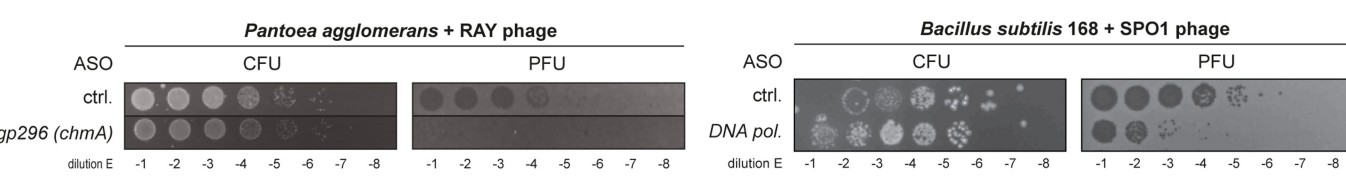

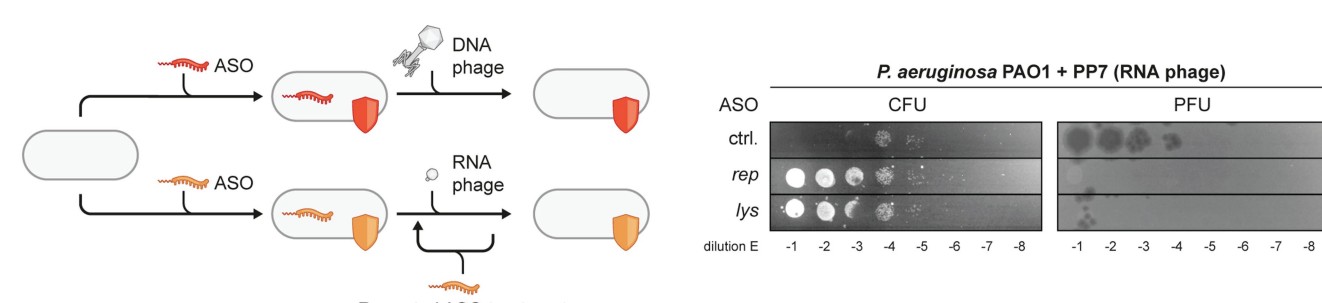

**Extended Data Fig. 4 | ASOs can target diverse clinical *Pseudomonas* isolates and other species of phages and phage-infected cells. a**. PaLo8/9/39/44 cells were pretreated with 6 µM ASO against *chmA*. Cells were infected with ΦKZ at an MOI = 0.0001 and incubated for 180 min followed by CFU/PFU determination. **b**. PaLo8/9/39/44 cells were pretreated with 6 µM ASO against *chmA*. Cells were infected with ΦKZ at an MOI = 5 and incubated for 20 min followed by quantification of ChmA levels via immunoblotting with an antibody against ChmA. **c**. PaLo8/9/39/44 cells were pretreated with 8 µM ASO against *chmA*. Cells were infected with ΦKZ at an MOI = 10 and incubated for 40 min followed by chemical crosslinking and staining with DAPI to visualise DNA and FM4-64 to visualize membranes. **d**. *Pantoea agglomerans* cells were pretreated with 6 µM ASO against *gp296* (*chmA*) for 30 min. Cells were infected with RAY at an

MOI = 0.0001 and incubated for 300 min followed by CFU/PFU determination. **e**. *B. subtilis* 168 cells were pretreated with 6 µM ASO against the transcript of gene *31* (DNA polymerase) for 30 min. Cells were infected with SPO1 at an MOI = 0.0001 and incubated for 180 min followed by CFU/PFU determination. **f**. PAO1 cells were pretreated with 0.05 µM ASO against replication (*rep*) and lysis (*lys*) transcripts for 30 min. Cells were infected with ΦKZ at an MOI = 0.00001. 0.05 µM ASO was added again with the infection and every 30 min thereafter. Cells were incubated for 180 min after infection followed by CFU/PFU determination. For panels **(a)-(f)**, a representative result of two independent experiments is shown. Uncropped images of blots are available in Supplementary Fig. 1.

**a**

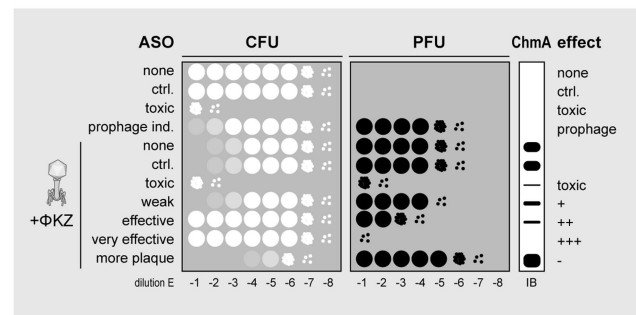

**b**

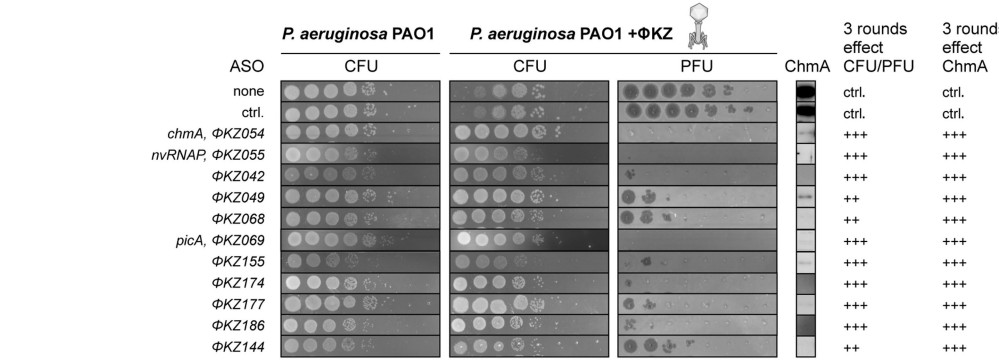

**c**

**Extended Data Fig. 5 | Screen scoring scheme, top hits and other effects caused by ASOs. a**. Schematic representation of possible effects of ASO treatment on CFU/PFU readouts and ChmA levels after multiple replication rounds. ASOs can be toxic to the host causing reduced CFU counts even without phage infection. ASOs can also induce a prophage, visible as PFUs. ASOs that showed one log reduction in PFUs and/or reduced ChmA levels were scored as weak (+). ASO that reduced PFUs by multiple logs, rescued CFUs in the first dilution, and/or depleted ChmA levels were scored as effective (++). ASOs that diminished PFUs down to the first dilution or completely abrogated PFUs, and/or depleted ChmA levels by more than 10-fold were scored as very effective (+++). Some ASOs caused increased plaque counts (−) and/or increased

ChmA levels. **b**. CFU/PFU and ChmA levels upon ASO-mediated knockdown of top candidates that showed a strong effect on phage plaques in our screen. PAO1 cells were pretreated with 6 μM ASO against the indicated transcripts. Cells were infected with ΦKZ at an MOI = 0.0001 and incubated for 180 min followed by CFU/PFU determination. Scoring as described in **(a)**. **c**. Examples of pleiotropic effects of ASO treatment. Experiment as in **(b)**. ASOs that were toxic to the host are denoted by their internal reference number because this effect is likely unspecific and not related to the intended phage target gene. ASO treatment can result in more phage plaques (−) and ASOs can induce prophages (prophage ind.).

**a**

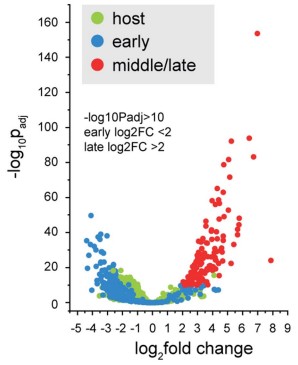

-log10Padj>10
early log2FC <2
late log2FC >2

**b**

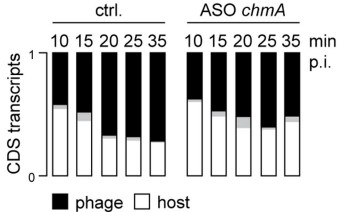

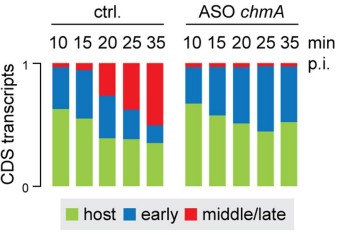

**c**

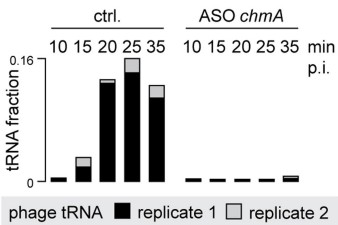

**d**

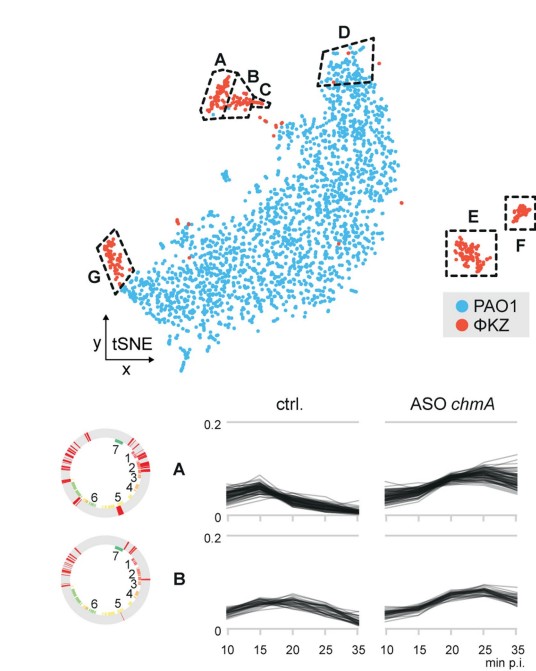

sum of reads in gene quantification > 400 (n=2,556), CDS only,
transcript abundance from all time points and conditions was normalised to the sum=1,
and used in tSNE with perplexity 40, exaggeration 1, PCA components 10

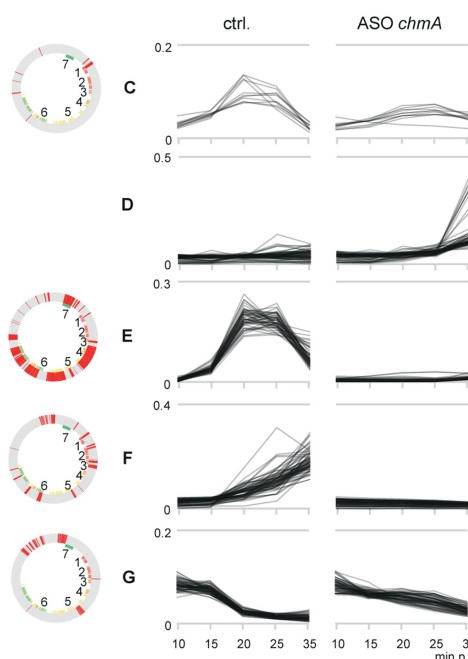

**Extended Data Fig. 6 | Early-middle/late gene classification and effects upon inhibition of ChmA synthesis. a.** Log2FC in transcript levels based on RPKM counts 35 vs. 10 min post infection in the samples treated with a non-targeting control ASO. Phage genes were classified as early (blue) upon depletion, and as middle/late (red) upon enrichment in the Log2FC plot 35 vs. 10 min. Other phage transcripts remained unaltered in the comparison of these two time points and were likely transcribed by the vRNAP. Two independent replicates were merged by geometrical averaging and the p-values were calculated by the Wald test (two-sided) without adjustments for multiple comparisons using DESeq2. **b.** PAO1 cells were pretreated with 6 µM ASO against *chmA*. Cells were infected with ΦKZ at an MOI = 5 and incubated for the indicated times followed by RNA extraction and sequencing of the transcriptomes. Relative quantification of protein-coding transcript (CDS) reads to total reads (RPKM) is shown. **c.** Relative read counts to total counts for phage tRNAs in the tRNA fraction in the samples treated with the non-targeting control ASO and the ASO targeting *chmA*; results are depicted as an overlay of two independent experiments. **d.** Transcript levels in samples treated with a non-targeting control ASO or an ASO targeting *chmA* were normalized over the course of infection and clustered by *t*-SNE. Individual clusters are represented together with genomic locations, core genes and blocks. Source data for panel **(a)** are available online.

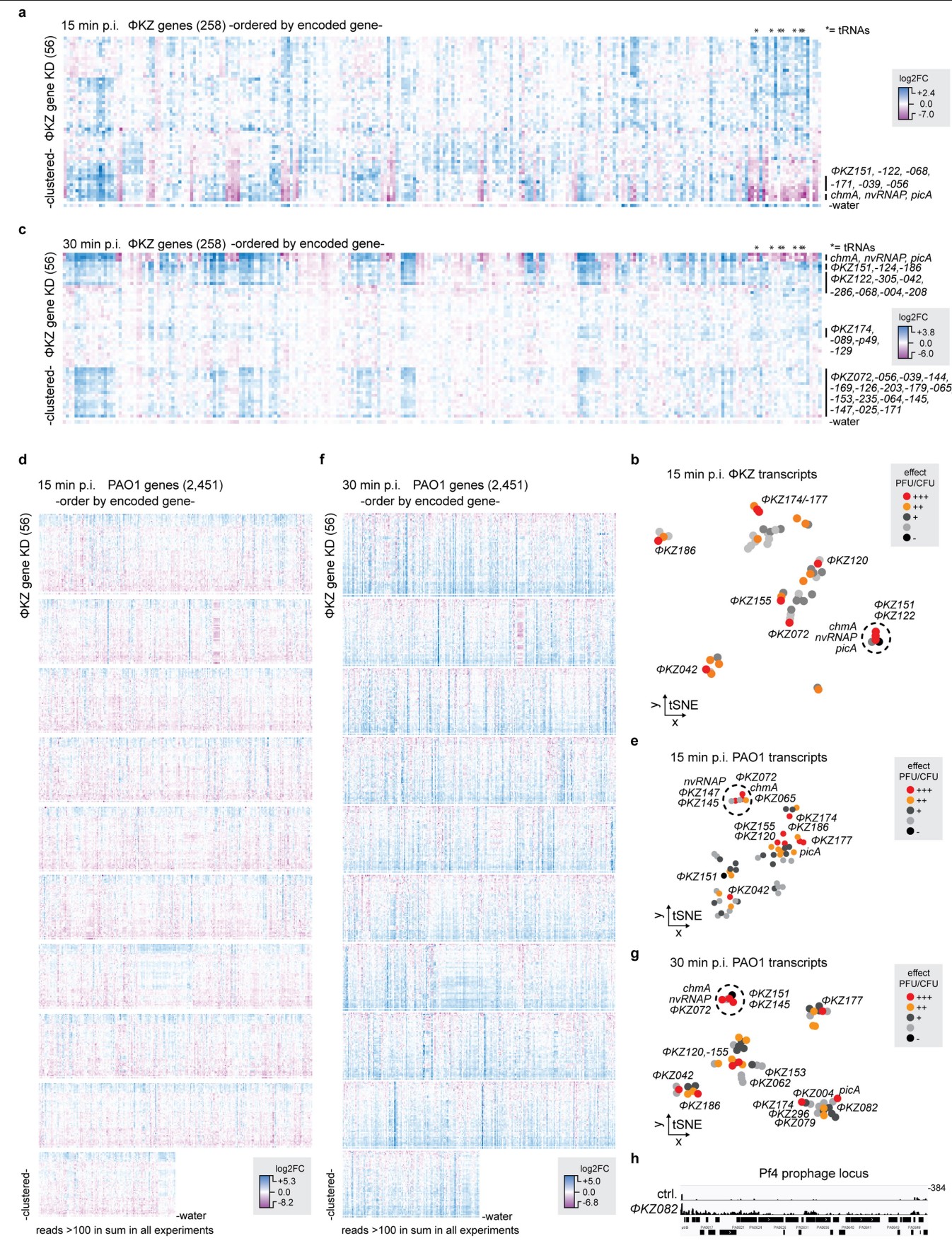

**Extended Data Fig. 7** | See next page for caption.

**Extended Data Fig. 7 | ASO-mediated silencing of top hits has effects on host and phage transcript levels.** PAO1 cells were pretreated with 6 μM ASO against the 56 top hits for 30 min. Cells were infected with ΦKZ at an MOI = 5 and incubated for 15 or 30 min followed by RNA extraction, sequencing of the transcriptomes, and quantification of fold-change compared to the non-targeting control ASO. **a., c**. Heatmap for Log2FC for ΦKZ transcripts at 15 min **(a)** and 30 min **(c)** p.i. **d., f**. Heatmap for Log2FC for PAO1 transcripts at 15 min **(d)** and 30 min **(f)** p.i. **b., e., g**. t-SNE clustering of the $\log_2$fold change (Log2FC) of transcript levels of ΦKZ 15 min **(b)** or of PAO1 15 min **(e)** and 30 min **(g)** p.i. Phage proteins whose knockdown led to a strong phenotype (+++) in the CFU/PFU assay (in Supplementary Table 1) are labelled. **h**. Read coverage of the Pf4 prophage locus in PAO1 upon knockdown of ΦKZ082.

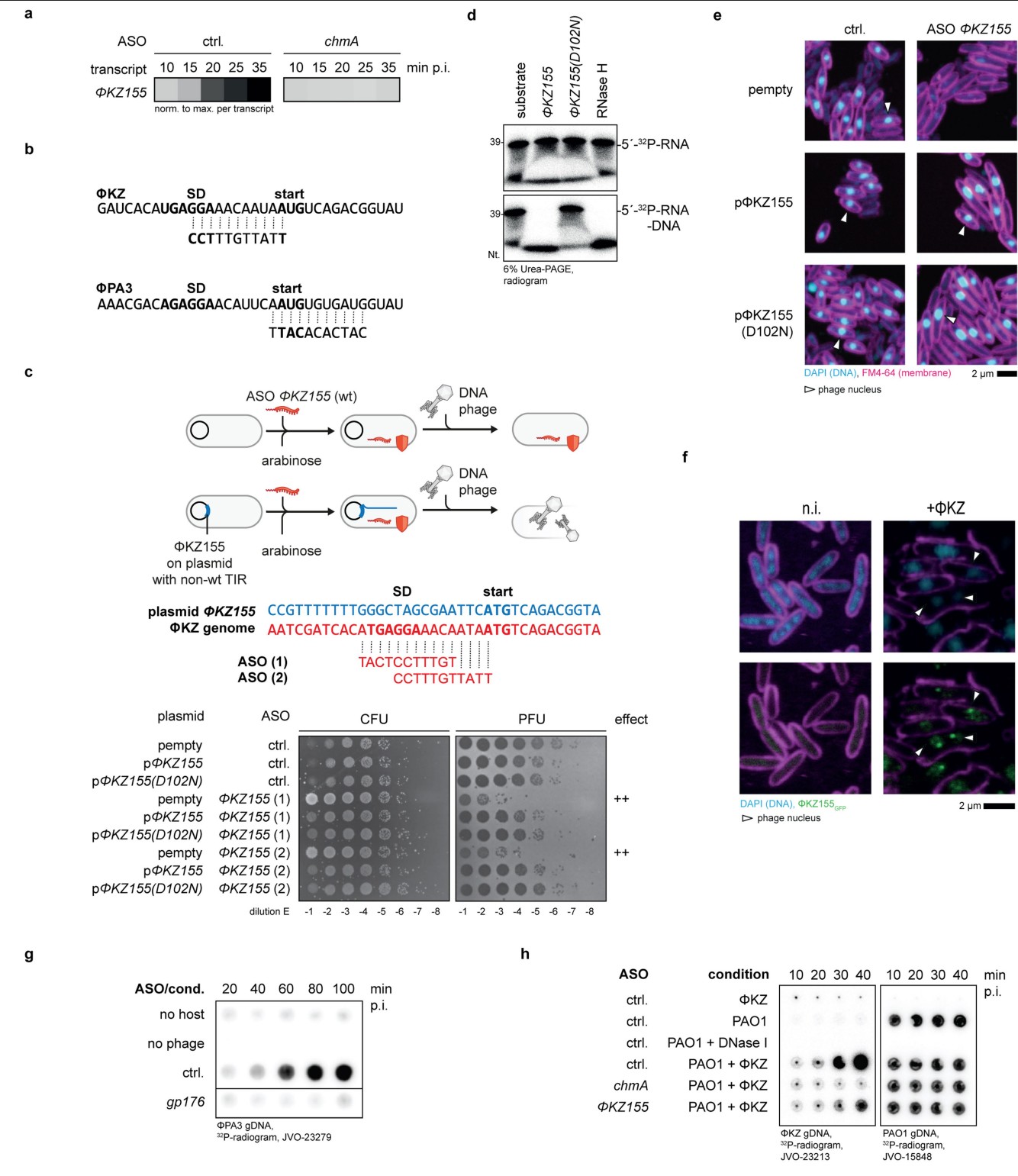

**Extended Data Fig. 8 |** See next page for caption.

**Extended Data Fig. 8 | I ΦKZ155 involvement in the phage replication cycle.**
**a**. PAO1 cells were pretreated with 6 μM ASO against *chmA* for 30 min. Cells were infected with ΦKZ at an MOI = 5 and incubated for the indicated times followed by RNA extraction and sequencing of the transcriptomes. Relative quantification of protein-coding transcript (CDS) reads and normalisation to the maximum for *ΦKZ155* over all conditions (analysis based on data described in Fig. 3a,b). **b**. Sequences of the targeted *ΦKZ155* (ΦKZ) or *gp176* (ΦPA3) transcripts and ASOs. **c**. PAO1 cells were transformed with a complementation plasmid encoding *ΦKZ155* or the catalytically dead mutant *ΦKZ155(D102N)*. PAO1 cells were pretreated with 6 μM ASO against *ΦKZ155* for 30 min. Cells were infected with ΦKZ at an MOI = 0.0001, plasmid-encoded *ΦKZ155* was induced with 0.2% arabinose, and cells were incubated for 180 min followed by CFU/PFU determination. Representative result of two independent experiments. **d**. RNA was 5′-[³²P]phosphorylated and left single-stranded (top) or duplexed with complementary DNA (bottom). ΦKZ155 and the catalytically dead D102N mutant were produced by in vitro translation and added to the oligomer for the cleavage reaction. Subsequently, the oligomers were analysed on an Urea-PAGE gel and autoradiographed. Commercially available *E. coli* RNase H protein was used as control. Representative result of two independent experiments. **e**. PAO1 cells were transformed as in (**c**). PAO1 cells were pretreated with 8 μM ASO against *ΦKZ155* for 30 min. Cells were infected with ΦKZ at an MOI = 10, plasmid-encoded *ΦKZ155* was induced with 0.2% arabinose, and the cells were incubated for 30 min followed by chemical crosslinking and staining with DAPI to visualise DNA and FM4-64 to visualize membranes. Data for pempty and p*ΦKZ155* are based on the same images as shown in Fig. 4d. TIR, translation initiation region. **f**. C-terminally GFP-tagged ΦKZ155 (ΦKZ155$_{GFP}$) was ectopically expressed from a plasmid in PAO1. PAO1 cells were infected with ΦKZ at an MOI = 10, the expression of *ΦKZ155*$_{GFP}$ was induced with 0.2% arabinose, and incubated for 30 min followed by chemical crosslinking and staining with DAPI to visualise DNA and FM4-64 to visualize membranes. Representative result of two independent experiments. **g**. PAO1 cells were pretreated with 8 μM ASO against *gp176* (ΦPA3). Cells were infected with ΦPA3 at an MOI = 5 and incubated for the indicated times followed by DNA extraction and dot-blotting. RNA was eliminated by alkaline treatment. The phage genomic DNA was detected with the radiolabelled oligo probe JVO−23279 followed by autoradiography. No host, no phage, and the non-targeting control ASO served as controls. Representative result of two independent experiments. **h**. PAO1 cells were pretreated with 6 μM ASO against *ΦKZ155* for 30 min. Cells were infected with ΦKZ at an MOI = 5 and incubated for the indicated times post infection followed by DNA extraction and dot blotting, RNA was eliminated by alkaline treatment. Phage genomic DNA was detected with the radiolabelled oligo probe JVO-23213 (complementary to the *chmA* ORF) and JVO-15848 (complementary to the 5S rRNA gene of PAO1) followed by autoradiography. A representative result of two independent experiments is shown. Uncropped images of blots are available in Supplementary Fig. 1.

# Reporting Summary

## Statistics

For all statistical analyses, confirm that the following items are present in the figure legend, table legend, main text, or Methods section.

| n/a | Confirmed | |
|---|---|---|
| ☐ | ☒ | The exact sample size (*n*) for each experimental group/condition, given as a discrete number and unit of measurement |
| ☐ | ☒ | A statement on whether measurements were taken from distinct samples or whether the same sample was measured repeatedly |
| ☐ | ☒ | The statistical test(s) used AND whether they are one- or two-sided<br>*Only common tests should be described solely by name; describe more complex techniques in the Methods section.* |
| ☒ | ☐ | A description of all covariates tested |
| ☒ | ☐ | A description of any assumptions or corrections, such as tests of normality and adjustment for multiple comparisons |
| ☐ | ☒ | A full description of the statistical parameters including central tendency (e.g. means) or other basic estimates (e.g. regression coefficient) AND variation (e.g. standard deviation) or associated estimates of uncertainty (e.g. confidence intervals) |
| ☐ | ☒ | For null hypothesis testing, the test statistic (e.g. *F*, *t*, *r*) with confidence intervals, effect sizes, degrees of freedom and *P* value noted<br>*Give P values as exact values whenever suitable.* |
| ☒ | ☐ | For Bayesian analysis, information on the choice of priors and Markov chain Monte Carlo settings |
| ☒ | ☐ | For hierarchical and complex designs, identification of the appropriate level for tests and full reporting of outcomes |
| ☒ | ☐ | Estimates of effect sizes (e.g. Cohen's *d*, Pearson's *r*), indicating how they were calculated |

*Our web collection on statistics for biologists contains articles on many of the points above.*

## Software and code

Policy information about availability of computer code

| Data collection | no software was used for data collection |
|---|---|
| Data analysis | Image J 1.53, READemption 0.4.3 and 2.0.4, DeSeq2 1.44.0, Integrated Genomic Viewer (IGV) 2.18.4, MaxQuant Perseus 2.1.3.0, Alphafold 3 server (no version given, https://alphafoldserver.com) |

For manuscripts utilizing custom algorithms or software that are central to the research but not yet described in published literature, software must be made available to editors and reviewers. We strongly encourage code deposition in a community repository (e.g. GitHub). See the Nature Portfolio guidelines for submitting code & software for further information.

## Data

Policy information about availability of data

All manuscripts must include a data availability statement. This statement should provide the following information, where applicable:

- Accession codes, unique identifiers, or web links for publicly available datasets
- A description of any restrictions on data availability
- For clinical datasets or third party data, please ensure that the statement adheres to our policy

The genome annotation files used for ASO design are available at the National Center for Biotechnology Information (NCBI) (https://www.ncbi.nlm.nih.gov/) under accession numbers NC_004629.1 (ΦKZ), NC_028999.1 (ΦPA3), NC_001628.1 (PP7), NC_011421.1 (SPO1), NC_041973.1 (RAY). Raw sequencing data and coverage files have been deposited at Gene Expression Omnibus68 for ChmA knockdown and the screen with accession numbers GSE269401 and GSE269911, respectively.

# Research involving human participants, their data, or biological material

Policy information about studies with human participants or human data. See also policy information about sex, gender (identity/presentation), and sexual orientation and race, ethnicity and racism.

| | |
|---|---|
| Reporting on sex and gender | n/a |
| Reporting on race, ethnicity, or other socially relevant groupings | n/a |
| Population characteristics | n/a |
| Recruitment | n/a |
| Ethics oversight | n/a |

Note that full information on the approval of the study protocol must also be provided in the manuscript.

# Field-specific reporting

Please select the one below that is the best fit for your research. If you are not sure, read the appropriate sections before making your selection.

☒ Life sciences  ☐ Behavioural & social sciences  ☐ Ecological, evolutionary & environmental sciences

For a reference copy of the document with all sections, see nature.com/documents/nr-reporting-summary-flat.pdf

# Life sciences study design

All studies must disclose on these points even when the disclosure is negative.

| | |
|---|---|
| Sample size | No sample size calculation was performed. For bacterial assays a sample size of 2 was chosen per condition in independent experiments. Based on experience this is sufficient to make conclusions about effects in the range of more than 2-fold. |
| Data exclusions | We did not observe outliers in our data that needed to be excluded. |
| Replication | Bacterial assays were completed in minimum two independent experiments and the error accounts were reported. ASO experiments were conducted multiple times in independent experiments and/or the gene was targeted with multiple ASOs with a similar outcome. Top hits were reproduced twice at minimum. Biochemical experiments were reproduced twice at minimum in independent experiments with similar outcomes. The sequencing screen assay was conducted once/condition, because we were able to validate the results for the top hits that showed strong phenotypes through comparing two time points at 15 and 30 min per targeted gene. All attempts to replicate data were successful. |
| Randomization | This study did not require randomization. Covariates in microbiological experiments were controlled by restricted and well characterized genotypes (e.g. PAO1) of the strains, identical inoculum, media, growth conditions related to temperature, shaking, and incubation times. The phage stocks were produced in large amounts and tested for titer frequently to ensure reproducibility. ASOs were freshly diluted in a standardized procedure described in the methods section. With this approach, we were able to reduce the systematic influence of covariance. |
| Blinding | This study does not involve procedures that require blinding. The experimental procedures were standardized and the readout of the data was not based on personal assessment. All data were evaluated in raw format by multiple scientists. Microscopy, plaque and biochemical experiments were conducted and interpreted multiple times by different researchers based on raw data. Blinding was not considered critical due to the objective and quantitative nature of the experimental measurements (e.g., CFU/PFU counts, or enzymatic activity assays). In case of microscopy, we aimed to image in one view many cells with infection ratios >90% that resulted in similar phenotypes for cell morphology and inner structures and did not require user-based selection of individual cells. In addition, the selection of the zoom-in images was conducted by other scientists than the ones who recorded the images. |

# Reporting for specific materials, systems and methods

We require information from authors about some types of materials, experimental systems and methods used in many studies. Here, indicate whether each material, system or method listed is relevant to your study. If you are not sure if a list item applies to your research, read the appropriate section before selecting a response.

## Materials & experimental systems

| n/a | Involved in the study |
|---|---|
| ☐ | ☒ Antibodies |
| ☒ | ☐ Eukaryotic cell lines |
| ☒ | ☐ Palaeontology and archaeology |
| ☒ | ☐ Animals and other organisms |
| ☒ | ☐ Clinical data |
| ☒ | ☐ Dual use research of concern |
| ☒ | ☐ Plants |

## Methods

| n/a | Involved in the study |
|---|---|
| ☒ | ☐ ChIP-seq |
| ☒ | ☐ Flow cytometry |
| ☒ | ☐ MRI-based neuroimaging |

## Antibodies

| Antibodies used | Commercial antibody sera were generated by Eurogentec. Rabbits were immunised with the purified ChmA protein. The rabbit serum (no. 2481) was used in a 1:10,000 dilution together with anti-rabbit-HRP antibody (Thermo Scientific, 31460) in a 1:10,000 dilution in 5% BSA/TBST for ChmA detection in immunoblotting. |
|---|---|
| Validation | For anti-rabbit-HRP (Thermo Scientific, 31460), see the product information sheets of the manufacturer; this antibody also showed reliable and consistent results in previous publications from the Vogel lab with various ectopically expressed targets.<br><br>Anti-ChmA Antibody specificity was validated in immunoblotting by comparison between ΦKZ-infected and non-infected cells that yielded a defined band at 70 kDa corresponding for ChmA only in infected cells (Fig. 1c). In addition, we validated it after ASO-based ChmA knockdown, with this approach we were able to discriminate between specific and unspecific interactions. |

## Plants

| Seed stocks | n/a |
|---|---|
| Novel plant genotypes | n/a |
| Authentication | n/a |

