## [Peer Review File · Nature]

Programmable antisense oligomers for phage functional genomics

Corresponding Author: Professor Jörg Vogel

Version 0:

Reviewer comments:

Referee #1

(Remarks to the Author)

In this study Gerovac et al. investigate the use of antisense oligomers (ASOs) to silence bacteriophage genes and interrogate gene function. A key property of ASOs is that they enable gene silencing without the need for genetic manipulation, eliminating an important bottleneck in the investigation of most bacteria and phages. The authors focus their efforts on the *P. aeruginosa* phage phiKZ which was described to form a protein nucleus. This original mode of replication largely remains to be understood at a molecular level and is currently attracting a lot of attention. The authors first characterized the properties of ASOs silencing by targeting *chmA*, showing convincing silencing of *ChmA* production and a dramatic effect on the phage replication. They show how ASOs can phenocopy the effect of a known gene deletions in phiKZ, how they can silence an RNA phage, and be used in a non-genetically amenable bacteria. Altogether this supports the flexibility and usefulness of the technique. They then go on to systematically silence genes of phiKZ and perform RNA-seq, identifying a large number of phenotypes that should lead to important insights into this phage. Finally the authors focus on the essential phage gene Φ KZ155 and its biochemical activity. My first impression reading this manuscript is that this technology and approach could be a small revolution in bacterial genetics. I however have major comments that I believe need to be addressed before I can be truly enthusiastic about the approach and results.

Major comments:

The use of ASO to systematically interrogate gene function in phage biology or bacterial genetics in general is very novel, which increases the impact of the work, but also increases the need to thoroughly characterize the silencing properties. In particular, I believe that it is of primary importance that conclusions are well supported and that alternative explanations are ruled out, which I believe to be lacking in this manuscript.

Importantly the authors describe that 22/176 (12.5%) of the ASOs tested were toxic to *P. aeruginosa*. This suggests that ASO have widespread off-target effects. Only ASOs silencing essential genes of *P. aeruginosa* are expected to be toxic, so this number is likely an undervaluation of the number of ASOs with off-target effects. Given that essential/fitness genes only represent ~1/10th of the total genes, it is likely that the majority of ASOs silence one or several other genes in addition to the target gene. We also have no insights into the way ASOs might impact the expression of genes co-transcribed in the same operon as the target gene.

An investigation of this could eventually be performed using proteomics on *P. aeruginosa* exposed to a few of the ASOs employed.

More details could also be given here regarding the concentration at which various ASOs become toxic.

This makes ASOs suitable for screening purposes, but the results need to be corroborated with alternative methods, such as deleting the genes of the phage before drawing conclusions on their function. This is however not done in the manuscript. Showing that different ASOs targeting the same gene give the same result could be another way to strengthen the conclusions. On that note the authors could better explain the design rules for effective silencing. Some results on the ASO size and position are shown in Extended Data Fig1c, but they are hard to interpret without seeing the sequence next to the figure. Given the novelty of the approach I would recommend putting these results in the main figure with clear schematics. I also wish the authors discussed and investigated more the choice of the CPP, its suitability for different strains and different

species, and the factors that might influence the success of the approach (nature of the cell envelope ? capsule ? other factors ?)

Minor comments:

Fig 1c could show the incubation times and ASO concentration

As a note to the editor and authors, having the figures separated from their legends does not ease the reviewing process.

Referee #2

(Remarks to the Author)

A variety of genetic approaches have been critical to the study of bacteria and their phages from the beginning of molecular biology, but not all bacteria are easily manipulable, even with the advent of CRISPR tools. What has become very obvious over the past few years is that many of our favorite genetic engineering tools come from the ability of bacteria to block phage growth, and from the phage's defenses to these systems, leading to an explosion of interest in studying phages and their interactions with their hosts. Here, Vogel and coworkers have used peptide-nucleic acid (PNA) antisense oligomers (ASOs), to silence genes of a particularly interesting phage of *Pseudomonas aeruginosa*. The paper has two major conclusions: 1) that ASOs, insensitive to host nucleases, are an appropriate tool for studying phage and that this can extend to bacterial hosts that are not otherwise genetically manipulable (blocking the use of CRISPR-based systems) and 2) that dissection of the genes of this phage uncovers new biology, defining putative functions for a number of previously uncharacterized genes. Both points are made in a convincing fashion, even if the phage biology part of the paper is a bit frustrating since nothing is really investigated in depth.

The Vogel lab has investigated ASOs for bacterial studies in a number of previous publications, defining the type of ASO most effective for inhibition of essential genes in *UPEC E. coli* and *Salmonella*, as well as the global effects of using the PNA, beyond the specific target. In those studies and others, effects on bacterial growth were primary measures of efficacy of ASOs. For all such targeting, one basic question is how much expression (whether via inhibiting translation or increasing mRNA degradation) is sufficient to lead to a useful phenotype. Here, the aim is to define important components of a nucleus-forming "jumbo phage", which has a unique lifestyle in which the genome first enters the cell along with its RNA polymerase inside a vesicle (early phage vesicle, EPI), is transcribed and then creates a compartment within the host cell, a "phage nucleus", where it replicates and is packaged into new phage. The genome is thus shielded from CRISPR or other defense systems. However, because the mRNAs are sent to the cytoplasm, outside the nucleus, for translation, they are therefore accessible both to RNA-targeting CRISPR and to ASOs, used here. The authors provide evidence that a non-transformable clinical strain can also be accessed/probed by the ASOs, and this is probably the context in which this approach has the most importance relative to CRISPR-based methods.

1. Some care with nomenclature as well as a better introduction to the phage that is the focus of much of the paper would be useful. For instance:

- a. Throughout, the authors refer to interfering with (or boosting) phage replication. That implies (to me at least) making more copies of the DNA, not necessarily the whole phage infection cycle. Please use some other term. What you are presumably measuring is the burst size (either after one round or three rounds of infection).
- b. More information on the phage infection cycle would help the reader understand what is being done. This can include some idea of how long things take, approximate burst size, and avoid saying, when ASOs block accumulation of plaque forming phage, that they are inhibiting phage replication. As the authors note, an ASO to a lysis gene is sufficient to block plaque formation (although phage may be forming in the cell).
- c. Does this phage degrade its host DNA (as suggested in at least one review of this class of phage)? When in the life cycle? Given that in the DAPI images, all of the DNA appears to be in the nucleus, I assume that the bacterial DNA is gone. Or does the bacterial DNA get incorporated into the nucleus? This is critical to understanding and interpreting the transcriptomic studies in this paper.

2. A bit more attention to the details of what is being done and/or adding some information to figure legends would help the reader.

a. Fig. 1b, lines 130-132: Pretreatment with ASO (for how long? In a later panel (1c) pretreatment is adjusted to 30', but for this initial experiment, MOI and timing was not at all clear. By replication time, assume the authors mean the time from infection to phage release?

b. Fig. 1d: the text eventually says what the MOI is for this experiment, but adding it to the legend would be more useful. It would also be better to have a different arrow type/symbol for the EPI and the nucleus (and in other figures, particularly 5b).

3. Given that the authors want to emphasize the general usefulness of the ASO approach, a bit more information on why particular protocols were chosen would be useful. For instance:

a. Fig. 2c, in which an ASO is silencing the *jukA* anti-phage defense system, uses repeated ASO pre-treatment and apparently multiple times after infection as well? Why is this repeated addition needed? What happens if it is only added once beforehand, and is there an explanation that would help someone hoping to use this in their own work?

b. Fig. 2d, with the RNA phage PP7, ASOs were added every 30'. Again, why is this necessary in this case? Is it different for targeting a lysis gene or the polymerase (where there should, presumably be less RNA made)?

4. Figure 3: There is a great deal of data here, but that also leads to some questions.

a. Given the three cycles of infection, would classes of ASOs that lead to formation of defective phage particles, lysing the first infected cells but not the next round, be missed? Presumably for this, a high MOI in a single round to look possibly for loss of CFUs but decreased PFUs would be expected? It would be useful to discuss the limits as well as the advantages of the approach used here with very low MOI.

b. A few minor questions:

i. what is meant by omitting structural genes, since there seem to be capsid and tail genes.

ii. Why does one entry have parentheses around the results column (014).

iii. Are the levels of ASOs used for all of these (early vs. late genes, for instance) similar?

5. A major point in this paper is that unknown genes can be linked in terms of likely roles by their behavior, with transcriptome changes included as part of that behavior. That seems to result in the clustering shown in Fig. 5a. The two clusters that are surrounded by dotted lines are discussed a bit in the text; what about other apparent clusters? It is striking that the clustering is very different in extended fig. 5e. Maybe a comment on this would be useful. If I understand the model for protein 155, the authors think it doesn't have any effect on phage genes because there is little or no DNA replication. I would think that would lead to much less RNA as well, or is all RNA made from an initial DNA copy? Are the other members of this cluster that have phage growth phenotypes either very late genes or might they also be involved in DNA replication? I would have found more depth for a couple of stories rather than the mention of various genes of possible interest more useful here.

6. One of the limits of the ASO approach as presented here is that it effectively knocks down translation, but that it is not necessarily useful to then dissect the function of the protein of interest. For instance, for the RNase H protein in Fig. 5, is it possible to complement an ASO (for a manipulable host) with a plasmid resistant to the ASO, expressing WT or mutant forms of the protein of interest?

7. The data sets 2 and 3 could use much more informative legends. Color code as used in other figures? Identify what the columns are, where to look? In the discussion, protein 072 as a predicted nuclease that "does not require phage nucleus expression" (lines 458-460), with a reference to Supplementary Data 2. I assume this is because expression is similar with or without the ASO to ChmA? Maybe that could be spelled out.

8. As the authors note, the ASO screening here was done under one condition in one host, and some ASOs to core genes had no phenotype (Fig. 3). However, beyond not being in the right host, for instance, I would imagine that other means for "escaping" inhibition might be worth mentioning: redundancy in function (presumably testable with combinations of ASOs?), a requirement for only a small amount of a product (what is the leakiness of the ASOs), or possibly highly abundant, stable RNAs would be more difficult to inhibit. A bit of discussion of these complications would be useful. I couldn't find information on the degree to which the two ASOs used for each target agreed, providing some sense of how good coverage would be expected to be.

Referee #3

(Remarks to the Author)

The article describes a system utilizing programmable antisense oligomers (ASO) technology to specifically silence genes of bacteriophage during infection. This represents an innovative and potentially highly impactful methodology. I concur with the authors that this approach holds significant promise for advancing both fundamental research on phage infection and its applications. The authors have done large-scale and impressive work, demonstrating the method's abilities for DNA and RNA phages as test subjects. This study is indeed worthy of publication. However, certain aspects require clarification.

The majority of the study demonstrating this technique is based on investigations of the giant bacteriophage phiKZ. Due to the extensive protection of phiKZ DNA from genetic manipulation at all stages of infection, this phage represents a challenging model; thus, the findings from this study are also crucial for understanding phiKZ development. And some aspects of data interpretation could benefit from adjustment.

1. The phiKZ phage encodes two distinct RNA polymerases (RNAPs), one of which, vRNAP, is injected into the cell along with the phage DNA. In this study, the PHIKZ176 vRNAP subunit (as described in Julie A. Thomas et al., 2016) was inactivated using ASO treatment. This important detail should be taken into account when interpreting the results.

2. Additionally, non-virion RNA polymerase (nvRNAP) subunits were also targeted in the study. However, only two subunits, PHIKZ55 and PHIKZ68, were explicitly noted as nvRNAP components. The ORF annotated as PHIKZ56.1 appears to be the terminal portion of PHIKZ55 (PDB: 7OGP_A; PSI-BLAST alignment with nvRNAP subunits in phiKZ-like phages), while PHIKZ73 represents a segment of the Gp71-73 subunit. The subunit composition of nvRNAP is documented in the works by Yakunina et al. (2015), Orekhova et al. (2019), and the structural studies of nvRNAP by Natàlia de Martín Garrido et al. (2021/2024). Based on this information, I question whether the using PHIKZ56.1 and PHIKZ73 are fitted for methods demonstration due to the design of ASO (rbs and start-codon), as the observed effects may relate not to protein synthesis inhibition but rather to mRNA splicing.

3. Continuing from the previous point, the presence of introns in coding sequences for subunits of both virion and non-virion RNA polymerases has been noted in several giant bacteriophages related to phiKZ (Pieter-Jan Ceysens et al., 2014; Daria Lavysch et al., 2016). Within these introns, various potential homing nucleases are often predicted. Specifically, in this study, this applies to ORFs PHIKZ56 (nuclease within the intron of Gp55), PHIKZ72 (nuclease within the intron of Gp71-73), and PHIKZ179 (nuclease within the intron of the Gp180 vRNAP subunit). The impact of these nucleases may be linked to both nuclease synthesis inhibition and interference with splicing processes.

4. The results presented in lines 284-287—"However, upon chmA knockdown, reads of phage genes expressed at intermediate times were strongly depleted from 15 min p.i. onward, indicating that formation of the phage nucleus is required for expression of these genes by the nvRNAP"—are complementary to the transcription regulation scheme for phiKZ described in Antonova D. et al. (2023). ChmA synthesis inhibition likely blocks phage DNA release from the EPI vesicle,

leaving it inaccessible to ν RNAP.

5. The authors demonstrated that PHIKZ155 ASO treatment significantly impacted infection progression due to its involvement in phage DNA replication. However, another statement about its role in phage nucleus formation is not convincing enough. First, the quality of the fluorescence image (Fig. 5b) cannot confirm the position of phage DNA relative to cell boundaries. Please consider improving these images. Moreover, the “Curiously, the Φ KZ155GFP protein was barely detectable in the absence of phage infection, indicating that it requires the phage nucleus for its stability” appears somewhat speculative. If PHIKZ155GFP abundance is based solely on fluorescence imaging, the interpretation should consider that a concentrated molecular signal will appear brighter than a signal diffused across the cell. Furthermore, PHIKZ155GFP stability may rely on interaction with a single protein partner rather than the entire phage nucleus.

Minor queries:

1. Different procedures were used for ASO treatment across experiments (e.g., single vs. multiple ASO administrations for ϕ iKZ genes, bacterial genes, and PP7 infection). Could you clarify the rationale for these differences?
2. Please update the reference for Antonova et al., “Genomic transfer via membrane vesicle: A strategy of giant phage ϕ iKZ for early infection,” from BioRxiv to the Journal of Virology.

Version 1:

Reviewer comments:

Referee #1

(Remarks to the Author)

The authors have now added a substantial amount of novel data to the manuscript which address all my previous concerns. I congratulate them on a truly excellent piece of work and I recommend it for publication.

Referee #2

(Remarks to the Author)

In this revised version of a previously reviewed ms by J. Vogel and collaborators, the authors present extensive data supporting the use of ASOs to analyze unknown gene functions in phage, in particular in nucleus-forming phage in *Pseudomonas*. They show all of the characteristics one would want for dissecting functions without the need for the bacterial cell to accept genetic material. This is a novel, well-developed and highly useful approach that will be important in the growing interest in probing gene function in a range of bacteria and bacteriophages. The revised manuscript makes the logic of the various experiments much clearer as well as adding further examples demonstrating the breadth of the approach. The comments here are for a few additional minor clarifications.

1. The result shown in rebuttal fig. R8 would be relevant to include in the paper (as a supplemental figure, briefly discussed in the text).
2. In Fig. 1d, the CFU don't seem to decrease when the PFUs are inhibited with ASO, while in other experiments, there is clear evidence of killing of the host when the phage can grow (extended data figure 1). Different protocol? Maybe worth a mention.
3. The authors point out (text, lines 217-223, extended Fig. 3) that inhibition of translation of a given gene does not affect expression of other genes in that operon, suggesting that in this organism, there is not a robust mechanism for degradation or termination in the absence of translation. The authors note that mRNAs are depleted in *E. coli*, however (lines 371-374). This is an important and interesting difference that may be worth a bit more discussion (and/or note if there are parallel *E. coli* experiments to mention with lines 217-223).
4. The ability to complement back with an ASO-resistant gene is, as the authors note, extremely useful, and this is a very nice addition (although only possible with organisms that are amenable to the genetics, introducing the ectopic copy, of course). I'd suggest making the figure (5d) a bit clearer, however; I had to go back and look at everything carefully to understand what I was seeing. Emphasize in legend and maybe by making the plasmid copy of the gene in the schematic a different color (not red, color of ASO), that this doesn't contain the target for the ASO. It is easy to miss the p, or not be thinking why the plasmid version is resistant. Color change could then also be used with the sequence to make it clearer what goes with what.
5. It was unclear to me why the information in Extended Fig. 8c-f wasn't in the results. Doesn't seem like it belongs in the discussion, even if it is not fully explored thus far.

Referee #3

(Remarks to the Author)

Thank you to the authors for their work and for addressing the comments raised by all reviewers. The expanded discussion on the ASO design and the demonstration of its broad applicability significantly improve the manuscript and enhance its value for future wide use. Below, I provide my comments on the authors' replies.

1. Unfortunately, the potential impact of this gene's inhibition on the activity of ν RNAP is still not mentioned in the manuscript. The observed reduction in ChmA levels could be associated with a decreased amount of active ν RNAP in the phage particle, which may reduce the efficiency of ChmA transcription during the second and third rounds of infection in your screening experiment. Supporting this hypothesis is the fact that in the single-round infection experiment, which you conducted to collect material for RNA-seq, there is no observed difference in the amount of ϕ iKZ transcript (Fig. 4c), which

would be expected if the decrease in ChmA were a direct effect.

line 325: as well as the uncharacterised core genes Φ KZ147, -153, -161, -176

2. First, I would like to draw the authors' attention to the fact that both transcripts in question are still present in Fig. 3. Although the authors acknowledge in their response to the reviewer that in these particular cases the ASO effect likely does not result from translation inhibition ("This would suggest that they do not inhibit translation, explaining the mild phenotypes observed"), in the main text of the manuscript these "exons" (to the extent that this term can be applied to phage genomes, though it seems the most accurate descriptor of these ORFs) continue to be discussed as individual genes, alongside other protein-coding genes. Consequently, the effects of ASO targeting them could be interpreted by readers as effects on translation. However, in these two cases, it remains unclear what the ASOs are targeting and how the observed effects should be interpreted.

Line 326: "and the non-core genes Φ KZ056.1"

Lines 394–397: "At 30 min p.i., ASO-mediated depletion of ChmA, the nvRNAP subunit Φ KZ055, the nvRNAP sigma factor Φ KZ068, PicA (Φ KZ069), Φ KZ056.1, -122, -124, and -151 showed a strong effect on phage transcription (Fig. 4c)"

Lines 411–413: "Equally interesting are the knockdowns of Φ KZ073 and -082, which have no general effect on the PAO1 transcriptome but specifically induce transcription of the Pf4 prophage locus"

Lines 416–417: " Φ KZ might use the Φ KZ073 and Φ KZ082 proteins as part of a specific inhibitory mechanism against the Pf4 prophage to ensure its own propagation"

3. Thank you for checking the potential ASO effect on splicing. It might be helpful to map the splicing regions directly onto the corresponding sequences, indicating the positions of the ASOs and primers used in the analysis. This could allow for more concrete conclusions regarding splicing efficiency and ASO design in the presence of introns within phage genes. For example, based on the provided images, different ASOs appear to result in varying amounts of unspliced RNA (lower panels in the gels). Notably, the highest levels of unspliced RNA are observed in the control samples without ASO treatment in both cases.

This observation may suggest that the "intron–exon" structure, present in both subunits of the nvRNAP and in one subunit of the vRNAP (including both phage RNA polymerase subunits that carry the essential DFDGD motif for RNA polymerization activity), could play a role in regulating phage genome transcription.

4. Let me clarify the point more precisely. According to Ceysens et al. 2014, three classes of phage genes are distinguished in phiKZ: early, middle, and late, each transcribed from distinct promoter sequences. Based on the model proposed by Antonova et al., 2023, only the transcription of late genes is strictly dependent on the formation of the mature nucleus, whereas the middle gene class, which although transcribed by nvRNAP, initiates before complete phage nucleuse maturation and likely involves a form of nvRNAP that differs in composition from that associated with late promoters. This raises an important question: does ChmA depletion affect transcription of both middle and late genes, or only the late ones? Unfortunately, Extended Data Figure 6 does not provide definitive evidence that both classes are blocked. Based on the figure, phage transcripts accumulate up to 25 minutes post-infection, albeit at reduced levels compared to the control. This decrease might reflect a lack of late gene transcription rather than a complete arrest of phage transcription. This distinction could be crucial for understanding the dynamics of infection. If middle gene transcription still occurs upon ChmA translation knockdown, it would imply that DNA release from EPI vesicles can proceed, at least partially, in the absence of significant accumulation of the major nuclear shell protein. Such a finding could offer new insights into the stability of EPI vesicles.

5. No additional comments. Thank you for the answer.

Version 2:

Reviewer comments:

Referee #3

(Remarks to the Author)

I would like to thank the authors for addressing my questions and for their careful attention to my comments.

I would like to add a brief remark regarding the authors' response to Reviewer #2 concerning the absence of bacterial DNA degradation during phiKZ phage infection. A similar observation was previously reported in the paper by Yana A. Danilova et al. 2020. Therefore, it appears clear that no degradation of host cell DNA occurs in cells during phiKZ infection, at least in the first half of the infection cycle. The reference to this paper, together with the observations described in the present manuscript, which show the same results, will likely help resolve the uncertainty on this issue that has existed in the literature.

In conclusion, I strongly recommend the manuscript for publication.

Point-to-point reply to the referees' comments (ms. no.: 2024-09-18911)

We thank all reviewers for their encouraging comments and valuable suggestions that allowed us to improve our manuscript. The main concerns that were raised were the specificity and robustness of the ASO-mediated knockdown approach across hosts and functional validation of observed effects by alternative methods. To address these points, we have performed new experiments and analyses

- to demonstrate ASO knockdown of phage genes in additional *Pseudomonas* strains as well as in other Gram-negative and Gram-positive bacteria (new Extended Data Fig. 4, new Fig. 2e-f);
- to show the specificity of ASO-mediated mRNA suppression at the protein level through proteomics (new Extended Data Fig. 3);
- to improve ASO design rules via tiling of the *chmA* mRNA 5' region with many ASOs (new Extended Data Fig. 2);
- to complement ASO-mediated knockdown of the newly discovered essential checkpoint protein Φ KZ155 by expressing its wild type or mutant form from a plasmid (new Fig. 5d,e, new Extended Data Fig. 8e,f);
- to rule out ASO-mediated effects on splicing of phage-borne autocatalytic introns (Fig. R12 for reviewers).

In addition, we responded to all technical concerns raised by the reviewers. For a detailed response to all questions raised by the referees, please see our point-by-point rebuttal below.

Text changes have been marked in red in the revised manuscript.

Referee #1 (Remarks to the Author):

In this study Gerovac et al. investigate the use of antisense oligomers (ASOs) to silence bacteriophage genes and interrogate gene function. A key property of ASOs is that they enable gene silencing without the need for genetic manipulation, eliminating an important bottleneck in the investigation of most bacteria and phages. The authors focus their efforts on the *P. aeruginosa* phage ϕ KZ which was described to form a protein nucleus. This original mode of replication largely remains to be understood at a molecular level and is currently attracting a lot of attention. The authors first characterized the properties of ASOs silencing by targeting *chmA*, showing convincing silencing of ChmA production and a dramatic effect on the phage replication. They show how ASOs can phenocopy the effect of a known gene deletions in ϕ KZ, how they can silence an RNA phage, and be used in a non-genetically amenable bacteria. Altogether this supports the flexibility and usefulness of the technique. They then go on to systematically silence genes of ϕ KZ and perform RNA-seq, identifying a large number of phenotypes that should lead to important insights into this phage. Finally the authors focus on the essential phage gene Φ KZ155 and its biochemical activity. My first impression reading this manuscript is that this technology and approach could be a small revolution in bacterial genetics. I however have major comments that I believe need to be addressed before I can be truly enthusiastic about the approach and results.

Major comments:

The use of ASO to systematically interrogate gene function in phage biology or bacterial genetics in general is very novel, which increases the impact of the work, but also increases the need to thoroughly characterize the silencing properties. In particular, I believe that it is of

primary importance that conclusions are well supported and that alternative explanations are ruled out, which I believe to be lacking in this manuscript.

Importantly the authors describe that 22/176 (12.5%) of the ASOs tested were toxic to *P. aeruginosa*. This suggests that ASOs have widespread off-target effects. Only ASOs silencing essential genes of *P. aeruginosa* are expected to be toxic, so this number is likely an undervaluation of the number of ASOs with off-target effects. Given that essential/fitness genes only represent ~1/10th of the total genes, it is likely that the majority of ASOs silence one or several other genes in addition to the target gene screen.

REPLY: We completely agree with the reviewer that, similar to nucleic acid-targeting knockdown screens using siRNAs or CRISPRi, off-targets are a concern when screening with ASOs. We have put a lot of effort into understanding the degree of off-targeting activity of delivered ASOs in different bacteria over the past few years. To this end, we established RNA-seq as a readout to look at the selectivity of ASO activity (Popella et al. 2021, 2022 *NAR*, PMID: 33849070, PMID: 35687096; Jung et al. 2023 *RNA*, PMID: 36750372) and developed INRI-seq, a variant of ribosome profiling that operates on synthetic transcriptomes (Hör et al. 2022 *NAR*, PMID: 36229039). The upshot of these analyses is that ASO off-targeting is predictable. For now, there is little evidence that ASOs affect transcript levels or translation by binding outside the narrow translation initiation region of bacterial mRNAs. Accordingly, our new global proteomics data described below, which covers many host and phage proteins, show no substantial off-target effects.

Our ASO design algorithm called MASON (Jung J et al. 2023 *RNA*, PMID: 36750372) accounts for complementarity (including mismatches) to other genomic translation initiation regions. In the current study, we have used this algorithm to choose ASOs with few or no off-target predictions in both the phage and host transcriptome (added to **Supplementary Table 1**). In addition, we ruled out off-target effects by validating positive hits (i.e., targets for which a knockdown caused a strong phenotype in the CFU/PFU assay (+++)) by multiple ASOs.

An alternative explanation for the observed toxic effects of some of the ASOs in our screen is that they enhance the activity of the attached CPP (here, RXR, which is derived from an antimicrobial peptide). We have just dissected such a case for a specific RXR-ASO conjugate in *Fusobacterium nucleatum* (Cosi V et al. 2025 *mBio*, in press; preprint doi: 10.1101/2025.02.12.637808). Therefore, the fact that 12.5% of the ASOs we designed are toxic to *P. aeruginosa* does not necessarily indicate wide-spread off-target effects.

For now, our recommendation is to exclude toxic ASOs by a simple pre-screening on non-infected bacteria using viability assays. This is a workable approach because we show with a *chmA* mRNA tiling experiment described further below that many ASOs can be designed for the same target. In addition, several laboratories including our own are working on new delivery methods, e.g., ASO conjugation to siderophores that work by active uptake rather than passive diffusion across the envelope, as in the case of CPP carriers. We have now discussed these points in a new section 'ASO design and specificity' (p. 7ff.).

We also have no insights into the way ASOs might impact the expression of genes co-transcribed in the same operon as the target gene. An investigation of this could eventually be performed using proteomics on *P. aeruginosa* exposed to a few of the ASOs employed.

REPLY: We have performed the proposed proteomics experiment for three different ASO targets—*chmA*, *nvRNAP* and *picA*—all of which are co-transcribed with at least one more gene. As shown below, each of these ASOs only suppresses the synthesis of the targeted

cistron. For example, the anti-*chmA* ASO reduced protein levels of ChmA (Φ KZ054) but not of nvRNAP (Φ KZ055) or HNH nuclease (Φ KZ056), which are encoded downstream of *chmA* on the same polycistronic mRNA. Also in the di-cistronic *phiKZ068-069* mRNA, the anti-*picA* ASO did not alter protein output from the upstream sigma factor-encoding gene *phiKZ068*, while PicA protein levels were reduced ~60-fold (Fig. R1a-c). In addition to the ChmA western blots shown previously in the manuscript (Fig. 1b), these data clearly demonstrate that ASOs successfully reduce protein levels.

These new results also support the selectivity of the ASO knockdown approach and have been added as **new Extended Data Fig. 3c-e**. The experiment also allowed us to take another look at the off-target effects of these ASOs. Among the 1,130 host proteins we were able to detect with high confidence, we observed very limited alterations in protein levels (**new Extended Data Fig. 3a**). On the phage side, we observe a specific downregulation of the on-target proteins ChmA, nvRNAP and PicA (**new Extended Data Fig. 3b**) but no broad effect on other phage proteins.

We would like to emphasize that these new results not only demonstrate the specificity of our ASO approach to phage genes, but also provide value for the growing antibacterial ASO community. To the best of our knowledge, this is the first instance of a global proteomics experiment conducted after ASO treatment. These results are discussed in the new section 'ASO design and specificity' (p. 7ff).

Fig. R1 | Proteome analysis after ASO knockdown indicates ASO specificity. a-c. Relative proteins levels and Log₂FC 10 min p.i. were determined based on LFQ counts. Transcriptional units (TU) were derived from Putzeys et al. 2024 *microLife* (PMID: 38444699).

More details could also be given here regarding the concentration at which various ASOs become toxic.

REPLY: To address this question systematically, we titrated ASOs targeting *chmA*, *nvRNAP*, *picA* and Φ KZ155 on uninfected *Pseudomonas* cells and determined CFU after three rounds of bacterial replication (**Fig. R2**). These RXR-ASO conjugates only started to show toxicity at $\sim 12 \mu\text{M}$ (or higher in the cases of *picA* or Φ KZ155), double the concentration used in our screen. It is therefore unlikely that the screening results are influenced by overlaid toxic effects. We included the data for the control and *chmA* ASO now in **Extended Data Fig. 1d**.

Fig. R2 | Concentration-dependent toxicity of ASOs. ASOs targeting the indicated phage transcripts were titrated from 3-18 μM and PAO1 cells were incubated for 3.5 h followed by spotting to determine CFUs.

This makes ASOs suitable for screening purposes, but the results need to be corroborated with alternative methods, such as deleting the genes of the phage before drawing conclusions on their function. This is however not done in the manuscript.

REPLY: We fully agree with the reviewer that an independent phenotypic assessment of ASO knockdown effects is important, although this is difficult for essential phage genes, because per definition, such genes cannot be deleted in the phage. Therefore, in the original manuscript, we opted to demonstrate how ASO knockdown phenocopies gene deletion with a non-essential and a conditionally essential gene.

The first example is the non-essential PhuZ spindle protein, which is known to position the phage nucleus in the cell center (e.g., Kraemer JA et al. 2012 *Cell*, PMID: 22726436). We showed that ASO-mediated knockdown of PhuZ leads to a decentralised phage nucleus

(previous **Extended Data Fig. 2a,b**), identical to the phenotype of a $\Delta phuZ$ phage described in Guan et al. 2022 *Nature Microbiology* Fig. 2a,b, PMID: 36316452. We have now used this $\Delta phuZ$ phage as a reference and added these new data together with the ASO knockdown to main **Fig. 1f,g**, so it will be more obvious for the reader.

The second example is ASO knockdown of $\Phi KZ014$; this phage-encoded ribosomal protein is essential in the clinical isolate PaLo44 but not in the PAO1 lab strain (Gerovac et al. 2024 *Nature Microbiology*, PMID: 38443577). We show in **Fig. 2c** that an anti- $\Phi KZ014$ ASO phenocopies the essentiality of the $\Phi KZ014$ gene in the PaLo44 strain. We have clarified this in the revised manuscript (**p. 9**).

Lastly, along the line of validation with an alternative method, we now also demonstrate complementation of ASO knockdown by providing the essential gene in question on a plasmid with an ASO-resistant 5'-mRNA region; see further below.

Showing that different ASOs targeting the same gene give the same result could be another way to strengthen the conclusions. On that note the authors could better explain the design rules for effective silencing.

REPLY: The established mode of action of antibacterial ASOs is translational inhibition through sequestration of mRNA sequences required for 30S ribosome binding. In our screen, we designed two ASOs for each target, which bind either Shine-Dalgarno (SD) or start codon sequences, i.e. the most crucial sequence elements for stable ribosome binding.

To address the reviewer's point more systematically for a phage mRNA, we have now performed a high-density tiling experiment to extend the number of ASOs targeting *chmA* (**Fig. R3, new Extended Data Fig. 2b**). Specifically, we designed 38 additional 11-mer ASOs that bind within a -37-nt and +44-nt window relative to the start codon of the *chmA* mRNA. The tiling was done in single-nucleotide steps in proximity to the SD and AUG. Inhibition of the *chmA* mRNA was determined by ChmA western blot analysis (direct readout) and CFU/PFU assays (indirect readout). Remarkably, we identified 19 ASOs that bind at or close to the SD or AUG and strongly inhibited ChmA synthesis and phage replication. Our analysis also indicates that ASOs directed against the A/U-rich region between the SD and AUG fail to repress ChmA protein synthesis, as expected because of their predicted low melting temperature.

Combined with unpublished data from a screen of close to 300 essential genes in *E. coli* that were targeted with two different PNAs (Popella L, Jung J, et al., manuscript in preparation), we use a set of parameters to predict effective ASOs. These include a melting temperature between 45-55 °C, avoiding long stretches of purines and self-complementarity, and a minimal number of predicted off-targets. To clarify this, we have expanded the discussion of these ASO design rules and include a clear recommendation to use multiple ASOs for a given target for cross-validation (p. 7ff). Gratifyingly, the results of the *chmA* mRNA tiling experiment—now included in the manuscript (**Extended Data Fig. 2b**)—show that it is possible to design multiple effective ASOs per target to mitigate the risk of off-targeting and false-positive readout of phenotypes.

Fig. R3 | ASO tiling indicates that multiple effective ASOs can be designed for a target transcript. Left. All tiling ASOs were assayed using the CFU/PFU assay and ChmA immunoblotting (left) in Φ KZ-infected cells at MOI 10 after 20 min. Right. Parameters (self-complementarity (SC_bases), melting temperature (Tm), purine percentage) are listed. The number of off-targets (OT) in the translation initiation region (TIR) with 0 or 1 mismatch (mm) are given; the relevance for off-targets decreases with the number of mismatches. In our screen we used JVPNA-72 and JVPNA-73 because they have no predicted off-target sequences in other TIRs of the Φ KZ or *P. aeruginosa* PAO1 genome.

Some results on the ASO size and position are shown in Extended Data Fig1c, but they are hard to interpret without seeing the sequence next to the figure. Given the novelty of the approach I would recommend putting these results in the main figure with clear schematics.

REPLY: Unfortunately, we are limited in space for the main figures and would like to keep these data in the Extended Data Figures together with the ASO optimisation and the tiling experiment that deals with specificity and versatility. However, we have now split the panel into a part that is focusing on the different ASO-lengths in **Extended Data Fig. 1d** and a part that focuses on specificity. The latter is now discussed in detail in the results section (p. 7ff). The data obtained using ASOs with mismatches is now included in the **new Extended Data Fig. 2a**. Throughout the manuscript, we have added ASO and target mRNA sequences to illustrate the binding site. In addition, we have added the nucleotide sequence of the 5' region of *chmA* and the ASOs used to the main **Fig. 1B**, to make it easier for the reader to visualize the main principle of the ASO approach.

a

b

Fig. R4 I ASO lengths and mismatch effects. a. Different ASO lengths between 9 to 12-mers were tested in a CFU/PFU assay. **b.** Mismatches (mm) were tested between 2 to 4 nucleotides.

I also wish the authors discussed and investigated more the choice of the CPP, its suitability for different strains and different species, and the factors that might influence the success of the approach (nature of the cell envelop ? capsule ? other factors ?)

REPLY: We chose RXR for our experiments, because this CPP has been shown to efficiently deliver ASOs to *Pseudomonas* (Ghosal A & Nielsen PE 2012 *Nucleic Acids Therapeutics*, PMID: 23030590; Howard JJ et al. 2017 *Antimicrobial Agents and Chemotherapy*, PMID: 28137807). RXR was also used to knock down essential genes in other Gram-negative γ -proteobacteria such as *E. coli*, *Salmonella* and *Klebsiella* and in Gram-positive *Staphylococcus* (reviewed in El-Fateh M et al. 2024 *International Journal of Antimicrobial Agents*, PMID: 38185398).

Nonetheless, to address the reviewer's request experimentally, we have now also tested two additional well-established CPPs, KFF and TAT, for their efficiency to deliver ASOs into *Pseudomonas* PAO1 and inhibit phage replication through targeting *chmA*. As shown below, a TAT-ASO conjugate worked as well as the original RXR-ASO, whereas the KFF-coupled ASO did not inhibit phage replication (**Fig. R5, new Extended Data Fig. 1c**). The latter is likely due to inefficient delivery. Whereas KFF-ASOs works well with *E. coli* and *Salmonella*, they are known to enter *Pseudomonas* with poor efficiency, potentially due to different uptake mechanisms in these bacteria (Ghosal A & Nielsen PE 2012 *Nucleic Acid Therapeutics*, PMID: 23030590).

Fig. R5 | Cell penetrating peptide selection for KD in *Pseudomonas*. RXR, KFF, and TAT CPPs were tested for their efficiency to deliver *chmA*-targeting ASOs, using the CFU/PFU assay as readout.

In general, CPP-mediated uptake is affected by outer membrane composition (Ebbensgaard A et al. *Frontiers in Microbiology*, PMID: 30245684). It has also been shown that LPS composition affects the susceptibility of bacteria to KFF-PNA conjugates (Goltermann L et al. 2022 *Frontiers in Microbiology*, PMID: 35794919). Regarding capsule, we have unpublished results from a *Klebsiella* project that tell us that capsule might not be a barrier for antibacterial ASOs. Nevertheless, to what extent envelope variations or other unknown factors contribute to the sensitivity of bacterial strains to specific CPPs will need further investigation. Ultimately, certain CPPs are more effective in some species than others, and the selection should be based on the host bacteria rather than the phage. We have now discussed these points in the manuscript in a separate paragraph in the discussion (p. 19).

Minor comments:

Fig 1c could show the incubation times and ASO concentration

REPLY: We have added the incubation times and ASO concentrations.

As a note to the editor and authors, having the figures separated from their legends does not ease the reviewing process.

REPLY: We perfectly understand the reviewer's suggestion but there is little we can do here as we must follow the submission guidelines for *Nature*.

Referee #2 (Remarks to the Author):

A variety of genetic approaches have been critical to the study of bacteria and their phages from the beginning of molecular biology, but not all bacteria are easily manipulable, even with the advent of CRISPR tools. What has become very obvious over the past few years is that many of our favorite genetic engineering tools come from the ability of bacteria to block phage growth, and from the phage's defenses to these systems, leading to an explosion of interest in studying phages and their interactions with their hosts. Here, Vogel and coworkers have used peptide-nucleic acid (PNA) antisense oligomers (ASOs), to silence genes of a particularly

interesting phage of *Pseudomonas aeruginosa*. The paper has two major conclusions: 1) that ASOs, insensitive to host nucleases, are an appropriate tool for studying phage and that this can extend to bacterial hosts that are not otherwise genetically manipulable (blocking the use of CRISPR-based systems) and 2) that dissection of the genes of this phage uncovers new biology, defining putative functions for a number of previously uncharacterized genes. Both points are made in a convincing fashion, even if the phage biology part of the paper is a bit frustrating since nothing is really investigated in depth.

The Vogel lab has investigated ASOs for bacterial studies in a number of previous publications, defining the type of ASO most effective for inhibition of essential genes in UPEC *E. coli* and *Salmonella*, as well as the global effects of using the PNA, beyond the specific target. In those studies and others, effects on bacterial growth were primary measures of efficacy of ASOs. For all such targeting, one basic question is how much expression (whether via inhibiting translation or increasing mRNA degradation) is sufficient to lead to a useful phenotype.

REPLY: We agree that this is an important basic question, so we decided to address it experimentally in our system. We chose the *chmA* mRNA as a target, because of the availability of a specific antibody against the ChmA protein, allowing us to quantify knockdown efficiency by western blot. We titrated ASO concentrations to achieve different knockdown levels prior to phage replication. Our results show that for this essential phage nucleus protein, >90% knockdown is required to fully block phage replication (**Fig. R6**).

Fig. R6 | ASO-based ChmA KD in *Pseudomonas*. **a.** PAO1 cells were grown to OD 0.3 and treated with the indicated ASO concentration for 30 min, followed by infection with Φ KZ at a MOI=0.0001, PFU/CFU were determined after 180 min p.i.. **b.** PAO1 cells were grown to OD 0.3 and treated with indicated ASO concentration for 30 min, followed by infection with Φ KZ at a MOI=5, at 30 min p.i. ChmA levels were quantified by western blot probed with an anti-ChmA antibody. **c.** Quantification of three biologically independent replicates from (b).

Here, the aim is to define important components of a nucleus-forming “jumbo phage”, which has a unique lifestyle in which the genome first enters the cell along with its RNA polymerase inside a vesicle (early phage vesicle, EPI), is transcribed and then creates a compartment within the host cell, a “phage nucleus”, where it replicates and is packaged into new phage. The genome is thus shielded from CRISPR or other defense systems. However, because the mRNAs are sent to the cytoplasm, outside the nucleus, for translation, they are therefore accessible both to RNA-targeting CRISPR and to ASOs, used here. The authors provide evidence that a non-transformable clinical strain can also be accessed/probed by the ASOs, and this is probably the context in which this approach has the most importance relative to CRISPR-based methods.

1. Some care with nomenclature as well as a better introduction to the phage that is the focus of much of the paper would be useful. For instance:

a. Throughout, the authors refer to interfering with (or boosting) phage replication. That implies (to me at least) making more copies of the DNA, not necessarily the whole phage infection cycle. Please use some other term. What you are presumably measuring is the burst size (either after one round or three rounds of infection).

REPLY: The reviewer's point regarding the nomenclature is well taken. The effects in the CFU/PFU assay are cumulative and affected by delays in replication, the burst size and other effects that may act only in subsequent rounds of replication. We suggest using the term *plaque efficiency* to express this phenotype and have defined it early in the manuscript (p.5).

b. More information on the phage infection cycle would help the reader understand what is being done. This can include some idea of how long things take, approximate burst size, and avoid saying, when ASOs **block accumulation of plaque forming phage, that they are inhibiting phage replication**. As the authors note, an ASO to a lysis gene is sufficient to block plaque formation (although phage may be forming in the cell).

REPLY: As per the reviewer's request, we have added more information about the phage infection cycle and added an illustration (Fig. R7, new Extended Data Fig. 1a). We also clarified the different possible outcomes of phage gene silencing in phage-infected cells, e.g. a delay or block of replication, block of phage particle release, inhibition of second-round infection, etc. on p. 6 first paragraph. As discussed above, we now refer to ASO that block accumulation of plaque forming phages as "affecting phage plaque efficiency".

Fig. R7 | Schematic overview of the replication cycle of nucleus-forming jumbo phages.

c. Does this phage degrade its host DNA (as suggested in at least one review of this class of phage)? When in the life cycle? Given that in the DAPI images, all of the DNA appears to be in the nucleus, I assume that the bacterial DNA is gone. Or does the bacterial DNA get incorporated into the nucleus? This is critical to understanding and interpreting the transcriptomic studies in this paper.

REPLY: We agree with the reviewer that this is a critical open question. To the best of our knowledge, the assumption that ΦKZ induces host genome degradation is based on an observed loss of DAPI stain outside the phage nucleus as the infection progresses (Fig. 1a in Chaikeratisak et al. 2017 PMID: 28082593). We, too, have seen a diminished DAPI signal, but we have attributed this to contrast effects during imaging.

To address the question directly, we have now probed host or phage DNA on a Southern-dot-blot during the course of infection. We observed that the signal for the host DNA remains constant until 40 min p.i., while the signal of the phage DNA increases, as one would expect upon phage genome amplification (Fig. R8). These data strongly suggest that under our assay

conditions, the host genome is not degraded, allowing for RNA-seq studies throughout the infection cycle.

Fig. R8 | The *Pseudomonas* genome is not degraded upon ΦKZ infection. The ΦKZ and PAO1 genomes were detected with probes directed against genomic DNA.

2. A bit more attention to the details of what is being done and/or adding some information to figure legends would help the reader.

a. Fig. 1b, lines 130-132: Pretreatment with ASO (for how long? In a later panel (1c) pretreatment is adjusted to 30', but for this initial experiment, MOI and timing was not at all clear. By replication time, assume the authors mean the time from infection to phage release?

REPLY: We apologize for the missing details. In this initial experiment, the MOI was 5 and the pretreatment was 30 min. When referring to replication time for the control conditions, we indeed mean the time from infection to phage release. We have now defined the terms used and added details of the experimental conditions to the figure legends.

b. Fig. 1d: the text eventually says what the MOI is for this experiment, but adding it to the legend would be more useful. It would also be better to have a different arrow type/symbol for the EPI and the nucleus (and in other figures, particularly 5b).

REPLY: We have added the details to the experimental conditions in the figure legends and used distinct arrows for the phage nucleus and EPI vesicle. We thank the reviewer for pointing out the misleading labeling in **Fig. 5**, in which we investigate the effects of the ΦKZ155 knockdown on the phage nucleus maturation. Our RNA-seq data argue that upon ΦKZ155 knockdown, *nvRNAP*-dependent transcription takes place. Therefore, we think that the initial phage nucleus is formed, although this is difficult to visualise. We have therefore removed the arrows for the EPI vesicle and the initial phage nucleus in **Fig. 5b** (now **Fig. 5c**).

3. Given that the authors want to emphasize the general usefulness of the ASO approach, a bit more information on why particular protocols were chosen would be useful. For instance:

a. Fig. 2c, in which an ASO is silencing the *jukA* anti-phage defense system, uses repeated ASO pre-treatment and apparently multiple times after infection as well? Why is this repeated addition needed? What happens if it is only added once beforehand, and is there an explanation that would help someone hoping to use this in their own work?

REPLY: The JukA protein is a special case: as a constitutively expressed host anti-defense protein, we needed to deplete it before starting the phage infection. To ensure that silencing was maintained throughout the experiment, we re-applied the ASO. As suggested by the reviewer, we have now tested a one-time addition of ASO to the bacterial cells, followed by a 2.5 h incubation prior to infection with phage. This was sufficient to inhibit JukA and allow phage infection. Thanks for your suggestion! We have updated the experiment in the manuscript and replaced **Fig. 2c** (now **Fig. 2b**). Of note, since the manuscript describing the JukA/B system has now been published, the respective *bioRxiv* reference has been updated to Yuping L. et al. 2025 *Cell* (PMID: 40112800).

b. Fig. 2d, with the RNA phage PP7, ASOs were added every 30'. Again, why is this necessary in this case? Is it different for targeting a lysis gene or the polymerase (where there should, presumably be less RNA made)?

REPLY: In this experiment, treatment with 0.2 μ M of the control ASO reduced plaque formation, potentially because this RNA phage is more sensitive to ASO effects on the bacterial envelope. Therefore, we used a lower ASO concentration but multiple applications to ensure lasting inhibition. We have now included this information on p. 9 and in the figure legends. We did not see different sensitivity with different target genes, which is in line with our observation that transcript abundance does not affect targeting efficiency (**Fig. 3**), presumably because cellular ASO concentrations remain in excess over target transcripts. We now discuss this in the text on **p. 9**.

4. Figure 3: There is a great deal of data here, but that also leads to some questions.

a. Given the three cycles of infection, would classes of ASOs that lead to formation of defective phage particles, lysing the first infected cells but not the next round, be missed? Presumably for this, a high MOI in a single round to look possibly for loss of CFUs but decreased PFUs would be expected?

REPLY: Formation of defective phage particles in the first round would result in two rounds of diminished bacterial lysis and to fewer phage progeny in the two subsequent rounds. Therefore, classes of ASOs that lead to formation of defective phage particles would not be missed in our assay set-up. That said, the assay is designed to amplify the effect over multiple rounds of infection and therefore we currently cannot distinguish ASOs that cause formation of defective phage particles in the first round of infection from other mechanisms of action. Importantly, ASOs are effective over a prolonged period of time (>90 min), which covers more than one replication round (~50 min) (**Fig. R9**).

It is possible to design experiments to specifically screen for factors that lead to formation of defective phage particles. As suggested by the reviewer, this would require a high MOI and monitoring bacterial lysis as well as PFUs after a single round of infection. A time course experiment would be important to detect delays in lysis. While this might not be an ideal set-up for a screen, we believe that it is feasible because we have already shown for ChmA, PhuZ as well as in the RNA-seq screen that ASOs can be used effectively at an MOI of ~10.

Fig. R9 | ASO-based knockdown has long-lasting effects (>90 min). Cells were grown and treated at different time-points post inoculation, which resulted in different pre-incubation times as indicated. Subsequently, cells were infected with phage and after 3 h the CFU/PFUs were determined.

It would be useful to discuss the limits as well as the advantages of the approach used here with very low MOI.

REPLY: The CFU/PFU assay is designed to amplify ASO-knockdown effects on phage replication. It covers several complete rounds of replication and allows screening of ASO libraries. The limitation of the CFU/PFU assay is that we cannot discriminate at which stage the phage infection cycle fails. It could be host-takeover, phage replication, cell lysis or second round infection. A brief discussion of the limits and advantages of our approach with very low MOI has been added to the main text (discussion, last paragraph focusing on limitations, **p. 19**).

b. A few minor questions:

i. what is meant by omitting structural genes, since there seem to be capsid and tail genes.

REPLY: The reviewer is correct - we did target some structural genes, primarily as positive controls when setting up the ASO approach. In the subsequent screen, we largely focused on phage core genes with unknown functions. We have now corrected this statement in the manuscript (**p.10**).

ii. Why does one entry have parentheses around the results column (014).

REPLY: Φ KZ014 is only essential for Φ KZ plaque efficiency in the *Pseudomonas* clinical isolate PaLo44 (**Fig. 2c**). We have now explained this in a footnote.

iii. Are the levels of ASOs used for all of these (early vs. late genes, for instance) similar?

REPLY: Yes, we consistently used an ASO concentration of 6 μ M for the screen. This is now clarified in the methods and figure legends. We nevertheless advise titrating the ASO concentration (including the control) for optimal mRNA inhibition when embarking on follow-up studies on individual factors.

5. A major point in this paper is that unknown genes can be linked in terms of likely roles by their behavior, with transcriptome changes included as part of that behavior. That seems to result in the clustering shown in Fig. 5a. The two clusters that are surrounded by dotted lines are discussed a bit in the text; what about other apparent clusters?

REPLY: The clustering of transcriptomes at 30 min p.i., revealed a unique transcriptional signature of the *chmA*, *nvRNAP* and *picA* knockdowns, evident in **Extended Data Fig. 7c**. Several other ASO-targets clustered in **Fig. 5a**, but these other clusters do not show a strongly discriminating transcriptional signature. Knockdown of these targets likely causes more subtle effects that were nevertheless picked up by the tSNE analysis, because non-linear dimensionality reduction approaches like tSNE tend to overinterpret small effects. We did not dwell on the other cluster because they appeared less unique in the complementary hierarchical clustering of transcriptome changes (**old Extended Data Fig. 5b**). We have now clarified this in the manuscript (**p. 13**).

It is striking that the clustering is very different in extended fig. 5e. Maybe a comment on this would be useful.

REPLY: Consistent with **Fig. 5a**, ChmA, nvRNAP and PicA are forming a well-separated cluster in Ext. Data. Fig. 5e. That said, the clustering is less defined because at 15 min p.i. more ASOs cause similar but less defined patterns of transcript deregulation (compare **Ext. Data. Fig. 7a** (15 mpi) vs. **7c** (30 mpi)). We interpret these more subtle effects on early transcripts as a temporary delay in phage replication caused by these ASOs, which is no longer evident at 30 min. The non-ChmA/nvRNAP/PicA clusters we observe at 15 min p.i. compared to 30 min p.i are therefore likely related to a delay in phage nucleus formation. We now included the tSNE plots for the phage and host at both time points in **new Extended Data Fig. 7b,e,g** and discuss this point in the manuscript (**p. 13**).

If I understand the model for protein 155, the authors think it doesn't have any effect on phage genes because there is little or not DNA replication. I would think that would lead to much less RNA as well, or is all RNA made from an initial DNA copy?

REPLY: The reviewer is correct, we do observe that Φ KZ155 knockdown prevents phage nucleus maturation and halts the replication cycle prior to the expression of late genes. This is why we consider it as a new important checkpoint protein. Curiously, we observe that the transcriptional profile is not influenced by genome copy number at 30 min p.i., suggesting that all RNA is made from the initial DNA copy. At this stage, the packaging of phage genomes into capsids starts, so these genomes might no longer be available for transcription.

Are the other members of this cluster that have phage growth phenotypes either very late genes or might they also be involved in DNA replication?

REPLY: In this cluster, there are seven transcripts whose ASO-mediated inhibition reduced plaques (+++/++, red: Φ KZ120, -155, and -177, orange: Φ KZ049, -050, -059, -067) (**Fig. 5a, Supplementary Data 3**). Five of these seven were expressed after 20 min p.i.. One of these genes, Φ KZ050, encodes the DNA polymerase, so there might be a link to functions late in the replication cycle, but that requires further research.

I would have found more depth for a couple of stories rather than the mention of various genes of possible interest more useful here.

REPLY: We agree that our data sets include many candidates for compelling follow-up stories. Since the focus of the screen is on the jumbo phage core genome, these data will be valuable

for phage biology beyond Φ KZ. We understand the reviewer's interest in exploring genes with knockdown effects; we do highlight a few such cases on **p. 11 and 18** and we are already following up on some of these. However, we would like to keep the main focus of the present paper on establishing ASO-mediated gene silencing as a new approach to discover essential genes in the phage infection cycle, which will be particularly helpful for the many phage-host pairs with no or poor genetics.

Nevertheless, we also showcase how this approach can uncover previously overlooked essential phage factors, such as Φ KZ155, whose role in the phage replication cycle we are analyzing in more detail. Importantly, Φ KZ155 is emerging as a model substrate for protein import into the phage nucleus; it has enabled the discovery of multi-interface licensing of protein import into a phage nucleus in the form of a Φ KZ155-PicA (Nlp2-Imp1) complex (Kokontis et al. 2025 *Nature*, PMID: 39910297).

Our own discovery of Φ KZ155 of an essential molecular checkpoint without which the infection cycle freezes clearly supports the idea that this protein is a key player in jumbo phage biology. To corroborate its importance, we have now gone a step further and silenced its predicted homolog in the nucleus-forming phage Φ PA3, which also infects PAO1. This Φ PA3 homolog is called gp176 (gene 178). This predicted homolog displays 65% sequence identity and its overall fold and the RNase H domain are conserved. Using an ASO that targets the 5' region of gp176 mRNA with a different sequence from the anti- Φ KZ155 ASO, we observe the very same phenotype with Φ PA3 as with Φ KZ: loss of phage nucleus formation and phage replication (**Fig. R10, new Fig. 5c & new Extended Data Fig. 8b**).

Fig. R10 | Knockdown of the Φ KZ155 homolog of phage Φ PA3 gp176 prevents phage nucleus maturation and genome amplification in Φ PA3-infected cells. a. PAO1 cells were pre-treated with 8 μ M ASO against Φ KZ155 (Φ KZ) or gp176 (Φ PA3) for 30 min. Cells were infected with Φ KZ or Φ PA3, respectively, at an MOI=5 and incubated for 35 or 50 min, respectively, followed by chemical crosslinking and staining with DAPI to visualise DNA and FM4-64 to visualize membranes **b.** PAO1 cells were pre-treated with 8 μ M ASO against gp176. Cells were infected with Φ PA3 at an MOI=5 and incubated for indicated time followed by DNA extraction, dot-blotting, RNA was eliminated by alkaline treatment. The phage genomic DNA was detected with the radiolabelled oligo probe JVO-23279 followed by autoradiography. No host, no phage, and the control ASO served as controls.

6. One of the limits of the ASO approach as presented here is that it effectively knocks down translation, but that it is not necessarily useful to then dissect the function of the protein of interest. For instance, for the RNase H protein in Fig. 5, is it possible to complement an ASO (for a manipulable host) with a plasmid resistant to the ASO, expressing WT or mutant forms of the protein of interest?

REPLY: In addressing this point, we have now achieved to clone a FLAG-tagged version of Φ KZ155 into a plasmid with an arabinose inducible promoter, with a 5'UTR that is resistant to the ASO. Expression of Φ KZ155-FLAG rescued plaque efficiency by 2-3 fold after ASO-mediated knockdown of phage-encoded Φ KZ155 (Fig. R11, new Extended Data Fig. 8e,f). Intriguingly, expression of the catalytic mutant of the RNase domain (Φ KZ155D102N) also complemented the phenotype, indicating that it might not necessarily be the cleavage of DNA-RNA hybrids (the typical targets of RNase H proteins) that underlies the essential function of Φ KZ155.

We have updated the manuscript to reflect these results and now refer to Φ KZ155 as an RNase H-like protein rather than a ribonuclease. Given the high conservation of this domain, we still think that the function of Φ KZ155 relates to nucleic acids and phage genome replication, but it could also be a sensory rather than a catalytic function. We are grateful to this reviewer for encouraging us to try the complementation again (something we had tried in vain before) as this new plasmid will provide a valuable tool to dissect the domain architecture of Φ KZ155, determine the responsible residues for the observed essentiality of Φ KZ155 and its homologs, and perhaps their functions to the phage nucleus import machinery.

Fig. R11 | Heterologous expression of an ASO insensitive Φ KZ155 variant rescued the CFU/PFU phenotype after ASO mediated knockdown of endogenous Φ KZ155. a. PAO1 cells were transformed with a complementation plasmid encoding Φ KZ155 or the catalytically dead mutant (CDM). PAO1 cells were pre-treated with 6 μ M ASO against Φ KZ155 for 30 min. Cells were infected with Φ KZ at an MOI=0.0001, the complementation gene was induced with 0.2% arabinose, and cells were incubated for 180 min followed by CFU/PFU determination. **b.** PAO1 cells were transformed as in (a). PAO1 cells were pre-treated with 8 μ M ASO against Φ KZ155 for 30 min. Cells were infected with Φ KZ at an MOI=10, the complementation gene was induced with 0.2% arabinose, and the cells were incubated for 30 min followed by chemical crosslinking and staining with DAPI to visualise DNA and FM4-64 to visualize membranes.

7. The **data sets 2 and 3** could use much more informative legends. Color code as used in other figures? Identify what the columns are, where to look?

REPLY: We have now extended the legends for the two datasets.

In the discussion, protein 072 as a predicted nuclease that “does not require phage nucleus expression” (lines 458-460), with a reference to Supplementary Data 2. I assume this is because expression is similar with or without the ASO to ChmA? Maybe that could be spelled out.

REPLY: Yes, that is correct - the reason we suggest that expression of Φ KZ072 does not require the phage nucleus is that the protein is expressed early in the infection cycle and its expression is not inhibited by *chmA* KD. We have now added this explanation on **p. 19**.

8. As the authors note, the ASO screening here was done under one condition in one host, and some ASOs to core genes had no phenotype (Fig. 3). However, beyond not being in the right host, for instance, I would imagine that other means for “escaping” inhibition might be worth mentioning: redundancy in function (presumably testable with combinations of ASOs?), a requirement for only a small amount of a product (what is the leakiness of the ASOs), or possibly highly abundant, stable RNAs would be more difficult to inhibit. A bit of discussion of these complications would be useful.

REPLY: That is true. We now discuss these limitations in more detail in the manuscript after the discussion in a limitations section (**p. 19**). Of note, we have previously shown that the abundance of the target mRNA does not correlate with the efficacy of the targeting ASO (Popella L. et al., 2022 *NAR*, PMID: 35687096). Moreover, the widely used essential target mRNA *acpP* is among the most abundant mRNAs in *E. coli* but also one of the most susceptible targets known so far (El-Fateh M. et al., 2024 *Int J Antimicrob Agents*, PMID: 38185398; Popella & Jung et al., in preparation).

I couldn't find information on the degree to which the two ASOs used for each target agreed, providing some sense of how good coverage would be expected to be.

REPLY: All ASOs and their effects on CFU/PFU are listed in **new Supplementary Data 2**. For nearly half of the targets both ASOs showed a similar phenotype, while for the other half only one of the two ASOs showed an effect (with some of the ASOs being toxic to the host). We now state this information more explicitly in the manuscript (**p. 11**).

As a general recommendation, we advise that multiple ASOs be designed per target. Our new tiling experiment with *chmA* (**Fig. R3, new Extended Data Fig. 2**) shows that it is easy to design several effective ASOs against the same target. Please note that RXR-PNAs are easy to synthesize in one go, can be ordered from many different vendors or made in house using a peptide synthesizer, which is a common piece of equipment at research institutes and universities with a chemistry department.

Referee #3 (Remarks to the Author):

The article describes a system utilizing programmable antisense oligomers (ASO) technology to specifically silence genes of bacteriophage during infection. This represents an **innovative and potentially highly impactful methodology**. I concur with the authors that this approach holds significant promise for advancing both fundamental research on phage infection and its applications. The authors have done large-scale and impressive work, demonstrating the method's abilities for DNA and RNA phages as test subjects. **This study is indeed worthy of publication**. However, certain aspects require clarification.

The majority of the study demonstrating this technique is based on investigations of the giant bacteriophage phiKZ. Due to the extensive protection of phiKZ DNA from genetic manipulation at all stages of infection, this phage represents a challenging model; thus, the findings from this study are also crucial for understanding phiKZ development. And some aspects of data interpretation could benefit from adjustment.

1. The phiKZ phage encodes two distinct RNA polymerases (RNAPs), one of which, vRNAP, is injected into the cell along with the phage DNA. In this study, the PHIKZ176 vRNAP subunit (as described in Julie A. Thomas et al., 2016) was inactivated using ASO treatment. This important detail should be taken into account when interpreting the results.

REPLY: Thank you for raising this point. We now discuss that during the first round of infection, vRNAP is present in the virion and injected into the host cell. It is therefore not amenable to ASO inhibition during the initial infection. Any effects would only manifest in the second round of infection, when phage particles lack vRNAP and initial transcription is impaired (p.11).

2. Additionally, non-virion RNA polymerase (nvRNAP) subunits were also targeted in the study. However, only two subunits, PHIKZ55 and PHIKZ68, were explicitly noted as nvRNAP components. The ORF annotated as PHIKZ56.1 appears to be the terminal portion of (PDB: 7OGP_A; PSI-BLAST alignment with nvRNAP subunits in phiKZ-like phages), while PHIKZ73 represents a segment of the Gp71-73 subunit. The subunit composition of nvRNAP is documented in the works by Yakunina et al. (2015), Orekhova et al. (2019), and the structural studies of nvRNAP by Natàlia de Martín Garrido et al. (2021/2024). Based on this information, I question whether the using PHIKZ56.1 and PHIKZ73 are fitted for methods demonstration due to the design of ASO (rbs and start-codon), as the observed effects may relate not to protein synthesis inhibition but rather to mRNA splicing.

REPLY: Thank you for pointing us to the splicing of RNAP subunit transcripts. As described in the next reply, we have directly addressed potential effects on splicing and seen none. In addition, we can infer from our RNA-seq experiments that nvRNAP-dependent transcription is unaffected by those ASOs in question; in other words, the mRNA of the nvRNAP is made and spliced correctly.

As the reviewer notes, Φ KZ056.1 and Φ KZ073 are likely expressed as part of upstream ORFs Φ KZ055 and Φ KZ071, respectively, indicating that our ASO likely targeted misannotated start sites. This would suggest that they do not inhibit translation, explaining the mild phenotypes observed. We have therefore removed these ASOs from the screen in **Fig. 3**. We also briefly discussed the challenge of annotation and intron splicing in a new 'limitations' section on **p. 19**.

3. Continuing from the previous point, the presence of introns in coding sequences for subunits of both virion and non-virion RNA polymerases has been noted in several giant

bacteriophages related to phiKZ (Pieter-Jan Ceysens et al., 2014; Daria Lavysch et al., 2016). Within these introns, various potential homing nucleases are often predicted. Specifically, in this study, this applies to ORFs PHIKZ56 (nuclease within the intron of Gp55), PHIKZ72 (nuclease within the intron of Gp71-73), and PHIKZ179 (nuclease within the intron of the Gp180 vRNAP subunit). The impact of these nucleases may be linked to both nuclease synthesis inhibition and interference with splicing processes.

REPLY: This is a topic close to the heart of the corresponding author, who graduated with a thesis on prokaryotic self-splicing introns. As far as we understand, all the introns mentioned are group I introns that harbour ORFs of HNH homing endonucleases (reviewed in Nielsen H & Johannsen ST 2009 *RNA Biology*, PMID: 19667762). Different from maturases of group II introns that assist in splicing of their host introns, most endonucleases encoded by group I introns are considered to be relevant for the parasitic spreading of the introns and for cleavage of target DNA sites for intron integration (Birkholz EA et al. 2024 *Science*, PMID: 38963841).

To directly test whether splicing is affected by knockdown of those homing nucleases, we have synthesized cDNA (with hexamer primers, "+RT") from RNA of Φ KZ-infected PAO1 cells upon treatment with the respective ASO. These samples, as well as the controls without reverse transcriptase ("-RT"), were subjected to PCR amplification using (i) intron spanning primers in the flanking exons (upper panels) or (ii) a primer pair at the intron-exon junction (lower panels). This assay readily detected the spliced product (Fig. R12a,b). We have also cloned these RT-qPCR products and subjected them to Sanger sequencing; the splice sites are correct (Fig. R12, bottom). We conclude from these experiments that mRNA splicing is unaffected by these nuclease-targeting ASOs, suggesting that the nucleases in question have no immediate role in the splicing of the phage group I introns they reside in.

Fig. R12 | ASOs targeting genes close to splicing sites did not impact splicing efficiency. PAO1 cells were treated with the indicated ASOs and infected with Φ KZ. At 15 min p.i. RNA was isolated and cDNA generated using reverse transcription (+RT) with random hexamer primers. Subsequently, PCR was used with primers spanning the exon sites (top) or the splice site (bottom) for the HNH nuclease loci Φ KZ056 (a) and Φ KZ072 (b) where the HNH nucleus is in an intron and spliced out. The PCR products were Sanger sequenced and revealed the splice sites, which was in agreement with de Martín Garrido et al. 2021 *Nucleic Acids Research*, PMID: 34181731.

4. The results presented in lines 284-287—“However, upon *chmA* knockdown, reads of phage genes expressed at intermediate times were strongly depleted from 15 min p.i. onward, indicating that formation of the phage nucleus is required for expression of these genes by the nvRNAP”—are complementary to the transcription regulation scheme for *phiKZ* described in Antonova D. et al. (2023). *ChmA* synthesis inhibition likely blocks phage DNA release from the EPI vesicle, leaving it inaccessible to nvRNAP.

REPLY: Thank you for pointing this out; we agree, our data fit well with the idea that initial transcription is driven by vRNAP, while the second round of transcription is mediated by nvRNAP within the phage nucleus. We have now discussed this link and included the reference to Antonova D et al. 2023 *Viruses* on p. 3 of the manuscript.

5. The authors demonstrated that PHIKZ155 ASO treatment significantly impacted infection progression due to its involvement in phage DNA replication. However, another statement about its role in phage nucleus formation is not convincing enough. First, the quality of the fluorescence image (Fig. 5b) cannot confirm the position of phage DNA relative to cell boundaries. Please consider improving these images.

REPLY: Thank you for encouraging us to try harder and improve the resolution of this image. In repeating this experiment, we switched to imaging cells following an optimised chemical crosslinking with glutaraldehyde and formaldehyde that is used in EM (Alpers K et al. 2023 *mSystems*, PMID: 36786632). As a result, we can now observe a defined position of the phage DNA in phage infected cells after Φ KZ155 knockdown (Fig. R13, new Fig. 5c).

Fig. R13 | Cellular structures after Φ KZ155 KD. PAO1 cells were pre-treated with 8 μ M ASO against Φ KZ155 (Φ KZ) for 30 min. Cells were infected with Φ KZ at an MOI=5 and incubated for 35 min, followed by chemical crosslinking and staining with DAPI to visualize DNA and FM4-64 to visualize membranes.

Moreover, the “Curiously, the Φ KZ155GFP protein was barely detectable in the absence of phage infection, indicating that it requires the phage nucleus for its stability” appears somewhat speculative. If PHIKZ155GFP abundance is based solely on fluorescence imaging, the interpretation should consider that a concentrated molecular signal will appear brighter

than a signal diffused across the cell. Furthermore, PHIKZ155-GFP stability may rely on interaction with a single protein partner rather than the entire phage nucleus.

REPLY: The reviewer is correct - the molecular reason for the weak fluorescent signal is currently unknown and we must not overinterpret these imaging data. We have therefore removed this speculation.

Minor queries:

1. Different procedures were used for ASO treatment across experiments (e.g., single vs. multiple ASO administrations for phiKZ genes, bacterial genes, and PP7 infection). Could you clarify the rationale for these differences?

REPLY: We have included more details on the rationale of the different approaches within the main text (p. 5f. and p. 9) and the methods section (p. 27f.). For more details, please see our reply to reviewer #2, point 3 above.

2. Please update the reference for Antonova et al., “Genomic transfer via membrane vesicle: A strategy of giant phage phiKZ for early infection,” from BioRxiv to the Journal of Virology.

REPLY: Thank you; we have updated this reference to the published paper.

Editor’s comments

“... but as noted by referee #1 in particular, it would be crucial to validate the method in terms of its specificity and robustness across hosts”

As requested by the editor, we have performed experiments to demonstrate that our ASO approach works with additional phages and hosts (both different strains and species; details below). In addition, we show that targeting homologous genes in different phages yields the same phenotype (loss of phage nucleus formation and genome amplification), despite the fact that both, the used ASOs and the targeted mRNAs differ in sequence (**Fig. 5c**). We also show that ASO-mediated knockdown fully phenocopies deletions of the respective phage genes (**Fig. 1f, 2c**) and that a targeted essential gene can be complemented with a plasmid (**Fig. 5d**).

To expand the approach to more hosts, we used additional clinical isolates of *P. aeruginosa* for which genome sequences are available, so we could select some that are different from the PAO1 lab strain. Infection rates with these isolates differ, perhaps due to the presence of different defense systems, but the ASOs work invariable well (**Fig. R14, new Extended Data Fig. 4**).

To expand the approach to phages of different bacterial species, we have successfully silenced phage genes targeted *Erwinia* phage RAY (Prichard et al. 2023 Cell Reports, PMID: 37120812), which also infects the plant pathogen, *Pantoea agglomerans* (**Fig. R15, new Fig. 2e**). These data are very promising against the background of efforts to use phages against crop pathogens.

Lastly, we demonstrate that our ASO approach works in the primary model organism of Gram-positive bacteria, *Bacillus subtilis*. A KFF-coupled ASO targeting the *gp146* gene (DNA polymerase) of the widely used SPO1 phage reduced PFU by 1,000-10,000 fold (**Fig. R15, new Fig. 2f**). This is also exciting because the DNA genomes of SPO1-like phages are heavily

modified, which renders genetic manipulation very challenging. To the best of our knowledge, this also happens to be the first report of successful use of antisense PNA in *B. subtilis*, opening up potential applications outside phage biology as well.

In closing, we believe that the ASO approach introduced here provides a method with specificity and robustness across hosts.

Fig. R14 | ASOs can be used to target diverse clinical *Pseudomonas* isolates. **a.** PaLo8/9/39/44 cells were pre-treated with 6 μM ASO against *chmA*. Cells were infected with ΦKZ at an MOI=0.0001 and incubated for 180 min followed by CFU/PFU determination. **b.** PaLo8/9/39/44 cells were pre-treated with 6 μM ASO against *chmA*. Cells were infected with ΦKZ at an MOI=5 and incubated for 20 min followed by quantification of ChmA levels via immunoblotting with an antibody against ChmA. **c.** PaLo8/9/39/44 cells were pre-treated with 8 μM ASO against *chmA*. Cells were infected with ΦKZ at an MOI=10 and incubated for 40 min followed by chemical crosslinking and staining with DAPI to visualise DNA and FM4-64 to visualize membranes.

Fig. R15 | ASO-based knockdown of RAY and SPO1 phage-infections in *P. agglomerans*, and *B. subtilis*, respectively. **a.** *Pantoea agglomerans* cells were pre-treated with 6 μ M ASO against *gp206 (chmA)* for 30 min. Cells were infected with RAY at an MOI=0.0001 and incubated for 300 min followed by CFU/PFU determination. **b.** *B. subtilis* 168 cells were pre-treated with 6 μ M ASO against the transcript of gene 31 (DNA polymerase) for 30 min. Cells were infected with SPO1 at an MOI=0.0001 and incubated for 180 min followed by CFU/PFU determination.

Other changes

The order of authors in the author list has been adjusted to acknowledge the additional contributions of authors Buhlmann and Zhu during the revision of our manuscript.

We thank all reviewers for their positive assessment of our revision and for their continued support. Please find our response to all remaining comments below.

Point-to-point reply to the referees' comments (ms. no.: 2024-09-18911A)

Referee #1 (Remarks to the Author):

The authors have now added a substantial amount of novel data to the manuscript, which address all my previous concerns. I congratulate them on a truly excellent piece of work and I recommend it for publication.

REPLY: Thank you for this evaluation and for your clear recommendation that this manuscript be published.

Referee #2 (Remarks to the Author):

In this revised version of a previously reviewed ms by J. Vogel and collaborators, the authors present extensive data supporting the use of ASOs to analyze unknown gene functions in phage, in particular in nucleus-forming phage in *Pseudomonas*. They show all of the characteristics one would want for dissecting functions without the need for the bacterial cell to accept genetic material. This is a novel, well-developed and highly useful approach that will be important in the growing interest in probing gene function in a range of bacteria and bacteriophages. The revised manuscript makes the logic of the various experiments much clearer as well as adding further examples demonstrating the breadth of the approach. The comments here are for a few additional minor clarifications.

1. The result shown in rebuttal fig. R8 would be relevant to include in the paper (as a supplemental figure, briefly discussed in the text).

REPLY: We have now included Fig. R8 (showing that the host genome is not degraded upon Φ KZ infection) as **Extended Data Fig. 8f** and mentioned this finding in the manuscript (**line 489f**).

2. In Fig. 1d, the CFU don't seem to decrease when the PFUs are inhibited with ASO, while in other experiments, there is clear evidence of killing of the host when the phage can grow (extended data figure 1). Different protocol? Maybe worth a mention.

REPLY: In **Fig. 1d**, we do observe a decrease in CFUs for the lower dilutions under the ASO control condition (ctrl.), i.e. when the phage propagates. This is consistent with **Extended Data Fig. 1**, although the effect is more evident for higher phage titers and longer replication times, please see **Ext. Data Fig. 1i**. We have now mentioned these dependencies in the text (**line 156ff**).

3. The authors point out (text, lines 217-223, extended Fig. 3) that inhibition of translation of a given gene does not affect expression of other genes in that operon, suggesting that in this organism, there is not a robust mechanism for degradation or termination in the absence of translation. The authors note that mRNAs are depleted in *E. coli*, however (lines 371-374). This is an important and interesting difference that may be worth a bit more discussion (and/or note if there are parallel *E. coli* experiments to mention with lines 217-223).

REPLY: Certainly, this is an interesting observation, and therefore we mention it in the manuscript. However, we do caution against general conclusions. For more context, in our previous study in *E. coli* (Popella L et al. 2022 *NAR*), we observed successful inhibition of translation for 10 different mRNAs upon ASO targeting but only ~50% of them also showed induced target mRNA decay. One important take-away from this study was that ASOs can effectively silence mRNAs without concomitant changes in mRNA levels.

In the current study with phage mRNAs in *Pseudomonas*, we observe induced mRNA decay in even fewer cases (10%, based on our RNA-seq data; **Extended Data Fig. 7, Supplementary Data 4**). This is relevant because—as we show by proteomics—ASOs can selectively repress translation of individual cistrons within polycistronic mRNAs. However, whether this means that *translationally-inhibited* mRNAs of this phage are less prone to accelerated degradation or whether the host (*Pseudomonas*) generally lacks a robust mechanism whereby translation inhibition triggers mRNA decay requires a more systematic study, looking at chromosomally encoded *Pseudomonas* mRNAs as well, not just phage mRNAs. For now, we think it is important to mention the possibility that ASOs might induce mRNA decay, as this has consequences for targeting polycistronic transcripts. We have tried to make this clearer in the manuscript:

Line 385ff.: *Interestingly, the chmA transcript itself was unaffected by the ASO treatment, suggesting that, in this case, successful translational inhibition does not entail mRNA depletion (Fig. 4b). This would also explain our observation that ASOs targeting polycistronic phage genes are cistron-specific (Extended Data Fig. 3c-e). However, it is important to note that accelerated mRNA decay was observed for several ASO-targeted mRNAs in E. coli (Popella et al. 2022), highlighting the importance of monitoring mRNA levels when targeting individual cistrons within polycistronic transcripts.*

4. The ability to complement back with an ASO-resistant gene is, as the authors note, extremely useful, and this is a very nice addition (although only possible with organisms that are amenable to the genetics, introducing the ectopic copy, of course). I'd suggest making the figure (5d) a bit clearer, however; I had to go back and look at everything carefully to understand what I was seeing. Emphasize in legend and maybe by making the plasmid copy of the gene in the schematic a different color (not red, color of ASO), that this doesn't contain the target for the ASO. It is easy to miss the p, or not be thinking why the plasmid version is resistant. Color change could then also be used with the sequence to make it clearer what goes with what.

REPLY: We thank the reviewer for this suggestion and have made **Fig. 5d** clearer by changing the color-coding, as suggested.

5. It was unclear to me why the information in Extended Fig. 8c-f wasn't in the results. Doesn't seem like it belongs in the discussion, even if it is not fully explored thus far.

REPLY: We now mention the CDN mutant data in the result section (**p. 15 line 468f**).

Referee #3 (Remarks to the Author):

Thank you to the authors for their work and for addressing the comments raised by all reviewers. The expanded discussion on the ASO design and the demonstration of its broad applicability significantly

improve the manuscript and enhance its value for future wide use. Below, I provide my comments on the authors' replies.

Thank you for your assessment and additional constructive input.

1. Unfortunately, the potential impact of this gene's inhibition on the activity of vRNAP is still not mentioned in the manuscript. The observed reduction in ChmA levels could be associated with a decreased amount of active vRNAP in the phage particle, which may reduce the efficiency of ChmA transcription during the second and third rounds of infection in your screening experiment. Supporting this hypothesis is the fact that in the single-round infection experiment, which you conducted to collect material for RNA-seq, there is no observed difference in the amount of phiKZ transcript (Fig. 4c), which would be expected if the decrease in ChmA were a direct effect.

line 325: as well as the uncharacterised core genes Φ KZ147, -153, -161, -176

REPLY: We sincerely apologize for this omission. We did agree in our previous rebuttal to discussing the special case of the vRNAP component Φ KZ176 in more detail, but accidentally ended up not including the information in the revised manuscript. As the reviewer correctly states, ASOs targeting the vRNAP subunit Φ KZ176 do not seem to affect overall Φ KZ transcription in the first round of infection (Fig. 4c). This makes sense, since the vRNAP is present in the virion and injected upon infection. In multi-round infection experiments, ASO-mediated mRNA knockdown of the vRNAP can take effect in the second and third round and reduce the initial vRNAP-dependent transcription of *chmA* from the EPI vesicle. Consistent with this idea, we do observe lower ChmA levels in our screen, which we read out after three rounds (Fig. 3). We have expanded the relevant section accordingly (line 332ff.), see below:

*In addition to CFU/PFU quantitation, we used ChmA levels as an additional readout, quantifying ChmA protein by immunoblotting after three rounds of replication. We observed reduced levels of ChmA for 11 additional target genes, although we saw no substantial changes of the infection rate as measured via CFU/PFU (Fig. 3). Examples include a putative RAD2/SF2 helicase (Φ KZ075), a predicted DEAD/DEAH box helicase (Φ KZ203), the macro domain-containing protein Φ KZ104, the virion RNAP subunit Φ KZ176 (Thomas et al. 2016), as well as the uncharacterised core genes Φ KZ147, -153, -161 and the non-core genes Φ KZ283, and -286. These genes can be expected to benefit phage fitness in more competitive situations such as non-laboratory environments or in *P. aeruginosa* strains with a different repertoire of defence systems. In the case of the vRNAP subunit Φ KZ176, the relatively modest effect on phage fitness is likely due to the fact that this protein is already present in the phage virion and injected into the host cell upon attack, hence ASO-mediated mRNA inhibition will not take effect in the initial round of infection. However, in the second and third round, the ASO can inhibit the vRNAP transcript, reducing Φ KZ176 protein levels. This, in turn, leads to reduced ChmA levels, since early transcription of *chmA* from the EPI vesicle is vRNAP dependent (Ceyssens et al. 2014, Antonova et al. 2023, Armbruster et al. 2025).*

2. First, I would like to draw the authors' attention to the fact that both transcripts in question are still present in Fig. 3. Although the authors acknowledge in their response to the reviewer that in these particular cases the ASO effect likely does not result from translation inhibition ("This would suggest that they do not inhibit translation, explaining the mild phenotypes observed"), in the main text of the manuscript these "exons" (to the extent that this term can be applied to phage genomes, though it

seems the most accurate descriptor of these ORFs) continue to be discussed as individual genes, alongside other protein-coding genes. Consequently, the effects of ASO targeting them could be interpreted by readers as effects on translation. However, in these two cases, it remains unclear what the ASOs are targeting and how the observed effects should be interpreted.

Line 326: "and the non-core genes Φ KZ056.1"

Lines 394–397: "At 30 min p.i., ASO-mediated depletion of ChmA, the nvRNAP subunit Φ KZ055, the nvRNAP sigma factor Φ KZ068, PicA (Φ KZ069), Φ KZ056.1, -122, -124, and -151 showed a strong effect on phage transcription (Fig. 4c)"

Lines 411–413: "Equally interesting are the knockdowns of Φ KZ073 and -082, which have no general effect on the PAO1 transcriptome but specifically induce transcription of the Pf4 prophage locus"

Lines 416–417: " Φ KZ might use the Φ KZ073 and Φ KZ082 proteins as part of a specific inhibitory mechanism against the Pf4 prophage to ensure its own propagation"

REPLY: We agree with the reviewer that, as stated in their original review, Φ KZ056.1 and Φ KZ073 are not suitable for demonstrating ASO-mediated mRNA inhibition, because these are mis-annotated segments of other genes. Therefore, as previously discussed, the respective ASOs are unlikely to affect translation. As suggested by the reviewer, we have now removed Φ KZ056.1 and Φ KZ073 from **Fig. 3**, the associated text and elsewhere. As these are specialized cases, their removal affects neither the validity of our approach nor the main conclusion of our manuscript.

3. Thank you for checking the potential ASO effect on splicing. It might be helpful to map the splicing regions directly onto the corresponding sequences, indicating the positions of the ASOs and primers used in the analysis. This could allow for more concrete conclusions regarding splicing efficiency and ASO design in the presence of introns within phage genes. For example, based on the provided images, different ASOs appear to result in varying amounts of unspliced RNA (lower panels in the gels). Notably, the highest levels of unspliced RNA are observed in the control samples without ASO treatment in both cases.

REPLY: We appreciate the reviewer's attention to these details. Below we indicated the sites of the ASOs designed to target Φ KZ056, -056.1 and -072 and -073 on the respective gene loci (**Fig. R1**). We agree with the reviewer on slightly higher levels of unspliced precursor RNA in the control sample. However, what is most important for the present question is the signal for the spliced mRNA (now indicated by green arrows), which is unaffected by all ASOs.

Likewise, our transcriptomics data show that nvRNAP-dependent transcription occurs, indicating that the synthesis of the nvRNAP is not overtly affected by these ASOs (**Fig. 4c and Extended Data Fig. 7**). Thus, while we cannot exclude a mild effect on splice precursor accumulation, mRNA splicing itself and translation of the spliced mRNA is largely unaffected.

Fig. R1 | ASOs targeting genes close to splicing sites. a. Linear sequences of group I introns in Φ KZ056 and -072 loci. ASOs (JVPNA) targeting the HNH nucleases and downstream nvRNAP exon are shown. The dashed black lines indicate the start of the auto-annotated ORFs (NCBI). **b.** Fig. R12 (previous rebuttal). PAO1 cells were treated with the indicated ASOs and infected with Φ KZ. At 15 min p.i. RNA was isolated and cDNA generated using reverse transcription (+RT) with random hexamer primers. Subsequently, PCR was used with primers spanning the exon sites (top) or the splice site (bottom) for the HNH nuclease loci Φ KZ056 (left) and Φ KZ072 (right).

This observation may suggest that the "intron–exon" structure, present in both subunits of the nvRNAP and in one subunit of the vRNAP (including both phage RNA polymerase subunits that carry the essential DFDGD motif for RNA polymerization activity), could play a role in regulating phage genome transcription.

REPLY: We thank the reviewer for pointing out the exciting possibility that the "intron–exon" structure present in both subunits of the nvRNAP and in one subunit of the vRNAP could play a role in regulating phage genome transcription. In principle, it should be feasible to address this question by developing an ASO design strategy for targeting splice sites or RNA structural elements of these group I introns (without inducing potential precursor mRNA decay). Having said this, this is a special application that would require extensive monitoring of splicing efficiencies and protein output (using antisera yet to be produced). This is not directly related to the scope of the current study, which is focused on establishing ASOs as translational inhibitors of phage mRNAs.

4. Let me clarify the point more precisely. According to Ceyssens et al. 2014, three classes of phage genes are distinguished in ϕ KZ: early, middle, and late, each transcribed from distinct promoter sequences. Based on the model proposed by Antonova et al., 2023, only the transcription of late genes is strictly dependent on the formation of the mature nucleus, whereas the middle gene class, which although transcribed by nvRNAP, initiates before complete phage nucleus maturation and likely involves a form of nvRNAP that differs in composition from that associated with late promoters.

This raises an important question: does ChmA depletion affect transcription of both middle and late genes, or only the late ones? Unfortunately, Extended Data Figure 6 does not provide definitive evidence that both classes are blocked. Based on the figure, phage transcripts accumulate up to 25 minutes post-infection, albeit at reduced levels compared to the control. This decrease might reflect a lack of late gene transcription rather than a complete arrest of phage transcription.

This distinction could be crucial for understanding the dynamics of infection. If middle gene transcription still occurs upon ChmA translation knockdown, it would imply that DNA release from EPI vesicles can proceed, at least partially, in the absence of significant accumulation of the major nuclear shell protein. Such a finding could offer new insights into the stability of EPI vesicles.

REPLY: Following the reviewer's suggestion, we took a closer look at the phage transcripts present after ChmA knockdown (**Fig. R2 below**). Interestingly, we find that most of these transcripts represent early genes transcribed by the vRNAP, likely because the viral genome remains in the EPI vesicle and vRNAP transcription continues in the absence of the major nuclear shell protein. We appreciate the reviewer's point that distinguishing between middle and late genes would be informative. However, our high-resolution RNA-seq time-course concludes at 35 minutes and can therefore only answer part of this specific question. This is to say that we can discriminate between middle and late genes based on **Extended Data Fig. 6d** (middle: clusters c, e, and late: cluster f), but do not observe a striking difference between these two categories as neither is transcribed efficiently upon ChmA knockdown.

To accommodate the reviewer's request, we have now updated **Extended Data Fig. 6a,b** to include the early and middle/late transcript classification. Given the challenges of accurate gene categorization of phage transcripts, we would not feel comfortable drawing strong conclusions regarding the stability of the EPI vesicle at this point. This would require a more thorough investigation that is beyond the scope of the current manuscript.

Fig. R2 | Classification of transcripts after knockdown of ChmA. Early, middle/late coding sequence (CDS) transcript fractions after ChmA knockdown.

5. No additional comments. Thank you for the answer.

REPLY: Thanks again for a very thoughtful review and constructive comments on our manuscript.

Point-to-point reply to the referee's comments (ms. no.: 2024-09-18911B)

Referee #3 (Remarks to the Author):

I would like to thank the authors for addressing my questions and for their careful attention to my comments.

I would like to add a brief remark regarding the authors' response to Reviewer #2 concerning the absence of bacterial DNA degradation during phiKZ phage infection. A similar observation was previously reported in the paper by Yana A. Danilova et al. 2020. Therefore, it appears clear that no degradation of host cell DNA occurs in cells during phiKZ infection, at least in the first half of the infection cycle. The reference to this paper, together with the observations described in the present manuscript, which show the same results, will likely help resolve the uncertainty on this issue that has existed in the literature.

In conclusion, I strongly recommend the manuscript for publication.

REPLY: We thank the reviewer for their positive assessment of our revision and their recommendation to publish our study. We have added the reference as suggested (ref. 48; cited on p. 12, line 361 in this context).